# OPTIMAL LEARNING OF KERNEL LOGISTIC REGRESSION FOR COMPLEX CLASSIFICATION SCENARIOS

**Hongwei Wen & Annika Betken**
Faculty of Electrical Engineering, Mathematics and Computer Science
University of Twente
Enschede, The Netherlands
{h.wen,a.betken}@utwente.nl

**Hanyuan Hang**
Hong Kong Research Institute
Contemporary Amperex Technology (Hong Kong) Limited
Hong Kong Science Park, New Territories, Hong Kong
hanyuan0725@gmail.com

## ABSTRACT

Complex classification scenarios, including long-tailed learning, domain adaptation, and transfer learning, present substantial challenges for traditional algorithms. Conditional class probability (CCP) predictions have recently become critical components of many state-of-the-art algorithms designed to address these challenging scenarios. Among kernel methods, kernel logistic regression (KLR) is distinguished by its effectiveness in predicting CCPs through the minimization of the cross-entropy (CE) loss. Despite the empirical success of CCP-based approaches, the theoretical understanding of their performance, particularly regarding the CE loss, remains limited. In this paper, we bridge this gap by demonstrating that KLR-based algorithms achieve minimax optimal convergence rates for the CE loss under mild assumptions in these complex tasks, thereby establishing their theoretical efficiency in such demanding contexts.

## 1 INTRODUCTION

Classification is a key problem in machine learning that involves predicting categorical labels from input features. It plays a crucial role in applications ranging from image and speech recognition to medical diagnoses and financial forecasting. In real-world scenarios, classification is often complicated by various challenges, like imbalanced label distributions, missing labels, and limited labeled data. These issues make direct classifier training particularly challenging, leading to the development of specialized approaches such as long-tailed learning (Kim et al., 2020; Lin, 2023; Chen & Su, 2023), domain adaptation (Lipton et al., 2018; Bai et al., 2022; Wen et al., 2024), and transfer learning (Radford et al., 2021; Zhu et al., 2024; Li et al., 2024).

A common strategy to address these challenges is to first estimate conditional class probabilities (CCPs) on related labeled datasets, followed by reweighting or adjusting these estimates for the target problem's CCP estimator and classifier. Since accurate CCP estimation on related datasets is crucial for the algorithm's practical performance, it is important to use methods that are proficient in CCP estimation across various tasks. Logistic regression (LR) (Kleinbaum & Klein, 2010), specifically designed for modeling CCPs in classification problems, estimates CCPs by fitting a logistic function that achieves linear separation of input data through the minimization of cross-entropy (CE) loss (or equivalently, logistic loss). However, LR is inherently limited to linearly separable data and performs poorly with non-linearly separable data. To address this limitation, kernel logistic regression (KLR) (Wahba et al., 1995; Zhu & Hastie, 2005) extends LR by incorporating the kernel trick, which maps data into a high-dimensional feature space using kernel functions. This approach enables linear separation in that space, effectively handling non-linearly separable data. Unlike other kernel-based classifiers, such as support vector machines or Gaussian processes, KLR not only han-

dles complex data structures but also provides interpretable CCP estimates without additional steps or approximations.

The existing literature offers limited theoretical exploration of CCP-based algorithms for classification in complex scenarios, particularly concerning the CE loss. As CCP estimates approach zero, the CE loss becomes unbounded, posing significant challenges to error analysis. This unboundedness invalidates many standard oracle inequalities typically used for error analysis under bounded losses (see Steinwart et al. (2006); Steinwart & Scovel (2007); Blaschzyk & Steinwart (2018; 2022)). Recently, some studies have attempted to address these issues by establishing new oracle inequalities for the CE loss. For instance, Farrell et al. (2021) assumed the true CCP function's range to be $[1/(e+1), e/(e+1)]$ instead of $[0,1]$, ensuring that the CE loss remains bounded. However, this assumption does not hold for many practical datasets. Similarly, Bos & Schmidt-Hieber (2022) truncated the unbounded excess CE loss to ensure boundedness and analyzed the resulting expected loss, but the derived oracle inequalities and convergence rates do not extend to the unbounded CE loss. More recently, Zhang et al. (2024) proposed a bivariate function to address the unboundedness of the CE loss in binary classification. However, their proof techniques for verifying the variance bound in the oracle inequality are not directly applicable to multi-class classification.

In this paper, we investigate the convergence rates of algorithms based on CCP estimation using KLR in complex classification scenarios, such as long-tailed learning, domain adaptation, and transfer learning. We begin by showing that the excess CE risk in these complex scenarios can be reduced to the excess CE risk encountered in standard classification. A significant challenge in this analysis arises from the unbounded nature of the CE loss derived from the true CCP function. To tackle this challenge, we decompose the CE loss into upper and lower components based on the true CCP values, proposing a novel error decomposition for the excess risk of KLR. This decomposition allows the unbounded terms to cancel out, resulting in bounded sample error terms. We can further address these sample errors by establishing a new oracle inequality for KLR. Next, we turn our attention to the approximation error associated with KLR. We construct an approximation of the target function that corresponds to the logarithm of the ratio of two CCP functions. However, as the true CCPs approach zero, this target function becomes unbounded. To resolve this issue, we introduce a bounded version of the target function: specifically, the logarithm of the ratio of truncated CCPs that exceed a predetermined threshold. Our findings reveal that selecting a higher threshold results in an approximation that significantly diverges from the target function, while a lower threshold yields a closer approximation but introduces a large upper bound on the CE loss. By carefully selecting an appropriate threshold, we can effectively balance these trade-offs, achieving an optimal approximation error with respect to the CE loss.

The contributions of this paper are summarized as follows.

*(i)* We propose a novel decomposition framework that separates the CE loss into upper and lower components, facilitating the decomposition of the excess CE risk of KLR into approximation and sample error terms. This innovative approach effectively tackles the analytical challenges posed by the unbounded nature of the CE loss, paving the way for the derivation of a new oracle inequality.

*(ii)* We establish an upper bound for the approximation error terms of KLR with respect to the CE loss. To achieve this, we design a bounded approximation function that closely approximates the unbounded true CCP function while remaining above a specified positive threshold. By carefully selecting this threshold, the constructed approximation minimizes the resulting error.

*(iii)* We derive an overall upper bound for the excess CE risk of KLR by integrating the results from *(i)* and *(ii)*. By judiciously selecting the appropriate KLR parameters, we further establish the convergence rates for KLR in the aforementioned complex classification scenarios.

*(iv)* We derive lower bounds for the excess CE risk, which align with the convergence rates of KLR established in *(iii)*. This consistency demonstrates the minimax optimality of these rates, underscoring the effectiveness of existing approaches based on CCP estimation.

*(v)* We conduct numerical experiments to demonstrate the effectiveness of CCP-based algorithms and to empirically validate the minimax optimal convergence rates established in *(iii)* and *(iv)*.

## 2 COMPLEX CLASSIFICATION SCENARIOS

**Notations.** We use the notation $a_n \lesssim b_n$ (or $a_n \gtrsim b_n$) to indicate that there exists a constant $c > 0$ such that $a_n \leq c b_n$ (or $a_n \geq c b_n$) for all $n \in \mathbb{N}$. We write $a_n \asymp b_n$ if there exists a positive constant $c \in (0,1]$ such that $c b_n \leq a_n \leq c^{-1} b_n$ for all $n \in \mathbb{N}$. For a natural number $M$, we denote $[M] :=$

$\{1, 2, \ldots, M\}$. For $a, b \in \mathbb{R}$, we define $a \wedge b := \min\{a, b\}$ and $a \vee b := \max\{a, b\}$, representing the smaller and larger values of $a$ and $b$, respectively. We denote the $(M-1)$-dimensional simplex as $\Delta^{M-1} := \{\theta \in \mathbb{R}^M : \sum_{m=1}^M \theta_m = 1, \theta_m \geq 0, \, m \in [M]\}$.

**Standard Classification Scenario.** In standard classification problems, we observe i.i.d. data $D := (X_i, Y_i)_{i=1}^n \in (\mathcal{X} \times \mathcal{Y})^n$ drawn from an unknown distribution $P$ on $\mathcal{X} \times \mathcal{Y}$, where $\mathcal{X} \subset \mathbb{R}^d$ denotes the input (feature) space and $\mathcal{Y} := [M]$ represents the output (label) space. Let $p(y|x)$ be the *conditional class probability* (*CCP*) function, and let $p(x)$ and $p(y)$ denote the *marginal density functions* of $X$ and $Y$, respectively. Based on the observations $D$, our goal is to find a classifier $\widehat{h} : \mathcal{X} \to \mathcal{Y}$. In practice, an important approach for training a classifier involves fitting a CCP estimator $\widehat{p}(y|x)$ and then inducing the *plug-in* classifier defined as $\arg\max_{y \in [M]} \widehat{p}(y|x)$.

For a loss function $L : \mathcal{Y} \times \mathbb{R} \to \mathbb{R}$, the expectation of $L(y, \widehat{p}(\cdot|x))$ with respect to $P$, denoted as $\mathcal{R}_{L,P}(\widehat{p}(y|x)) := \mathbb{E}_P[L(Y, \widehat{p}(\cdot|X))]$, is referred to as the *risk*. The smallest corresponding risk is termed the *Bayes risk* and is denoted as $\mathcal{R}_{L,P}^* := \inf\{\mathcal{R}_{L,P}(\widehat{p}(y|x)) : \widehat{p}(y|x) \text{ is measurable}\}$. Let $D := n^{-1} \sum_{i=1}^n \delta_{(X_i, Y_i)}$ be the Dirac measure, and the expectation of a function $h$ with respect to $D$ is given by $\mathbb{E}_D h := n^{-1} \sum_{i=1}^n h(X_i, Y_i)$. The expectation of the loss function $L(y, \widehat{p}(\cdot|x))$ with respect to $D$ is referred to as the *empirical risk* and is expressed as $\mathcal{R}_{L,D}(p_f(y|x)) := \mathbb{E}_D[L(Y, p_f(\cdot|X))] = n^{-1} \sum_{i=1}^n L(Y_i, p_f(\cdot|X_i))$. The *cross-entropy* (*CE*) *loss* is defined as $L_{\mathrm{CE}}(y, \widehat{p}(\cdot|x)) := -\log \widehat{p}(y|x)$, while the *classification loss* is defined by $L_{\mathrm{class}}(y, \widehat{p}(\cdot|x)) := \mathbf{1}\{y = \arg\max_{m \in [M]} \widehat{p}(m|x)\}$. Since $\mathcal{R}_{L_{\mathrm{CE}},P}^* = \mathcal{R}_{L_{\mathrm{CE}},P}(p(y|x))$, the Bayes risk $\mathcal{R}_{L_{\mathrm{CE}},P}^*$ can be achieved by the true CCP function $p(y|x)$. Therefore, the excess CE risk $\mathcal{R}_{L_{\mathrm{CE}},P}(\widehat{p}(y|x)) - \mathcal{R}_{L_{\mathrm{CE}},P}^*$ measures the accuracy of the CCP estimator $\widehat{p}(y|x)$ in estimating the true CCP function $p(y|x)$. The calibration inequality presented in (Steinwart & Christmann, 2008, Theorem 8.29) establishes a relationship between the excess classification risk and the excess CE risk. More precisely, it states

$$\mathcal{R}_{L_{\mathrm{class}},P}(\widehat{p}(y|x)) - \mathcal{R}_{L_{\mathrm{class}},P}^* \leq 2\sqrt{2}(\mathcal{R}_{L_{\mathrm{CE}},P}(\widehat{p}(y|x)) - \mathcal{R}_{L_{\mathrm{CE}},P}^*)^{1/2}. \tag{1}$$

This illustrates that if the CE risk of a CCP estimator converges to the CE risk of the true CCP function, then the classification error of its induced classifier will converge to the smallest possible classification error. Thus, CCP estimation offers more informative insights than mere label prediction, enabling improved handling of uncertainty and enhancing decision-making processes.

**Complex Classification Scenarios.** Unlike the standard classification setting, complex classification scenarios involve labeled samples $D_p := (X_i, Y_i)_{i=1}^{n_p}$ drawn from a distribution $P$ on $\mathcal{X} \times \mathcal{Y}$, while inference is required for a different distribution $Q$ on the same space. In this paper, we primarily investigate three complex classification scenarios in which the class-conditional probability $q(x|y)$ is assumed to be identical to the class-conditional probability $p(x|y)$, that is,

$$p(x|y) = q(x|y), \qquad \forall \, x \in \mathcal{X}, y \in \mathcal{Y} = [M]. \tag{2}$$

Using Bayes' theorem and the assumption in Eq. (2), the CCP function $q(y|x)$ for the test data can be expressed as

$$q(y|x) = \frac{q(y)q(x|y)}{\sum_{m=1}^M q(m)q(x|m)} = \frac{q(y)p(x|y)}{\sum_{m=1}^M q(m)p(x|m)} = \frac{q(y)p(y|x)p(x)/p(y)}{\sum_{m=1}^M q(m)p(m|x)p(x)/p(m)}$$
$$= \frac{(q(y)/p(y))p(y|x)}{\sum_{m=1}^M (q(m)/p(m))p(m|x)} = \frac{w^*(y)p(y|x)}{\sum_{m=1}^M w^*(m)p(m|x)},$$

where $w^*(y) := q(y)/p(y)$ is referred to as the *class probability ratio*. Let $\widehat{w}(y)$ and $\widehat{p}(y|x)$ denote the estimators for $w^*(y)$ and $p(y|x)$, respectively. Then we can derive the CCP estimator for the test distribution as

$$\widehat{q}(y|x) = \frac{\widehat{w}(y)\widehat{p}(y|x)}{\sum_{m=1}^M \widehat{w}(m)\widehat{p}(m|x)}. \tag{3}$$

Consequently, if the CCP estimation $\widehat{p}(y|x)$ is available, it is sufficient to estimate $w^*(y)$ to obtain $\widehat{q}(y|x)$. In the following sections, we will present approaches to estimate $w^*(y)$ for classification problems in three different complex scenarios that satisfy Eq. (2).

## 2.1 LONG-TAILED LEARNING

In a long-tailed learning problem (Kang et al., 2020; Yang & Xu, 2020), in addition to satisfying Eq. (2), the label probability $q(y)$ is known to be uniform, specifically $q(y) = 1/M$ for any $y \in [M]$. In contrast, the label probabilities $p(y)$, where $y \in \mathcal{Y}$, may significantly deviate from a uniform distribution. The probability $p(y)$ can be easily estimated by $\widehat{p}(y) := n_p^{-1} \sum_{i=1}^{n_p} \mathbf{1}\{Y_i = y\}$. Consequently, the class probability ratio $w^*(y)$ can be estimated as $\widehat{w}(y) := \widehat{q}(y)/\widehat{p}(y) = 1/(M\widehat{p}(y))$. Furthermore, it is worth noting that in long-tailed learning, the classifier $\widehat{h}_q(x)$, derived from the estimator $\widehat{q}(y|x)$ in Eq. (3), aligns with the *logit adjustment classifier* proposed in Menon et al. (2021). A detailed proof of this equivalence is provided in Appendix B.

## 2.2 DOMAIN ADAPTATION: LABEL SHIFT

Label shift problems (Saerens et al., 2002) are characterized by Eq. (2) and an unknown label distribution $q(y)$. In addition to labeled samples $D_p$ from the source distribution $P$, unlabeled samples $D_q^u := (X_i)_{i=n_p+1}^{n_p+n_q}$ are drawn from the marginal probability density $q(x)$ of the target distribution $Q$. To estimate $w^* := (w^*(m))_{m \in [M]}$ based on the observations $D := (D_p, D_q^u)$, Wen et al. (2024) proposes a class probability matching-based estimator $\widehat{w}$. This estimator aligns two components: the class probability estimator in the source domain, $\widehat{p}(y) := n_p^{-1} \sum_{i=1}^{n_p} \mathbf{1}\{Y_i = y\}$, and the empirical mean of the weighted CCP estimation for the target domain samples, i.e.

$$\widehat{p}_q^w(y) := n_q^{-1} \sum_{X_i \in D_q^u} \frac{\widehat{p}(y|X_i)}{\sum_{m=1}^M w(m)\widehat{p}(m|X_i)}. \tag{4}$$

Clearly, the CCP estimation $\widehat{p}(y|x)$ influences the estimation of $w^*$ through $\widehat{p}_q^w(y)$ in Eq. (4). Therefore, having an accurate CCP estimator $\widehat{p}(y|x)$ in the source domain is essential for effective label shift adaptation.

## 2.3 TRANSFER LEARNING: LABEL BIAS

In transfer learning, we assume that the condition in Eq. (2) holds and that the label probability distribution $q$ over $\mathcal{Y}$ is uniform. We assume that only the pre-trained model $\widehat{p}(y|x)$ on $D_p$ is available, while labeled pre-trained data $D_p$ itself is not accessible. Additionally, we can observe a small number of auxiliary samples $D_s := (X_i^s, Y_i^s)_{i=1}^{n_s}$, drawn from an unknown data distribution $S$ defined on $\mathcal{X} \times \mathcal{Y}$, which is assumed to satisfy $s(x|y) = p(x|y)$. Due to the discrepancy between $p(y)$ and $q(y)$, directly applying the pre-trained model $\widehat{p}(y|x)$, fitted on $D_p$, for classification on $Q$ results in poor performance, a phenomenon referred to as *label bias*. Since $p(y) = \sum_m \mathbb{E}_{x \sim p(\cdot|m)} p(y|x)p(m)$ holds for any $y \in [M]$, Zhu et al. (2024) propose the estimator $\widehat{p}(y)$ as the stationary distribution of a Markov chain characterized by the transition matrix $\widehat{\mathcal{C}} = (\widehat{\mathcal{C}}_{kj}) \in \mathbb{R}^{M \times M}$, with entries given by

$$\widehat{\mathcal{C}}_{kj} := n_{s,j}^{-1} \sum_{i \in [n_s], Y_i^s = j} \widehat{p}(k|X_i^s), \tag{5}$$

where $n_{s,j}$ denotes the number of samples from the $j$-th class in $D_s$. Eq. (5) indicates that the CCP estimator $\widehat{p}(y|x)$ also influences the estimation of the label distribution $\widehat{p}(y)$. Given that $q(m) = \frac{1}{M}$ for all $m \in [M]$, the optimal weight $w^*(m)$ can be estimated as $\widehat{w}(y) := \widehat{q}(y)/\widehat{p}(y) = 1/(M\widehat{p}(y))$.

In summary, for these complex classification scenarios—namely long-tailed learning, domain adaptation, and transfer learning—the CCP estimation $\widehat{p}(y|x)$ not only directly influences the estimator $\widehat{q}(y|x)$ through Eq. (3), but also impacts the estimation of the class probability ratio $\widehat{w}(y)$ in the latter two scenarios. Consequently, the CCP estimation $\widehat{p}(y|x)$ based on the labeled data $D_p$ is a crucial and fundamental component in algorithms designed to tackle these complex classification tasks.

## 3 MINIMAX CONVERGENCE RATES OF KERNEL LOGISTIC REGRESSION FOR COMPLEX CLASSIFICATION SCENARIOS

As mentioned in Section 2, in many complex classification scenarios, our objective extends beyond merely training a classifier for label prediction; we are also interested in estimating the CCPs to

inform subsequent decision-making. *Kernel logistic regression* (*KLR*) (Zhu & Hastie, 2005) is particularly well-suited for this purpose, as it is specifically designed to model CCPs in classification problems. Specifically, let $f = (f_m)_{m \in [M]}$ be a score function, where each element $f_m : \mathcal{X} \to \mathbb{R}$ belongs to the *reproducing kernel Hilbert space* (*RKHS*) $H$, with the norm $\| \cdot \|_H$ induced by the Gaussian kernel function $k(x, x') := \exp(-\|x - x'\|_2^2 / \gamma^2)$ for $x, x' \in \mathbb{R}^d$ and a bandwidth parameter $\gamma$ (Steinwart & Christmann, 2008). The collection of score functions $f$ is denoted as

$$\mathcal{F} := \{ f := (f_m)_{m=1}^M : f_m \in H, m \in [M-1], f_M = 0 \}, \tag{6}$$

where we set the score function for the $M$-th class to zero to ensure uniqueness. The corresponding CCP estimator for $f \in \mathcal{F}$ is given by the softmax function of $f$, i.e.,

$$p_f(m|x) := \frac{\exp(f_m(x))}{\sum_{j=1}^M \exp(f_j(x))}, \qquad m \in [M], \tag{7}$$

which models the true CCP function $p(m|x)$. Note that if the CCP estimate of $p_f(y|x)$ is close to zero, the CE loss $-\log p_f(y|x)$ can become extremely large. To prevent an arbitrarily large CE loss, we truncate the CCP estimator $p_f(m|x)$. Specifically, given $t \in (0, 1/(2M))$, we define

$$p_f^t(m|x) := \begin{cases} t, & \text{if } p_f(m|x) < t, \\ p_f(m|x) - (p_f(m|x) - t) \cdot \dfrac{\sum_{j : p_f(j|x) < t}(t - p_f(j|x))}{\sum_{\ell : p_f(\ell|x) \geq t}(p_f(\ell|x) - t)}, & \text{if } p_f(m|x) \geq t. \end{cases} \tag{8}$$

The CCP function $p_f(m|x)$ is truncated at $t$ for values of $m$ less than $t$, while those $p_f(m|x)$ greater than $t$ are proportionally adjusted to ensure that $\sum_{m \in [M]} p_f^t(m|x) = 1$. It can be easily shown that for all $m \in [M]$ and $x \in \mathcal{X}$, the condition $p_f^t(m|x) \geq t$ holds, thereby bounding the CE loss. More precisely, we have $L_{\mathrm{CE}}(y, p_f^t(\cdot|x)) = -\log p_f^t(y|x) \leq -\log t$.

Accordingly, given the bandwidth parameter $\gamma > 0$, the truncation parameter $t \in [0, 1/(2M)]$, and a regularization parameter $\lambda > 0$, the KLR function $f_D$ is obtained through

$$f_D := \underset{f \in \mathcal{F}}{\arg\min} \, \mathcal{R}_{L_{\mathrm{CE}}, D}(p_f^t(y|x)) + \lambda \|f\|_H^2, \tag{9}$$

where $\|f\|_H^2 := \sum_{j=1}^{M-1} \|f_j\|_H^2$. The regularization term $\lambda \|f\|_H^2$ penalizes the RKHS norm of functions to prevent overfitting. The corresponding CCP estimator is expressed as

$$\widehat{p}(m|x) := p_{f_D}^t(m|x), \quad m \in [M]. \tag{10}$$

To analyze the proposed CCP estimator, the following restrictions on the distribution $P$ on $\mathcal{X} \times \mathcal{Y}$ need to be introduced.

**Assumption 3.1.** *We impose the following assumptions on the probability distribution $P$.*

(i) *[Hölder Smoothness] Assume that for any $x, x' \in \mathcal{X}$, there exists an $\alpha \in [0, 1]$ and a Hölder constant $c_\alpha \in (0, \infty)$ such that $|p(m|x') - p(m|x)| \leq c_\alpha \|x' - x\|_2^\alpha$ for all $m \in [M]$.*

(ii) *[Small Value Bound] Assume that for all $t \in (0, 1]$, there exist constants $\beta \geq 0$ and $c_\beta > 0$ such that $P_X(p(m|X) \leq t) \leq c_\beta t^\beta$ for all $m \in [M]$.*

The Hölder Smoothness Assumption *(i)* regarding the conditional probability function $p(m|x)$ is a common assumption in classification tasks (Chaudhuri & Dasgupta, 2014; Döring et al., 2017; Xue & Kpotufe, 2018; Khim et al., 2020). According to *(i)*, when $\alpha$ is small, the conditional probability function $p(m|x)$ exhibits sharper fluctuations, making accurate estimation more challenging and consequently leading to slower convergence rates. The Small Value Bound (SVB), i.e. Assumption 3.1 *(ii)*, adapted from Bos & Schmidt-Hieber (2022), quantifies the size of the set where the conditional probabilities $p(m|x)$ are small. As the conditional probability $p(m|x)$ approaches zero, the value of $-\log p(m|x)$ increases towards infinity at an accelerating rate. Thus, accurately estimating small conditional probabilities significantly impacts the value of the CE loss. Therefore, the classification problem demonstrates faster convergence rates with respect to the CE loss when the probability of regions with small conditional probabilities is low (i.e., when $\beta$ is large).

### 3.1 LONG-TAILED LEARNING

The following theorem presents the convergence rate of the CCP estimator $\widehat{q}(y|x)$ defined in Eq. (3).

**Theorem 3.2** (Upper Bound). *Under the setting in Section 2.1, let Assumption 3.1 hold. Moreover, let $\widehat{q}(y|x)$ be the CCP estimator as in Eq. (3) with $\widehat{p}(y|x)$ being KLR's CCP estimator as in Eq. (10). Then by choosing $\lambda \asymp n_p^{-1}$, $\gamma \asymp n_p^{-1/((1+\beta \wedge 1)\alpha + d)}$, and $t \asymp n_p^{-\zeta}$ with $\zeta \geq 1$, there exists an $N \in \mathbb{N}$ such that for any $n_p \geq N$ and for any $\xi > 0$, with probability $P^{n_p}$ at least $1 - 2/n_p$, there holds*

$$\mathcal{R}_{L_{\mathrm{CE}},Q}(\widehat{q}(y|x)) - \mathcal{R}_{L_{\mathrm{CE}},Q}^* \lesssim n_p^{-\frac{(1+\beta \wedge 1)\alpha}{(1+\beta \wedge 1)\alpha + d} + \xi}. \tag{11}$$

Under the same assumptions—namely, that the distributions $P$ and $Q$ adhere to the long-tailed setting and that $P$ satisfies Assumption 3.1—the following theorem establishes a lower bound.

**Theorem 3.3** (Lower Bound). *Under the setting in Section 2.1, let Assumption 3.1 hold. Moreover, let $\mathcal{A}$ be a learning algorithm that accepts data $D_p$ and outputs a CCP estimator. Then, there exists a constant $c \in (0,1)$ such that with probability $P^{n_p}$ at least $c$, there holds*

$$\inf_{\mathcal{A}} \sup_{P,Q} \mathcal{R}_{L_{\mathrm{CE}},Q}(\mathcal{A}(D_p)) - \mathcal{R}_{L_{\mathrm{CE}},Q}^* \gtrsim n_p^{-\frac{(1+\beta \wedge 1)\alpha}{(1+\beta \wedge 1)\alpha + d}}. \tag{12}$$

Up to the polynomial term $n^\xi$ with an arbitrarily small $\xi$, the lower bound in Eq. (12) aligns with the upper bound in Eq. (11). Consequently, the CCP estimator $\widehat{q}(y|x)$ in Eq. (3) is minimax optimal.

### 3.2 LABEL SHIFT ADAPTATION

For the label shift problem described in Section 2.2, the following theorem presents the convergence rate of $\widehat{q}(y|x)$ for the target domain.

**Theorem 3.4** (Upper Bound). *Under the setting in Section 2.2, let Assumption 3.1 hold. Moreover, let $\widehat{q}(y|x)$ be the CCP estimator as in Eq. (3) with $\widehat{p}(y|x)$ being KLR's CCP estimator in Eq. (10). Then, by choosing $\lambda \asymp n_p^{-1}$, $\gamma \asymp n_p^{-1/((1+\beta \wedge 1)\alpha + d)}$, and $t \asymp n_p^{-\zeta}$ with $\zeta \geq 1$, there exists an $N_1 \in \mathbb{N}$ such that for any $n_p \wedge n_q \geq N_1$ and for any $\xi > 0$, there holds*

$$\mathcal{R}_{L_{\mathrm{CE}},Q}(\widehat{q}(y|x)) - \mathcal{R}_{L_{\mathrm{CE}},Q}^* \lesssim n_p^{-\frac{(1+\beta \wedge 1)\alpha}{(1+\beta \wedge 1)\alpha + d} + \xi} + \log n_q / n_q \tag{13}$$

*with probability $P^{n_p} \otimes Q_X^{n_q}$ at least $1 - 1/n_p - 1/n_q$.*

Eq. (13) indicates that the convergence rate of the CCP estimator $\widehat{q}(y|x)$ is influenced primarily by the larger term in $n_p^{-\theta}$ and $n_q^{-1}$, where $\theta := (1 + \beta \wedge 1)\alpha/((1 + \beta \wedge 1)\alpha + d)$. The terms $n_p^\xi$ and $\log n_q$ are not critical for the rate of decay. In practice, we typically maintain a fixed sample size $n_p$ for the source domain while the sample size $n_q$ for the target domain gradually increases. As $n_q$ rises from zero to approximately $n_p^\theta$, $n_q^{-1}$ decreases from infinity to $n_p^{-\theta}$, resulting in a faster convergence rate in Eq. (13) that ultimately aligns with the order of $n_p^{-\theta}$. If $n_q$ continues to grow, the order of the convergence rate in Eq. (13) remains unchanged. This illustrates that given the labeled source domain data, a certain number of unlabeled target domain samples are sufficient to achieve efficient performance.

**Theorem 3.5** (Lower Bound). *Under the setting in Section 2.2, let Assumption 3.1 hold. In addition, let $\mathcal{A}$ be a learning algorithm that accepts data $D := (D_p, D_q^u)$ and outputs a CCP predictor. Then, there exist a constant $c \in (0,1)$ such that with probability $P^{n_p} \otimes Q_X^{n_q}$ at least $c$, there holds*

$$\inf_{\mathcal{A}} \sup_{P,Q} \mathcal{R}_{L_{\mathrm{CE}},Q}(\mathcal{A}(D)) - \mathcal{R}_{L_{\mathrm{CE}},Q}^* \gtrsim n_p^{-\frac{(1+\beta \wedge 1)\alpha}{(1+\beta \wedge 1)\alpha + d}} + n_q^{-1}. \tag{14}$$

Theorems 3.4 and 3.5 demonstrate that the CCP estimator $\widehat{q}(y|x)$ in Eq. (3) attains minimax optimal rates, up to an arbitrarily small polynomial term, for label shift adaptation.

## 3.3 MITIGATING LABEL BIAS IN TRANSFER LEARNING

For transfer learning with label bias, as described in Section 2.3, the following theorem presents the convergence rate for $\widehat{q}(y|x)$.

**Theorem 3.6** (Upper Bound). *Under the setting in Section 2.3, let Assumption 3.1 hold. Moreover, let $\widehat{q}(y|x)$ be the CCP estimator Eq. (3) in the target domain with $\widehat{p}(y|x)$ being KLR's CCP estimator in Eq. (10). Then, by choosing $\lambda \asymp n_p^{-1}$, $\gamma \asymp n_p^{-1/((1+\beta\wedge 1)\alpha+d)}$, and $t \asymp n_p^{-\zeta}$ with $\zeta \geq 1$, there exists an $N_2 \in \mathbb{N}$ such that for any $n_p \wedge n_s \geq N_2$ and for any $\xi > 0$, there holds*

$$\mathcal{R}_{L_{\mathrm{CE}},Q}(\widehat{q}(y|x)) - \mathcal{R}^*_{L_{\mathrm{CE}},Q} \lesssim n_p^{-\frac{(1+\beta\wedge 1)\alpha}{(1+\beta\wedge 1)\alpha+d}+\xi} + \log n_s/n_s \tag{15}$$

*with probability $P^{n_p} \otimes P_s^{n_s}$ at least $1 - 1/n_p - 1/n_s$.*

In practice, the sample size of the pre-trained dataset is typically much larger than that of the auxiliary dataset, i.e., $n_p \gg n_s$. If access to the pre-trained model $\widehat{p}(y|x)$ is not available, the transfer learning problem discussed in Section 2.3 becomes the long-tailed learning problem addressed in Section 2.1. In this case, Theorem 3.2 provides the convergence rate $n_s^{-(1+\beta\wedge 1)\alpha/(1+(\beta\wedge 1)\alpha+d)+\xi}$, which is significantly slower than that in Eq. (15). This highlights that mitigating label bias in the pre-trained model is more advantageous than training a new model from scratch, thereby illustrating the effectiveness of transfer learning.

**Theorem 3.7** (Lower Bound). *Under the setting in Section 2.3, let Assumption 3.1 hold. Moreover, let $\mathcal{A}$ be a learning algorithm that accepts data $D := (D_p, D_s)$ and outputs a CCP predictor. Then, there exists a constant $c \in (0, 1)$ such that with probability $P^{n_p} \otimes S^{n_s}$ at least $c$, there holds*

$$\inf_{\mathcal{A}} \sup_{P,S,Q} \mathcal{R}_{L_{\mathrm{CE}},Q}(\mathcal{A}(D)) - \mathcal{R}^*_{L_{\mathrm{CE}},Q} \gtrsim (n_p \vee n_s)^{-\frac{(1+\beta\wedge 1)\alpha}{(1+\beta\wedge 1)\alpha+d}} + n_s^{-1}. \tag{16}$$

Since we typically have $n_p \gg n_s$ in practice, the upper bound in Eq. (15) aligns with the lower bound in Eq. (16), up to the polynomial term $n_p^\xi$ with an arbitrarily small $\xi$. Therefore, the predictor $\widehat{q}(y|x)$ in Eq. (3) is minimax optimal with respect to the CE loss for addressing label bias in transfer learning.

In summary, for the three scenarios discussed, the convergence rates with respect to the CE loss in Theorems 3.2, 3.4, and 3.6 are minimax optimal. By incorporating the calibration inequality in Eq. (1), we can derive convergence rates for the classification loss as well. Elementary calculations indicate that when the exponent of the SVB $\beta \geq 1$ in Assumption 3.1 *(ii)*, the convergence rates for the classification loss are also minimax optimal. Further details can be found in Appendix E.

## 4 ERROR ANALYSIS

In this section, we present the error analysis of the excess CE risk $\mathcal{R}_{L_{\mathrm{CE}},Q}(\widehat{q}(y|x)) - \mathcal{R}^*_{L_{\mathrm{CE}},Q}$ of the CCP estimation $\widehat{q}(y|x)$ for the complex classification scenarios in Section 3.

### 4.1 REDUCING ERROR IN COMPLEX CLASSIFICATION SCENARIOS TO ERROR IN STANDARD CLASSIFICATION SCENARIO

The following inequalities indicate that the excess CE risk $\mathcal{R}_{L_{\mathrm{CE}},Q}(\widehat{q}(y|x)) - \mathcal{R}^*_{L_{\mathrm{CE}},Q}$ of the CCP estimation $\widehat{q}(y|x)$ for the complex classification scenarios in Section 3 can be reduced to the excess CE risk $\mathcal{R}_{L_{\mathrm{CE}},P}(\widehat{p}(y|x)) - \mathcal{R}^*_{L_{\mathrm{CE}},P}$ of the CCP estimation $\widehat{p}(y|x)$ for the standard classification.

*(a)* Under the long-tailed learning setting in Section 2.1, we have

$$\mathcal{R}_{L_{\mathrm{CE}},Q}(\widehat{q}(y|x)) - \mathcal{R}^*_{L_{\mathrm{CE}},Q} \lesssim \mathcal{R}_{L_{\mathrm{CE}},P}(\widehat{p}(y|x)) - \mathcal{R}^*_{L_{\mathrm{CE}},P} + \log n_p/n_p. \tag{17}$$

*(b)* Under the label shift adaptation setting in Section 2.2, we have

$$\mathcal{R}_{L_{\mathrm{CE}},Q}(\widehat{q}(y|x)) - \mathcal{R}^*_{L_{\mathrm{CE}},Q} \lesssim \mathcal{R}_{L_{\mathrm{CE}},P}(\widehat{p}(y|x)) - \mathcal{R}^*_{L_{\mathrm{CE}},P} + \log n_q/n_q + \log n_p/n_p. \tag{18}$$

*(c)* Under the transfer learning setting in Section 2.3, we have

$$\mathcal{R}_{L_{\mathrm{CE}},Q}(\widehat{q}(y|x)) - \mathcal{R}^*_{L_{\mathrm{CE}},Q} \lesssim \mathcal{R}_{L_{\mathrm{CE}},P}(\widehat{p}(y|x)) - \mathcal{R}^*_{L_{\mathrm{CE}},P} + \log n_s/n_s. \tag{19}$$

All proofs of the above inequalities Eq. (17), Eq. (18), and Eq. (19) can be found in Appendix D.2.

## 4.2 EXCESS CE RISK OF KLR IN STANDARD CLASSIFICATION SCENARIOS

In this section, we analyze the excess CE risk $\mathcal{R}_{L_{\mathrm{CE}},P}(\widehat{p}(y|x)) - \mathcal{R}^*_{L_{\mathrm{CE}},P}$ associated with the KLR estimator $\widehat{p}(y|x)$ for standard classification. For simplicity, we omit the subscript CE in $L_{\mathrm{CE}}$ w.l.o.g. To address the analytical challenges posed by the unbounded nature of the CE loss, we decompose the CE loss into an upper part $L^u$ and a lower part $L^l$, based on whether the true CCP exceeds or falls below the truncation parameter $t$ as defined in Eq. (8). Specifically, for any CCP function $p_f : \mathcal{X} \to \Delta^{M-1}$, we define these two components of its CE loss as follows:

$$(L^u \circ p_f)(x,y) := L^u(y, p_f(\cdot|x)) := \mathbf{1}\{p(y|x) \geq t\}(-\log p_f(y|x)), \qquad (20)$$

$$(L^l \circ p_f)(x,y) := L^l(y, p_f(\cdot|x)) := \mathbf{1}\{p(y|x) < t\}(-\log p_f(y|x)). \qquad (21)$$

According to Eq. (20), the CE loss of the true CCP function $p(y|x)$ on the upper part $L^u \circ p$ is always bounded. Therefore, it suffices to focus on the loss in the lower part $L^l \circ p$, which is unbounded. For notational simplicity, we denote the excess CE loss of the truncated CCP estimator $p_f^t(\cdot|x)$ defined in Eq. (8) as $h_{p_f^t} := L \circ p_f^t - L \circ p$. Similarly, we define the excess CE loss for the upper part and the lower part as $h^u_{p_f^t} := L^u \circ p_f^t - L^u \circ p$ and $h^l_{p_f^t} := L^l \circ p_f^t - L^l \circ p$, respectively.

Using the CE loss decomposition in Eq. (20) and Eq. (21), we can perform an error decomposition of the excess CE risk associated with KLR's CCP estimator $\widehat{p}(y|x)$ as defined in Eq. (10). To this end, let $f_0$ be an approximating function within the space $\mathcal{F}$ specified in Eq. (6). Thus, we have

$$\lambda\|f_D\|_H^2 + \mathcal{R}_{L,P}(\widehat{p}(y|x)) - \mathcal{R}^*_{L,P} = \lambda\|f_D\|_H^2 + \mathcal{R}_{L,P}(p^t_{f_D}(y|x)) - \mathcal{R}^*_{L,P} \qquad \text{(By Eq. (10))}$$

$$= \lambda\|f_D\|_H^2 + \mathbb{E}_P h_{p^t_{f_D}} = \lambda\|f_D\|_H^2 + \mathbb{E}_D h_{p^t_{f_D}} - \mathbb{E}_D h_{p^t_{f_D}} + \mathbb{E}_P h_{p^t_{f_D}} \qquad \text{(By definition of } h_{p_f^t})$$

$$\leq \lambda\|f_0\|_H^2 + \mathbb{E}_D h_{p^t_{f_0}} - \mathbb{E}_D h_{p^t_{f_D}} + \mathbb{E}_P h_{p^t_{f_D}} \qquad \text{(By definition of } f_D \text{ in Eq. (9))}$$

$$= \lambda\|f_0\|_H^2 + \mathbb{E}_P h_{p^t_{f_0}} + \mathbb{E}_D h_{p^t_{f_0}} - \mathbb{E}_P h_{p^t_{f_0}} + \mathbb{E}_P h_{p^t_{f_D}} - \mathbb{E}_D h_{p^t_{f_D}}$$

$$= \lambda\|f_0\|_H^2 + \mathbb{E}_P h_{p^t_{f_0}} + (\mathbb{E}_D h^l_{p^t_{f_0}} - \mathbb{E}_P h^l_{p^t_{f_0}} + \mathbb{E}_P h^l_{p^t_{f_D}} - \mathbb{E}_D h^l_{p^t_{f_D}})$$

$$+ (\mathbb{E}_D h^u_{p^t_{f_0}} - \mathbb{E}_P h^u_{p^t_{f_0}} + \mathbb{E}_P h^u_{p^t_{f_D}} - \mathbb{E}_D h^u_{p^t_{f_D}}) \quad \text{(Loss decomposition } h_{p_f^t} = h^u_{p_f^t} + h^l_{p_f^t}) \quad (22)$$

$$= \lambda\|f_0\|_H^2 + \mathbb{E}_P h_{p^t_{f_0}} + (\mathbb{E}_D L^l \circ p^t_{f_0} - \mathbb{E}_P L^l \circ p^t_{f_0}) + (\mathbb{E}_P L^l \circ p^t_{f_D} - \mathbb{E}_D L^l \circ p^t_{f_D})$$

$$+ (\mathbb{E}_D h^u_{p^t_{f_0}} - \mathbb{E}_P h^u_{p^t_{f_0}}) + (\mathbb{E}_P h^u_{p^t_{f_D}} - \mathbb{E}_D h^u_{p^t_{f_D}}), \quad \text{(Simplification by definition of } h^l_{p_f^t}) \quad (23)$$

where the first two summands represent the *approximation error*, while the last four summands correspond to the *sample error* terms. Note that although $L^l \circ p$ is unbounded in Eq. (22), which causes the excess losses on the lower part—$h^l_{p^t_{f_0}}$ and $h^l_{p^t_{f_D}}$—to be unbounded as well, the unbounded component $L^l \circ p$ appears in both $h^l_{p^t_{f_0}}$ and $h^l_{p^t_{f_D}}$ and subsequently cancels out. As a result, only the bounded parts $L^l \circ p^t_{f_D}$ and $L^l \circ p^t_{f_0}$, which lie within $[\log t, -\log t]$, remain in the final Eq. (23). This ensures that all four sample error terms are bounded, highlighting the ingenuity of our analysis.

### 4.2.1 BOUNDING THE SAMPLE ERRORS

The following theorem presents our new oracle inequality of KLR for CCP estimation.

**Theorem 4.1** (Oracle inequality). *Let $f_D$ be as in Eq. (9) and the truncated KLR's CCP estimator $\widehat{p}(y|x)$ as in Eq. (10). Furthermore, let $\mathcal{F}$ be as in Eq. (6) and the truncation threshold $t < 1/(2M)$. Then for any $f_0 \in \mathcal{F}$, $\xi \in (0, 1/2)$, and $\zeta > 0$, with probability at least $1 - 4e^{-\zeta}$, there holds*

$$\lambda\|f_D\|_H^2 + \mathcal{R}_{L_{\mathrm{CE}},P}(\widehat{p}(y|x)) - \mathcal{R}^*_{L_{\mathrm{CE}},P}$$
$$\lesssim (\lambda\|f_0\|_H^2 + \mathcal{R}_{L_{\mathrm{CE}},P}(p^t_{f_0}(y|x)) - \mathcal{R}^*_{L_{\mathrm{CE}},P}) + (-\log t) \cdot (t + \lambda^{-\xi}\gamma^{-d}n^{-1} + \zeta/n). \qquad (24)$$

Theorem 4.1 establishes that the excess CE risk of KLR's CCP estimator $\widehat{p}(y|x)$ is bounded by the sum of the approximation error and the sample error, which correspond to the two terms on the right-hand side of Eq. (24). Specifically, the second term on the right-hand side of Eq. (24) provides an upper bound for the four sample error components in the error decomposition presented in Eq. (23). Further discussions about the oracle inequality can be found in Appendix C.

### 4.2.2 BOUNDING THE APPROXIMATION ERROR

In this section, we establish the upper bound of the approximation error, i.e. the first two summands in Eq. (23). To this end, let $p(m|x)$ be the CCP funtion and define the score function

$$f^* := (f^*(m))_{m=1}^M \qquad \text{with} \qquad f_m^*(x) := \log(p(m|x)/p(M|x)), \quad m \in [M]. \qquad (25)$$

Since $\mathcal{R}_{L_{\mathrm{CE}},P}^* = \mathcal{R}_{L_{\mathrm{CE}},P}(p(y|x))$ and $p(y|x) = p_{f^*}(y|x)$ in Eq. (7), $f^*$ is the score function that achieves the minimal CE risk. Therefore, in order to bound the approximation error, we need to construct a function $f_0 \in \mathcal{F}$ in Eq. (6) to approximate the Bayes score function $f^*$. However, since $p(m|x)$ can be close to zero, $f^*$ in Eq. (25) can be unbounded. To address this issue, we consider a truncated version of $f^*$, namely, for a given threshold $\tau \in [0, 1/(2M))$, define

$$f^{*\tau} := (f_m^{*\tau})_{m\in[M]} \qquad \text{with} \qquad f_m^{*\tau}(x) := \log\big(p^\tau(m|x)/p^\tau(M|x)\big), \quad m \in [M], \qquad (26)$$

where the truncated CCP function $p^\tau(m|x)$ is defined as in Eq. (8) with $p_f$ replaced by $p$.

Now, we construct a function $f_0 \in \mathcal{F}$ in Eq. (6) to approximate the bounded function $f^{*\tau}$. To this end, let $K(x) := \big(2/(\gamma^2\pi)\big)^{d/2}\exp\big(-2\|x\|_2^2/\gamma^2\big)$ be the convolution operator induced by the Gaussian kernel function $k$ and define the score function

$$\widetilde{f}^\tau := (\widetilde{f}_m^\tau)_{m\in[M]} \qquad \text{with} \qquad \widetilde{f}_m^\tau(x) := (K * f_m^{*\tau})(x), \quad m \in [M], \qquad (27)$$

where $K * f_m^{*\tau}$ is the convolution of $K$ and $f_m^{*\tau}$ in Eq. (26).

The following proposition presents the approximation error bound when the approximating function $f_0$ is chosen as $\widetilde{f}^\tau$ in Eq. (27) with properly chosen $\tau$.

**Proposition 4.2.** *Let Assumption 3.1 hold. Furthermore, let $\gamma$ be the bandwidth of the Gaussian kernel and let $t \in [0, 1/(2M)]$ be the truncation parameter. Then there exists an $f_0 := \widetilde{f}^\tau \in \mathcal{F}$ in Eq. (27) with $\tau := \gamma^\alpha$ such that for any $t \le \tau$, there holds*

$$\lambda\|f_0\|_H^2 + \mathcal{R}_{L_{\mathrm{CE}},P}(p_{f_0}^t(y|x)) - \mathcal{R}_{L_{\mathrm{CE}},P}^* \lesssim \lambda\gamma^{-d}\log^2(\gamma^{-\alpha}) + \log(\gamma^{-\alpha}) \cdot \gamma^{\alpha(1+\beta\wedge 1)}. \qquad (28)$$

Proposition 4.2 provides the approximation error bound for $f_0 := \widetilde{f}^\tau$ as defined in Eq. (27), relative to $f^*$ in Eq. (25). This is achieved by bounding the approximation error of $f_0$ with respect to $f^{*\tau}$ in Eq. (26), along with the approximation error of $f^{*\tau}$ to $f^*$. The approximation error bound in Eq. (28) can be upper bounded by $\log^2(\gamma^{-\alpha})(\lambda\gamma^{-d} + \gamma^{\alpha(1+\beta\wedge 1)}) \to 0$ as the bandwidth $\gamma \to 0$ and $\lambda\gamma^{-d}\log^2(\gamma^{-\alpha}) \to 0$. This illustrates that $f_0$ can effectively approximate the unbounded score function $f^*$ with respect to the CE loss, aided by the intermediate function $f^{*\tau}$. Importantly, the appropriately chosen value of $\tau := \gamma^\alpha$ not only facilitates the closeness of $\widetilde{f}^\tau$ to the unbounded target function $f^*$ through $f^{*\tau}$ but also ensures that $\widetilde{f}^\tau$ remains above a specified positive threshold, thus guaranteeing that the CE loss of $\widetilde{f}^\tau$ is bounded. This effectively resolves the issue of unboundedness in the approximation error analysis.

## 5 EXPERIMENTS

In this section, we evaluate the CCP-based methods on three classification datasets: the `Dionis` and `Satimage` datasets from the OpenML Science Platform (Vanschoren et al., 2014), and the `Gas Sensor` dataset from the UCI ML Repository (Kelly et al., 2007). A detailed description of the data generation process and the hyperparameter grids for KLR is provided in Appendix F.

**Validation of Effectiveness.** The CCP-based estimator $\widehat{q}(y|x)$ in Eq. (3), tailored for complex classification tasks on distribution $Q$, is referred to as the *CCP* method. Similarly, the estimator $\widehat{p}(y|x)$ in Eq. (10), designed for standard classification on data from $P$, is referred to as the *baseline* method. To validate the effectiveness of the CCP-based method in complex classification tasks, we compare its performance against that of the baseline method.

Table 1 summarizes the label prediction accuracy of the CCP-based estimator $\widehat{q}(y|x)$ in Eq. (3) and the baseline estimator $\widehat{p}(y|x)$ in Eq. (10) on the test data $D_t^u$ drawn from distribution $Q$, evaluated across three complex classification scenarios. The consistently higher accuracy of the CCP-based method compared to the baseline demonstrates the effectiveness of CCP-based algorithms, particularly those leveraging KLR, in tackling complex learning challenges using real-world datasets.

Table 1: Accuracy on real-world datasets across different complex classification scenarios.

| Dataset | Method | Long-tailed Learning | Domain Adaptation | Transfer Learning |
|---------|--------|---------------------|-------------------|-------------------|
| Dionis | Baseline | 80.71 ± 0.87 | 77.69 ± 1.58 | 80.72 ± 0.88 |
|        | CCP | **83.67 ± 1.10** | **82.73 ± 1.40** | **84.22 ± 0.99** |
| Gas Sensor | Baseline | 85.57 ± 5.97 | 96.14 ± 0.89 | 85.97 ± 5.57 |
|          | CCP | **90.49 ± 4.47** | **96.52 ± 1.30** | **90.27 ± 4.78** |
| Satimage | Baseline | 80.51 ± 3.70 | 89.80 ± 3.90 | 80.51 ± 3.70 |
|         | CCP | **84.56 ± 1.32** | **96.46 ± 2.42** | **84.47 ± 1.98** |

For each dataset and each method, we highlight the best performance in **bold**.

**Validation of Theoretical Results.** As an illustration, we conduct experiments on the `Dionis` dataset, chosen for its large number of classes, which makes it particularly suitable for this analysis. To validate the theoretical results presented in Section 3, we investigate the effect of varying sample sizes ($n_p$, $n_q$, and $n_s$) on the performance of CCP-based methods.

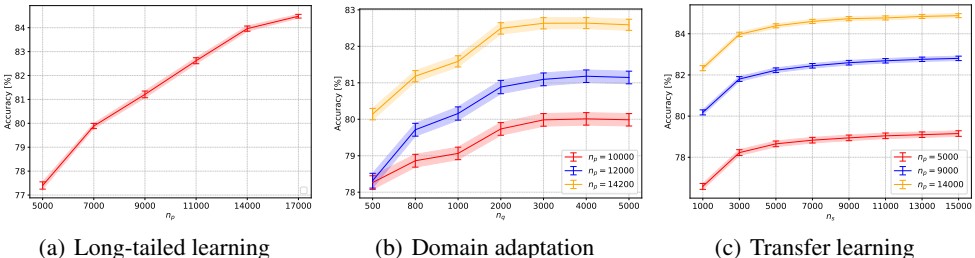

(a) Long-tailed learning      (b) Domain adaptation      (c) Transfer learning

Figure 1: The impact of sample sizes on accuracy in complex classification scenarios.

*(a)* In the context of long-tailed learning, Figure 1(a) illustrates that the accuracy on test data from $Q$ improves as $n_p$ increases. This observation aligns with Theorem 3.2, which asserts that the excess CE risk on $Q$ diminishes with larger $n_p$. Furthermore, according to Eq. (1), this reduction in excess CE risk indicates that the classification error converges toward its theoretical minimum. Therefore, Figure 1(a) provides empirical support for the theoretical findings in Theorem 3.2.

*(b)* In the case of domain adaptation, Figure 1(b) demonstrates that, for a fixed $n_p$, accuracy improves as $n_q$ increases from 500 to 3000. However, as $n_q$ grows beyond 3000 to 5000, the performance stabilizes. This pattern suggests that, given the labeled source domain data $D_p$, a sufficient number of unlabeled target domain samples $n_q$ is essential to achieve optimal performance. Additionally, Figure 1(b) highlights that when $n_q$ is sufficiently large, increasing $n_p$ leads to higher accuracy. These findings are consistent with the results presented in Theorem 3.4 and the associated discussion.

*(c)* In the context of transfer learning, Figure 1(c) shows that the impact of $n_p$ and $n_s$ on accuracy mirrors the trends observed for $n_p$ and $n_q$ in label shift adaptation (Figure 1(b)). This similarity indicates that the observations in Figure 1(c) serve as empirical validation of the theoretical results presented in Theorem 3.6, specifically regarding the roles of $n_p$ and $n_s$.

## 6    CONCLUSION AND FUTURE WORK

In this paper, we investigate the theoretical properties of kernel logistic regression (KLR) in addressing complex classification scenarios, including long-tailed learning, domain adaptation, and transfer learning. By emphasizing KLR's strengths in conditional class probability (CCP) estimation, we demonstrate that algorithms based on KLR can achieve minimax optimal convergence rates for cross-entropy (CE) loss under mild conditions. These results provide robust theoretical support for the empirical success of KLR-based methods, showcasing their capacity to operate with optimal efficiency in challenging scenarios. Our findings enhance the understanding of the theoretical foundations of CCP-based algorithms and validate their effectiveness in solving real-world, complex classification tasks. Looking ahead, recognizing that deep neural networks (DNNs) are highly effective CCP estimators, we plan to extend our error decomposition framework to include DNNs. This extension will deepen our theoretical understanding of DNNs in complex classification scenarios. Furthermore, we aim to explore even more intricate learning environments, such as those characterized by dynamic conditions or evolving data distributions.

ACKNOWLEDGMENTS

Annika Betken gratefully acknowledges financial support from the Dutch Research Council (NWO) through VENI grant 212.164. The authors acknowledge Tao Huang's involvement in some discussions.

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

## A  SOME NOTATIONS

For $1 \leq p < \infty$, the $L_p$-norm of $x = (x_1, \ldots, x_d)$ is defined as $\|x\|_p := (|x_1|^p + \ldots + |x_d|^p)^{1/p}$, while the $L_\infty$-norm is defined as $\|x\|_\infty := \max_{i=1,\ldots,d} |x_i|$. For any $x \in \mathbb{R}^d$ and $r > 0$, we explicitly denote $B_r(x) := B(x, r) := \{x' \in \mathbb{R}^d : \|x' - x\|_2 \leq r\}$ as the closed ball centered at $x$ with radius $r$. In addition, denote $\mu(A)$ as the Lebesgue measure of the set $A \subset \mathbb{R}^d$.

## B  DISCUSSION ON THE METHODS FOR LONG-TAILED LEARNING

In this section, we demonstrate that the classifier $\widehat{h}_q(x)$ induced by our CCP-based estimator $\widehat{q}(y|x)$ from Eq. (3) aligns with the *logit adjustment classifier* introduced in Menon et al. (2021). Specifically, Eq. (7) and Eq. (8) in Menon et al. (2021) show that the optimal classifier $h_q^*$ under the distribution $Q$ can be expressed as

$$h_q^*(x) = \arg\max_{m \in [M]} \frac{p(x|m)}{M} = \arg\max_{m \in [M]} p(x|m) = \arg\max_{m \in [M]} \frac{p(m|x)p(x)}{p(m)}$$

$$= \arg\max_{m \in [M]} \frac{p(m|x)}{p(m)} = \arg\max_{m \in [M]} \left[\log p(m|x) - \log p(m)\right].$$

Since the posterior probability $p(m|x)$ can be estimated as

$$\widehat{p}(m|x) := \frac{\exp(f_m(x))}{\sum_{j \in [M]} \exp(f_j(x))},$$

it follows that $\log \widehat{p}(m|x) \propto f_m(x)$. Using this insight, Menon et al. (2021) proposed the *logit adjustment classifier* defined as

$$\widehat{h}_{\mathrm{LA}}(x) = \arg\max_{m \in [M]} \left[\log \widehat{p}(m|x) - \log \widehat{p}(m)\right] = \arg\max_{m \in [M]} \left[f_m(x) - \log \widehat{p}(m)\right]. \tag{29}$$

This formulation adjusts the *logits* $f_m(x)$ by subtracting the logarithm of the estimated class prior $\widehat{p}(m)$, effectively mitigating the influence of class imbalances on the classification decision.

Leveraging Eq. (3) with $\widehat{w}(y) = 1/(M\widehat{p}(y))$, our CCP-based classifier can be expressed as

$$\widehat{h}_q(x) = \arg\max_{m \in [M]} \widehat{q}(m|x) = \arg\max_{m \in [M]} \frac{\widehat{p}(m|x)/\widehat{p}(m)}{\sum_{j \in [M]} \widehat{p}(j|x)/\widehat{p}(j)}$$

$$= \arg\max_{m \in [M]} \frac{\widehat{p}(m|x)}{\widehat{p}(m)} = \arg\max_{m \in [M]} \left[\log \widehat{p}(m|x) - \log \widehat{p}(m)\right]. \tag{30}$$

Therefore, the classifier $\widehat{h}_q(x)$ in Eq. (30), derived from the CCP-based estimator $\widehat{q}(y|x)$ in Eq. (3), is consistent with the logit adjustment classifier $\widehat{h}_{\mathrm{LA}}$ in Eq. (29) introduced in Menon et al. (2021).

## C  DISCUSSION ON THE ORACLE INEQUALITY

Zhang et al. (2024) also established a new oracle inequality applicable to the CE loss; however, their oracle inequality is limited to binary classification. In this section, we demonstrate that directly extending their proof technique to the multi-class scenario causes their variance bound in Eq. (2.5) of Zhang et al. (2024) to increase to a polynomial order, as opposed to a logarithmic order in the binary case, leading to suboptimal convergence rates. We will illustrate this with examples in the following.

To be specific, in order to cope with the unbounded CE loss, Zhang et al. (2024) construct a bounded function $\phi(x,y)$ such that the expectation of $\phi$ is equal to the Bayes CE risk, i.e. $\mathbb{E}\phi(X,Y) = \mathcal{R}^*_{L_{\mathrm{CE}},P} = \mathbb{E}(-\log p(Y|X))$. When analyzing the excess risk $\mathcal{R}_{L_{\mathrm{CE}},P}(\widehat{p}(y|x)) - \mathcal{R}^*_{L_{\mathrm{CE}},P}$, they investigate the expectation of a bounded term, that is, $\mathbb{E}\big(L_{\mathrm{CE}}(\widehat{p}(Y|X)) - \phi(X,Y)\big)$, instead of the expectation of the original unbounded excess loss, i.e., $\mathbb{E}\big(L_{\mathrm{CE}}(\widehat{p}(Y|X)) - L_{\mathrm{CE}}(p(Y|X))\big)$. Given a small $\delta_1 > 0$, for the binary classification with $\mathcal{Y} = \{1,2\}$, the specific form of bounded function $\phi(x,y)$ is chosen as

$$\phi(x,y) := \begin{cases} -\log(p(y|x)), & \text{if } p(1|x) \in [\delta_1, 1-\delta_1], \\ 0, & \text{if } p(1|x) \in \{0,1\}, \\ \sum_{m\in[2]} -p(m|x)\log p(m|x), & \text{if } p(1|x) \in (0,\delta_1) \cup (1-\delta_1,1). \end{cases} \tag{31}$$

Note that in Eq. (31), when $p(1|x) \in (0,\delta_1) \cup (1-\delta_1,1)$, that is, the CCP is close to 0 or 1, $\phi(x,y)$ is defined as the inner risk of the true CCP function, i.e. $\phi(x,y) = \mathbb{E}_{Y|X=x}L_{\mathrm{CE}}(p(Y|X))$, which is bounded by the fixed quantity $-\log \delta_1$. Then they prove that $\phi$ can satisfy the variance bound by a careful case-by-case analysis, especially for the special case that $p(1|x) \in (0,\delta_1) \cup (1-\delta_1,1)$. They show that when setting $\widehat{p}(y|x) \in [\delta_0, 1-\delta_0]$ and choosing $\delta_1 := \delta_0/(10\log(1/\delta_0))$ with $\delta_0 \in (0,1/3)$, the variance bound

$$\mathbb{E}(L_{\mathrm{CE}}(\widehat{p}(y|X)) - \phi(X,Y))^2 \le V\mathbb{E}(L_{\mathrm{CE}}(\widehat{p}(y|X)) - \phi(X,Y)) \tag{32}$$

holds with $V := 125000(-\log\delta_0)^2$ for any distribution $P$. By choosing a polynomial order for $\delta_0$, i.e. $\delta_0 \asymp n^{-\theta}$ with $\theta \ge \alpha/(d+\alpha))$, then the variance bound holds with $V \asymp (\log n)^2$ as a logarithmic order. In addition, there holds $L_{\mathrm{CE}}(\widehat{p}(y|x)) \vee \phi(x,y) \lesssim -\log(\delta_0 \wedge \delta_1) \asymp \log n$ for any $(x,y)$, i.e., the upper bound of $L_{\mathrm{CE}}(\widehat{p}(y|x))$ and $\phi(x,y)$ are both of logarithmic orders. These results lead to optimal convergence rates up to a logarithmic term for binary classification.

Following the idea of constructing a bounded function $\phi(x,y)$ as in Zhang et al. (2024), for multi-class cases, we obtain the function $\phi(x,y)$ of the form

$$\phi(x,y) := \begin{cases} -\log(p(y|x)), & \text{if } \underset{m\in[M]}{\forall}\, p(m|x) \in [\delta_1, 1-\delta_1], \\ 0, & \text{if } \underset{m\in[M]}{\exists}\, p(m|x) = 1, \\ \sum_{m\in[M]} -p(m|x)\log p(m|x), & \text{if } \underset{m\in[M]}{\exists}\, p(m|x) \in [0,\delta_1) \cup (1-\delta_1,1). \end{cases} \tag{33}$$

In the following example with $K = 3$, we show that to ensure that the variance bound Eq. (32) holds for any distribution $P$, the order of $V$ must be at least a polynomial order in the sample size, which significantly negatively impacts the convergence rates.

**Example C.1.** *Let $\mathcal{Y} = \{1,2,3\}$. For any $x \in \mathcal{X}$, let $\eta(x) \ge 0$ and $p(\cdot|x) = (\eta(x), 1/3, 2/3 - \eta(x))$. Suppose that for a given $n$, we have $P_X(x \in \mathcal{X} : \eta(x) \le e^{-n}) = 1$. Moreover, let $\delta_0 \in (0,0.3)$ and suppose $\widehat{p}(\cdot|x) = (\delta_0, 1/3, 2/3 - \delta_0)$ holds for any $x$.*

*By Eq. (33), we have $L_{\mathrm{CE}}(\widehat{p}(y|x)) \le -\log\delta_0$ and $\phi(x,y) \le -\log\delta_1$. In order to ensure that both upper bound of $L_{\mathrm{CE}}(\widehat{p}(y|x))$ and $\phi(x,y)$ are of logarithmic orders in the sample size, the minimum order of $\delta_0$ and $\delta_1$ should only be negative polynomial orders. Therefore, for any $x$ with $\eta(x) \le e^{-n}$, we have $\eta(x) < \delta_1$ for sufficiently large $n$. In Example C.1, we have $p(1|x) = \eta(x)$. By Eq. (33), we have $\phi(x,y) = -\sum_{m\in[3]} p(m|x)\log p(m|x)$ and thus the left hand side in Eq. (32) turns out to be*

$$\mathbb{E}\big(L_{\mathrm{CE}}(\widehat{p}(Y|X)) - \phi(X,Y)\big)^2 \ge \mathbb{E}_X\big(p(2|X)\big(L_{\mathrm{CE}}(\widehat{p}(2|X)) - \phi(X,2)\big)^2\big)$$

$$\ge \mathbb{E}_X\big((1/3) \cdot \big(\eta(X)\log\eta(X) + (2/3)\cdot\log 3 + (2/3 - \eta(X))\cdot\log(2/3 - \eta(X))\big)^2\big) \ge 0.14$$

*if $n \ge 4$. On the other hand, by using the definition of $\phi$ and Lemma 2.7 in Tsybakov (2008), the excess risk in the right hand side of Eq. (32) is*

$$\mathbb{E}\big(L_{\mathrm{CE}}(\widehat{p}(Y|X)) - \phi(X,Y)\big) = \mathbb{E}\big(L_{\mathrm{CE}}(\widehat{p}(Y|X)) - L_{\mathrm{CE}}(p(Y|X))\big)$$

$$\le \mathbb{E}_X \sum_{m=1}^{3} \frac{(p(m|X) - \widehat{p}(m|X))^2}{\widehat{p}(m|X)} \le \mathbb{E}_X\left(\frac{(\eta(X)-\delta_0)^2}{\delta_0} + \frac{(\eta(X)-\delta_0)^2}{2/3 - \delta_0}\right) \lesssim \delta_0.$$

*Therefore, to ensure that the variance bound Eq. (32) holds, we must have*

$$V \geq \frac{\mathbb{E}\big(L_{\mathrm{CE}}(\widehat{p}(Y|X)) - \phi(X, Y)\big)^2}{\mathbb{E}\big(L_{\mathrm{CE}}(\widehat{p}(Y|X)) - \phi(X, Y)\big)} \gtrsim \frac{0.14}{\delta_0} \gtrsim \delta_0^{-1}.$$

*In order to achieve a small approximation error, Theorem 2.2 in Zhang et al. (2024) selects $\delta_0 \asymp n^{-\theta}$ with the constant $\theta > \alpha/(d + \alpha)$, which implies that $V \gtrsim n^{\alpha/(d+\alpha)}$. This results in significantly slower convergence rates.*

Similar to Example C.1, we can provide examples for any $K > 3$ for which $V$ must be of some polynomial order in the sample size to satisfy the variance bound. Therefore, the direct extension of constructing a function $\phi$ to the multi-class case leads to essentially slower convergence rates.

Furthermore, the oracle inequality presented in Bos & Schmidt-Hieber (2022) does not provide any theoretical guarantees concerning the unbounded CE loss. Their results regarding the truncated excess CE risk may not hold for the true excess CE risk. Specifically, in their Theorem 3.5, the sample error bound for the truncated excess CE risk is shown to increase linearly with the truncation threshold. Consequently, if the threshold is taken to infinity to convert the truncated excess CE risk to the actual CE excess risk, the resulting sample error bound diverges to infinity.

It is important to note that our proof technique in Section 4.2.1 can be applied to other models that yield CCP predictions by minimizing the CE loss, such as neural networks. Furthermore, our approach can be generalized to other unbounded loss functions that increase more rapidly than the CE loss, including the exponential loss (Hastie et al., 2009), the large-margin unified machine (Liu et al., 2011), and the distance-weighted discrimination loss (Marron et al., 2007).

# D PROOFS

In this section, we present the proofs related to Sections 4 and 3 in Sections D.1 and D.2, respectively. To be specific, Section D.1.1 and D.1.2 derive the upper bound of the sample error and the approximation error for KLR. The proofs of the convergence rates of KLR and the lower bound of multi-class classification with respect to the CE loss are presented in Section D.1.3. Moreover, we prove the convergence rates of approaches using KLR for complex scenarios of classification problems in Section D.2.

## D.1 PROOFS RELATED TO SECTION 4

### D.1.1 PROOFS RELATED TO SECTION 4.2.1

Before we proceed, we need to introduce the following concept of entropy numbers (van der Vaart & Wellner, 1996) to measure the capacity of a function set.

**Definition D.1** (Entropy Numbers). Let $(\mathcal{X}, d)$ be a metric space, $A \subset \mathcal{X}$ and $i \geq 1$ be an integer. The $i$-th entropy number of $(A, d)$ is defined as

$$e_i(A, d) = \inf\left\{ \varepsilon > 0 : \exists x_1, \ldots, x_{2^{i-1}} \in \mathcal{X} \text{ such that } A \subset \bigcup_{j=1}^{2^{i-1}} B_d(x_j, \varepsilon) \right\}.$$

The following lemma gives the upper bound of the entropy number for Gaussian kernels.

**Lemma D.2.** *Let $\mathcal{X} \subset \mathbb{R}^d$, $p(x)$ be a distribution on $\mathcal{X}$ and let $\mathrm{supp}(P_X) \subset \mathcal{X}$ be the support of $P_X$. Moreover, for $\gamma > 0$, let $H(A)$ be the RKHS of the Gaussian RBF kernel $k_\gamma$ over the set $A$. Then, for all $N \in \mathbb{N}^*$, there exists a constant $c_{N,d} > 0$ such that*

$$e_i(\mathrm{id} : H(\mathcal{X}) \to L_2(P_X)) \leq 2^N c_{N,d} \gamma^{-N} i^{-N/d}, \qquad i > 1.$$

*Proof of Lemma D.2.* Let us consider the commutative diagram

$$
\begin{array}{ccc}
H(\mathcal{X}) & \xrightarrow{\quad \mathrm{id} \quad} & L_2(P_X) \\
{\scriptstyle \mathcal{I}_{\mathrm{supp}(P_X)}} \Big\downarrow & & \Big\uparrow {\scriptstyle \mathrm{id}} \\
H(\mathrm{supp}(P_X)) & \xrightarrow{\quad \mathrm{id} \quad} & \ell_\infty(\mathrm{supp}(P_X))
\end{array}
$$

where the extension operator $\mathcal{I}_{\mathrm{supp}(\mathcal{X})} : H_\gamma(\mathcal{X}) \to H_\gamma(\mathrm{supp}(P_X))$ given by Corollary 4.43 in Steinwart & Christmann (2008) are isometric isomorphisms such that $\|\mathcal{I}_{\mathrm{supp}(P_X)} : H_\gamma(\mathcal{X}) \to H_\gamma(\mathrm{supp}(P_X))\| = 1$.

Let $\ell_\infty(B)$ be the space of all bounded functions on $B$. Then for any $f \in \ell_\infty(B)$, we have $\|f\|_{L_2(P_X)} = (\frac{1}{n}\sum_{i=1}^{n}|f(x_i)|^2)^{1/2} \leq \|f\|_\infty$ and thus $\|\mathrm{id} : \ell_\infty(\mathrm{supp}(\mathcal{X})) \to L_2(D)\| \leq 1$. This together with (A.38), (A.39), and Theorem 6.27 in Steinwart & Christmann (2008) implies that for all $i \geq 1$ and $N \geq 1$, there holds

$$
\begin{aligned}
& e_i(\mathrm{id} : H_\gamma(\mathcal{X}) \to L_2(p(x))) \\
& \leq \|\mathcal{I}_{\mathrm{supp}(P_X)} : H_\gamma(\mathcal{X}) \to H_\gamma(\mathrm{supp}(\mathcal{X}))\| \cdot e_i(\mathrm{id} : H(\mathrm{supp}(P_X)) \to \ell_\infty(\mathrm{supp}(P_X))) \\
& \qquad \cdot \|\mathrm{id} : \ell_\infty(\mathrm{supp}(\mathcal{X})) \to L_2(P_X)\| \\
& \leq 2^N c_{N,d} \gamma^{-N} i^{-\frac{N}{d}},
\end{aligned}
$$

where $c_{N,d}$ is the constant as in (Steinwart & Christmann, 2008, Theorem 6.27). $\qquad\square$

Before we proceed, we need to introduce some notations. To this end, let $(L_{\mathrm{CE}} \circ p_f^t)(x,y) := L_{\mathrm{CE}}(y, p_f^t(x))$ and $h_{p_f^t} := L_{\mathrm{CE}} \circ p_f^t - L_{\mathrm{CE}} \circ p$. Similarly, for the upper part $L_{\mathrm{CE}}^u$ Eq. (20) and the lower part $L_{\mathrm{CE}}^l$ Eq. (21) of the CE loss, we define $(L_{\mathrm{CE}}^u \circ p_f^t)(x,y) := L_{\mathrm{CE}}^u(y, p_f^t(x))$ and $(L_{\mathrm{CE}}^l \circ p_f^t)(x,y) := L_{\mathrm{CE}}^l(y, p_f^t(x))$.

Let the function space $\mathcal{F}$ be as in Eq. (6) and $r^* := \inf_{f \in \mathcal{F}}(\lambda \sum_{m=1}^{M-1} \|f_m\|_H^2 + \mathbb{E}_P h_{p_f^t})$. For any $r \geq r^*$, we define the function space

$$
\mathcal{F}_r := \left\{ f \in \mathcal{F} : \lambda \sum_{m=1}^{M-1} \|f_m\|_H^2 + \mathbb{E}_P h_{p_f^t} \leq r \right\}
$$

and denote the upper part of the loss difference of the functions in $\mathcal{F}_r$ as

$$
\mathcal{G}_r^u := \left\{ L_{\mathrm{CE}}^u \circ p_f^t - L_{\mathrm{CE}}^u \circ p : f \in \mathcal{F}_r \right\}. \tag{34}
$$

Let $r_l^* := \inf_{f \in \mathcal{F}}(\lambda \sum_{m=1}^{M-1} \|f_m\|_H^2 + \mathbb{E}_P(L_{\mathrm{CE}}^l \circ p_f^t))$. For any $r \geq r_l^*$, we define the function space concerning the lower part by

$$
\mathcal{F}_r^l := \left\{ f \in \mathcal{F} : \lambda \sum_{m=1}^{M-1} \|f_m\|_H^2 + \mathbb{E}_P(L_{\mathrm{CE}}^l \circ p_f^t) \leq r \right\}
$$

and denote the lower part of the loss of the functions in $\mathcal{F}_r^l$ as

$$
\mathcal{G}_r^l := \left\{ L_{\mathrm{CE}}^l \circ p_f^t : f \in \mathcal{F}_r^l \right\}. \tag{35}
$$

**Lemma D.3.** *Let $\mathcal{G}_r^u$ and $\mathcal{G}_r^l$ be defined as in Eq. (34) and Eq. (35), respectively. Then we have*

$$
\begin{aligned}
e_i(\mathcal{G}_r^u, L_2(D)) &\leq 2c_{\xi,d} M^{1+1/(2\xi)} (r/\lambda)^{1/2} \gamma^{-d/(2\xi)} i^{-1/(2\xi)}, \\
e_i(\mathcal{G}_r^l, L_2(D)) &\leq 2c_{\xi,d} M^{1+1/(2\xi)} (r/\lambda)^{1/2} \gamma^{-d/(2\xi)} i^{-1/(2\xi)},
\end{aligned}
$$

*where $c_{\xi,d}$ is a constant depending only on $\xi$ and $d$.*

*Proof of Lemma D.3.* Since for any $f \in \mathcal{F}_r$, we have $\lambda \|f_m\|_H^2 \leq r$, $m \in [M-1]$. Therefore,

$$
\mathcal{F}_r \subset \{ f \in \mathcal{F} : f_m \in (r/\lambda)^{1/2} B_H, m \in [M-1] \},
$$

where $B_H := \{h \in H : \|h\|_H \leq 1\}$ is the unit ball in the space $H$. By applying Lemma D.2 with $\xi := d/(2N)$, we obtain $e_i(\mathrm{id} : H(\mathcal{X}) \to L_2(D)) \leq ai^{-1/(2\xi)}$, where $a := c_{\xi,d}\gamma^{-d/(2\xi)}$ with the constant $c_{\xi,d}$ depending only on $\xi$ and $d$. Thus we have

$$
e_i((r/\lambda)^{1/2} B_H, L_2(D)) \leq (r/\lambda)^{1/2} ai^{-1/(2\xi)}.
$$

By Definition D.1 and Lemma D.2, there exists an $\epsilon := (r/\lambda)^{1/2} a i^{-1/(2\xi)}$-net $\mathcal{N}$ of $(r/\lambda)^{1/2} B_H$ with respect to $L_2(D)$ with $|\mathcal{N}| = 2^{i-1}$. Define the function set

$$\mathcal{B} := \{g := (g_m)_{m=1}^M : g_M = 0, g_m \in \mathcal{N}, m \in [M-1]\}.$$

Then we have $|\mathcal{B}| = 2^{(i-1)(M-1)}$. Moreover, for any function $f \in \mathcal{F}_r$, there exists a $g \in \mathcal{B}$ such that $\|f_m - g_m\|_{L_2(D)} \le \epsilon$ for $m \in [M-1]$. Let us define

$$p_f(y|x) := \frac{\exp(f_y(x))}{\sum_{m=1}^M \exp(f_m(x))}$$

and truncate $p_f(y|x)$ to obtain $p_f^t(y|x)$ as in Eq. (8). By Lemma D.9, we get

$$\|L_{\mathrm{CE}}^u \circ p_f^t - L_{\mathrm{CE}}^u \circ p - (L_{\mathrm{CE}}^u \circ p_g^t - L_{\mathrm{CE}}^u \circ p)\|_{L_2(D)} = \|L_{\mathrm{CE}}^u \circ p_f^t - L_{\mathrm{CE}}^u \circ p_g^t\|_{L_2(D)}$$
$$\le \big\| -\log p_f^t + \log p_g^t \big\|_{L_2(D)} \le (4M+1)\big\| -\log p_f + \log p_g \big\|_{L_2(D)}$$
$$\le \|f_Y(X) - g_Y(X)\|_{L_2(D)} + \left\| \log \frac{\sum_{m=1}^M \exp(f_m(X))}{\sum_{m=1}^M \exp(g_m(X))} \right\|_{L_2(D)}. \tag{36}$$

For any $a > 0$ and $z \in \mathbb{R}$, the derivative function of the function $h(z) := \log(a + \exp(z))$ is $h'(z) = \exp(z)/(a + \exp(z)) \in (0, 1)$. Therefore, by the Lagrange mean value theorem, we have $|h(z) - h(z')| = |h'(\theta z + (1-\theta)z') \cdot (z - z')| \le |z - z'|$. Applying this to $a := \sum_{m=1}^{\ell-1} \exp(g_m(X)) + \sum_{m=\ell+1}^M \exp(f_m(X))$, $z = f_\ell(X)$ and $z' = g_\ell(X)$ for $\ell \in [K]$, we get

$$\left| \log \frac{\sum_{m=1}^M \exp(f_m(X))}{\sum_{m=1}^M \exp(g_m(X))} \right| = \left| \sum_{\ell=1}^M \log \frac{\sum_{m=1}^{\ell-1} \exp(g_m(X)) + \sum_{m=\ell}^M \exp(f_m(X))}{\sum_{m=1}^\ell \exp(g_m(X)) + \sum_{m=\ell+1}^M \exp(f_m(X))} \right|$$
$$\le \sum_{\ell=1}^M \left| \log \frac{\sum_{m=1}^{\ell-1} \exp(g_m(X)) + \sum_{m=\ell}^M \exp(f_m(X))}{\sum_{m=1}^\ell \exp(g_m(X)) + \sum_{m=\ell+1}^M \exp(f_m(X))} \right| \le \sum_{\ell=1}^M |f_\ell(X) - g_\ell(X)|.$$

This together with Eq. (36) yields

$$\|L_{\mathrm{CE}}^u \circ p_f^t - L_{\mathrm{CE}}^u \circ p - (L_{\mathrm{CE}}^u \circ p_g^t - L_{\mathrm{CE}}^u \circ p)\|_{L_2(D)}$$
$$\le \|f_Y(X) - g_Y(X)\|_{L_2(D)} + \sum_{\ell=1}^M \|f_\ell(X) - g_\ell(X)\|_{L_2(D)} \le M\epsilon.$$

Therefore, we get $\|L_{\mathrm{CE}}^u \circ p_f^t - L_{\mathrm{CE}}^u \circ p - (L_{\mathrm{CE}}^u \circ p_g^t - L_{\mathrm{CE}}^u \circ p)\|_{L_2(D)} \le M\epsilon$. Thus, the function set $\{L_{\mathrm{CE}}^u \circ p_f^t - L_{\mathrm{CE}}^u \circ p : f \in \mathcal{B}\}$ is a $(M\epsilon)$-net of $\mathcal{G}_r^u$. A similar analysis yields that the function set $\{L_{\mathrm{CE}}^l \circ p_f^t : f \in \mathcal{B}\}$ is a $(M\epsilon)$-net of $\mathcal{G}_r^l$. These together with (A.36) in Steinwart & Christmann (2008) yield

$$e_{(M-1)i}(\mathcal{G}_r^u, L_2(D)) \le 2M\varepsilon = 2M(r/\lambda)^{1/2} a i^{-1/(2\xi)},$$
$$e_{(M-1)i}(\mathcal{G}_r^l, L_2(D)) \le 2M\varepsilon = 2M(r/\lambda)^{1/2} a i^{-1/(2\xi)},$$

which are equivalent to

$$e_i(\mathcal{G}_r^u, L_2(D)) \le 2M\varepsilon = 2M^{1+\frac{1}{2\xi}}(r/\lambda)^{\frac{1}{2}} a i^{-\frac{1}{2\xi}} = c_{\xi,d} 2M^{1+\frac{1}{2\xi}}(r/\lambda)^{\frac{1}{2}} \gamma^{-\frac{d}{2\xi}} i^{-\frac{1}{2\xi}},$$
$$e_i(\mathcal{G}_r^l, L_2(D)) \le 2M\varepsilon = 2M^{1+\frac{1}{2\xi}}(r/\lambda)^{\frac{1}{2}} a i^{-\frac{1}{2\xi}} = c_{\xi,d} 2M^{1+\frac{1}{2\xi}}(r/\lambda)^{\frac{1}{2}} \gamma^{-\frac{d}{2\xi}} i^{-\frac{1}{2\xi}}.$$

This finishes the proof. $\qquad\square$

**Lemma D.4.** *Let $P$ be a probability distribution on $\mathcal{X} \times \mathcal{Y}$. Let $t \in (0, 1/(2M))$ and $p_f^t$ be the truncation of $p_f$ as in Eq. (8). Then for any $V \ge -2\log t + 2$, there holds*

$$\mathbb{E}_P\big(L_{\mathrm{CE}}(Y, p_f^t(\cdot|X)) - L_{\mathrm{CE}}(Y, p(\cdot|X))\big)^2 \le V \cdot \mathbb{E}_P\big(L_{\mathrm{CE}}(Y, p_f^t(\cdot|X)) - L_{\mathrm{CE}}(Y, p(\cdot|X))\big).$$

*Proof of Lemma D.4.* By definition of the CE loss, we have

$$\mathbb{E}_P\big(L_{\mathrm{CE}}(Y, p_f^t(\cdot|X)) - L_{\mathrm{CE}}(Y, p(\cdot|X)))\big)^2 = \mathbb{E}_{x\sim p}\sum_{m=1}^M p(m|x)\Big(\log\frac{p(m|x)}{p_f^t(m|x)}\Big)^2,$$

$$\mathbb{E}_P\big(L_{\mathrm{CE}}(Y, p_f^t(\cdot|X)) - L_{\mathrm{CE}}(Y, p(\cdot|X)))\big) = \mathbb{E}_{x\sim p}\sum_{m=1}^M p(m|x)\Big(\log\frac{p(m|x)}{p_f^t(m|x)}\Big).$$

For $\theta \in \mathbb{R}$, we define the function $h$ by

$$h(p_f^t(\cdot|x)) := \sum_{m=1}^M p(m|x)\Big(\log\frac{p(m|x)}{p_f^t(m|x)}\Big)^2$$
$$- V\sum_{m=1}^M p(m|x)\Big(\log\frac{p(m|x)}{p_f^t(m|x)}\Big) + \theta\Big(\sum_{m=1}^M p_f^t(m|x) - 1\Big).$$

Then we have

$$\frac{\partial h(p_f^t(\cdot|x))}{\partial p_f^t(m|x)} = 2\cdot\frac{p(m|x)}{p_f^t(m|x)}\cdot\log\frac{p_f^t(m|x)}{p(m|x)} + V\cdot\frac{p(m|x)}{p_f^t(m|x)} + \theta$$
$$= \frac{p(m|x)}{p_f^t(m|x)}\cdot\Big(-2\log\frac{p(m|x)}{p_f^t(m|x)} + V\Big) + \theta.$$

Let $g(x) := x(-2\log x + V) + \theta$ for $x \in [0, 1/t]$ and $V \geq -2\log t + 2$. Since the derivative of $g$ is $g'(x) = -2(\log x + 1) + V \geq 0$, the function $g(x)$ is non-decreasing with respect to $x$. Therefore, the zero point of $\partial h(p_f^t(\cdot|x))/\partial p_f^t(m|x)$ is the same for all $m \in [M]$. In other words, $p(m|x)/p_f^t(m|x)$ should be the same and thus we have $p_f^t(m|x) = p(m|x)$ due to the constraint $\sum_{m=1}^M p_f^t(m|x) = 1$. Therefore, $p_f^t(m|x) := p(m|x)$ attains the maximum of

$$\sum_{m=1}^M p(m|x)\Big(\log\frac{p(m|x)}{p_f^t(m|x)}\Big)^2 - V\Big(\sum_{m=1}^M p(m|x)\Big(\log\frac{p(m|x)}{p_f^t(m|x)}\Big)\Big)$$

which turns out to be zero. Consequently, for any $p_f^t(m|x)$ satisfying $\sum_{m=1}^M p_f^t(m|x) = 1$, there holds

$$\sum_{m=1}^M p(m|x)\Big(\log\frac{p(m|x)}{p_f^t(m|x)}\Big)^2 \leq V\Big(\sum_{m=1}^M p(m|x)\Big(\log\frac{p(m|x)}{p_f^t(m|x)}\Big)\Big),$$

which finishes the proof. $\square$

The following lemma provides the variance bound for the lower part of the CE loss function and the upper bound for the lower part of the CE risk of the truncated estimator.

**Lemma D.5.** *Let $L_{\mathrm{CE}}^l$ be the lower part of the CE loss function as in Eq. (21) with $t \in (0, 1/(2M))$. Then for any $f : \mathcal{X} \to \mathbb{R}^M$ and any $t \in (0, 1/(2M))$, we have*

$$\mathbb{E}_P(L_{\mathrm{CE}}^l \circ p_f^t)^2 \leq (-\log t)\cdot\mathbb{E}_P(L_{\mathrm{CE}}^l \circ p_f^t), \qquad \mathbb{E}_P(L_{\mathrm{CE}}^l \circ p_f^t) \leq -Mt\log t.$$

*Proof of Lemma D.5.* By the definition of $L_{\mathrm{CE}}^l \circ p_f^t$, we have

$$\mathbb{E}_P(L_{\mathrm{CE}}^l \circ p_f^t)^2 = \mathbb{E}_{x\sim p}\sum_{m=1}^M p(m|x)\mathbf{1}\{p(m|x) < t\}(-\log p_f^t(m|x))^2,$$

$$\mathbb{E}_P(L_{\mathrm{CE}}^l \circ p_f^t) = \mathbb{E}_{x\sim p}\sum_{m=1}^M p(m|x)\mathbf{1}\{p(m|x) < t\}(-\log p_f^t(m|x)).$$

Since $p_f^t(m|x) \in [t, 1)$ for any $m \in [M]$, we have $-\log p_f^t(m|x) \in (0, -\log t]$. Thus we obtain

$$\mathbb{E}_P(L_{\mathrm{CE}}^l \circ p_f^t)^2 \leq \mathbb{E}_{x \sim p} \sum_{m=1}^{M} p(m|x)\mathbf{1}\{p(m|x) < t\}(-\log t) \cdot (-\log p_f^t(m|x))$$

$$\leq (-\log t) \cdot \mathbb{E}_{x \sim p} \sum_{m=1}^{M} p(m|x)\mathbf{1}\{p(m|x) < t\}(-\log p_f^t(m|x)) = (-\log t)\mathbb{E}_P(L_{\mathrm{CE}}^l \circ p_f^t),$$

which proves the first assertion. Moreover, we have

$$\mathbb{E}_P(L_{\mathrm{CE}}^l \circ p_f^t) = \mathbb{E}_{x \sim p} \sum_{m=1}^{M} \mathbf{1}\{p(m|x) < t\}p(m|x)(-\log p_f^t(m|x))$$

$$\leq \mathbb{E}_{x \sim p} \sum_{m=1}^{M} \mathbf{1}\{p(m|x) < t\}t(-\log t) \leq -Mt\log t,$$

which proves the second assertion. $\qquad\square$

Before we proceed, we need to introduce another concept to measure the capacity of a function set, which is a type of expectation of supermum with respect to the Rademacher sequence, see e.g., Definition 7.9 in Steinwart & Christmann (2008).

**Definition D.6** (Empirical Rademacher Average). Let $\{\varepsilon_i\}_{i=1}^m$ be a Rademacher sequence with respect to some distribution $\nu$, that is, a sequence of i.i.d. random variables, such that $\nu(\varepsilon_i = 1) = \nu(\varepsilon_i = -1) = 1/2$. The $n$-th empirical Rademacher average of $\mathcal{F}$ is defined as

$$\mathrm{Rad}_D(\mathcal{F}, n) := \mathbb{E}_\nu \sup_{h \in \mathcal{F}} \left| \frac{1}{n} \sum_{i=1}^{n} \varepsilon_i h(x_i) \right|.$$

*Proof of Theorem 4.1.* By the definition of $L_{\mathrm{CE}}^u$ in Eq. (20) and $L_{\mathrm{CE}}^l$ in Eq. (21), we have $L_{\mathrm{CE}} = L_{\mathrm{CE}}^u + L_{\mathrm{CE}}^l$. Let us denote $h_{p_f^t} := L_{\mathrm{CE}} \circ p_f^t - L_{\mathrm{CE}} \circ p$, $h_{p_f^t}^u := L_{\mathrm{CE}}^u \circ p_f^t - L_{\mathrm{CE}}^u \circ p$ and $g_{p_f^t}^l := L_{\mathrm{CE}}^l \circ p_f^t$. By Eq. (10), we have $\widehat{p}(y|x)) = p_{f_D}^t(y|x)$. Then by Eq. (9), for any $f_0 \in \mathcal{F}$, we have $\lambda\|f_D\|_H^2 + \mathcal{R}_{L_{\mathrm{CE}}, D}(p_{f_D}^t(y|x)) \leq \lambda\|f_0\|_H^2 + \mathcal{R}_{L_{\mathrm{CE}}, D}(p_{f_0}^t(y|x))$ and consequently

$$\lambda\|f_D\|_H^2 + \mathcal{R}_{L_{\mathrm{CE}}, P}(\widehat{p}(y|x)) - \mathcal{R}_{L_{\mathrm{CE}}, P}^* = \lambda\|f_D\|_H^2 + \mathbb{E}_P h_{\widehat{p}} = \lambda\|f_D\|_H^2 + \mathbb{E}_P h_{p_{f_D}^t}$$

$$= \lambda\|f_D\|_H^2 + \mathbb{E}_D h_{p_{f_D}^t} - \mathbb{E}_D h_{p_{f_D}^t} + \mathbb{E}_P h_{p_{f_D}^t}$$

$$\leq \lambda\|f_0\|_H^2 + \mathbb{E}_D h_{p_{f_0}^t} - \mathbb{E}_D h_{p_{f_D}^t} + \mathbb{E}_P h_{p_{f_D}^t}$$

$$= \lambda\|f_0\|_H^2 + \mathbb{E}_P h_{p_{f_0}^t} + \mathbb{E}_D h_{p_{f_0}^t} - \mathbb{E}_P h_{p_{f_0}^t} + \mathbb{E}_P h_{p_{f_D}^t} - \mathbb{E}_D h_{p_{f_D}^t}$$

$$= \lambda\|f_0\|_H^2 + \mathbb{E}_P h_{p_{f_0}^t} + (\mathbb{E}_D h_{p_{f_0}^t}^l - \mathbb{E}_P h_{p_{f_0}^t}^l + \mathbb{E}_P h_{p_{f_D}^t}^l - \mathbb{E}_D h_{p_{f_D}^t}^l)$$

$$\quad + (\mathbb{E}_D h_{p_{f_0}^t}^u - \mathbb{E}_P h_{p_{f_0}^t}^u + \mathbb{E}_P h_{p_{f_D}^t}^u - \mathbb{E}_D h_{p_{f_D}^t}^u)$$

$$= \lambda\|f_0\|_H^2 + \mathbb{E}_P h_{p_{f_0}^t} + (\mathbb{E}_D L_{\mathrm{CE}}^l \circ p_{f_0}^t - \mathbb{E}_P L_{\mathrm{CE}}^l \circ p_{f_0}^t + \mathbb{E}_P L_{\mathrm{CE}}^l \circ p_{f_D}^t - \mathbb{E}_D L_{\mathrm{CE}}^l \circ p_{f_D}^t)$$

$$\quad + (\mathbb{E}_D h_{p_{f_0}^t}^u - \mathbb{E}_P h_{p_{f_0}^t}^u + \mathbb{E}_P h_{p_{f_D}^t}^u - \mathbb{E}_D h_{p_{f_D}^t}^u)$$

$$= \lambda\|f_0\|_H^2 + \mathbb{E}_P h_{p_{f_0}^t} + \mathbb{E}_D(g_{p_{f_0}^t}^l - \mathbb{E}_P g_{p_{f_0}^t}^l) + \mathbb{E}_D(\mathbb{E}_P g_{p_{f_D}^t}^l - g_{p_{f_D}^t}^l)$$

$$\quad + \mathbb{E}_D(h_{p_{f_0}^t}^u - \mathbb{E}_P h_{p_{f_0}^t}^u) + \mathbb{E}_D(\mathbb{E}_P h_{p_{f_D}^t}^u - h_{p_{f_D}^t}^u), \tag{37}$$

where $\mathbb{E}_D h_{p_f^t} := n^{-1} \sum_{i=1}^{n} h_{p_f^t}(X_i, Y_i)$. In the following, we provide the estimates for the last four terms in Eq. (37).

For any $f \in \mathcal{F}$, we observe that $\|g_{p_f^t}^l - \mathbb{E}_P g_{p_f^t}^l\|_\infty = \|L_{\mathrm{CE}}^l \circ p_f^t - \mathbb{E}_P L_{\mathrm{CE}}^l \circ p_f^t\|_\infty \leq -\log t$. By Lemma D.5, we have

$$\mathbb{E}_P(g_{p_f^t}^l - \mathbb{E}_P g_{p_f^t}^l)^2 \leq \mathbb{E}_P(g_{p_f^t}^l)^2 \leq (-\log t) \cdot \mathbb{E}_P g_{p_f^t}^l. \tag{38}$$

Applying Bernstein's inequality in (Steinwart & Christmann, 2008, Theorem 6.12) to $\{g^l_{p^t_f}(X_i, Y_i) - \mathbb{E}_P g^l_{p^t_f} : i \in [n]\}$, we obtain that

$$
\mathbb{E}_D(g^l_{p^{t}_{f_0}} - \mathbb{E}_P g^l_{p^{t}_{f_0}}) \leq \sqrt{2\zeta(-\log t)\mathbb{E}_P g^l_{p^{t}_{f_0}}/n} + 4(-\log t)\zeta/(3n)
$$
$$
\leq \zeta(-\log t/2)/n + \mathbb{E}_P g^l_{p^{t}_{f_0}} + (-4\zeta\log t)/(3n) \leq \mathbb{E}_P g^l_{p^{t}_{f_0}} - 2\zeta\log t/n \qquad (39)
$$

holds with probability $P^n$ at least $1 - e^{-\zeta}$, where the last inequality is due to $2ab \leq a^2 + b^2$. To estimate the term $\mathbb{E}_D g^l_{p^{t}_{f_0}} - \mathbb{E}_P g^l_{p^{t}_{f_0}}$, let us define the function

$$
G_{f,r} := \frac{\mathbb{E}_P g^l_{p^t_f} - g^l_{p^t_f}}{\lambda\|f\|^2_H + \mathbb{E}_P g^l_{p^t_f} + r}, \qquad f \in \mathcal{F}, \ r > r^*_l,
$$

where $r^*_l := \inf\{f \in \mathcal{F} : \lambda\|f\|^2_H + \mathbb{E}_P g^l_{p^t_f}\}$. Then we have $\|G_{f,r}\|_\infty \leq (-\log t)/r$. By Eq. (38), we have

$$
\mathbb{E}_P G^2_{f,r} \leq \frac{\mathbb{E}_P(g^l_{p^t_f} - \mathbb{E}_P g^l_{p^t_f})^2}{(\mathbb{E}_P g^l_{p^t_f} + r)^2} \leq \frac{\mathbb{E}_P(g^l_{p^t_f})^2}{2r\mathbb{E}_P g^l_{p^t_f}} \leq \frac{-\log t}{2r}.
$$

Let $\mathcal{G}^l_r := \{g^l_{p^t_f} : f \in \mathcal{F}^l_r\}$ and $\mathcal{F}^l_r := \{f \in \mathcal{F} : \lambda\|f\|^2_H + \mathbb{E}_P g^l_{p^t_f} \leq r\}$. Symmetrization in Proposition 7.10 of Steinwart & Christmann (2008) yields

$$
\mathbb{E}_{D\sim P^n} \sup_{f\in\mathcal{F}^l_r} |\mathbb{E}_D(\mathbb{E}_P g^l_{p^t_f} - g^l_{p^t_f})| \leq 2\mathbb{E}_{D\sim P^n}\mathrm{Rad}_D(\mathcal{G}^l_r, n) \leq 2\psi_n(r).
$$

For any $L^l_{\mathrm{CE}} \circ p^t_f \in \mathcal{G}^l_r$, we have $\|L^l_{\mathrm{CE}} \circ p^t_f\|_\infty \leq -\log t$ and $\mathbb{E}_P(L^l_{\mathrm{CE}} \circ p^t_f)^2 \leq -\log t \cdot \mathbb{E}_P(L^l_{\mathrm{CE}} \circ p^t_f) \leq -r\log t$. By applying Theorem 7.16 in Steinwart & Christmann (2008) and Lemma D.3, we obtain

$$
\mathbb{E}_{D\sim P^n}\mathrm{Rad}_D(\mathcal{G}^l_r, n)
$$
$$
\leq C\big(r^{\frac{1}{2}}\lambda^{-\frac{\xi}{2}}\gamma^{-d/2}(-\log t)^{\frac{1-\xi}{2}}n^{-\frac{1}{2}} \vee (r/\lambda)^{\frac{\xi}{1+\xi}}\gamma^{-\frac{d}{1+\xi}}(-\log t)^{\frac{1-\xi}{1+\xi}}n^{-\frac{1}{1+\xi}}\big) =: \psi_n(r), \qquad (40)
$$

where $C := C_1(\xi)c^\xi_{\xi,d}M^{\xi/2+1}2^{2-\xi} \vee C_2(\xi)c^{2\xi/(1+\xi)}_{\xi,d}2M^{(2\xi+1)/(1+\xi)}2^{(1-\xi)/(1+\xi)}$. Thus we have

$$
\mathbb{E}_{D\sim P^n} \sup_{f\in\mathcal{F}^l_r} |\mathbb{E}_D(\mathbb{E}_P g^l_{p^t_f} - g^l_{p^t_f})| \leq 2\psi_n(r).
$$

It is easy to verify that $\psi_n(4r) \leq 2\psi_n(r)$. Then by applying the peeling technique in Theorem 7.7 of Steinwart & Christmann (2008) on $\mathcal{F}^l_r$, we obtain

$$
\mathbb{E}_{D\sim P^n} \sup_{f\in\mathcal{F}} |\mathbb{E}_D G_{f,r}| \leq 8\psi_n(r)/r.
$$

Applying Talagrand's inequality in Theorem 7.5 of Steinwart & Christmann (2008) to $\gamma := 1/4$, we obtain that for any $r > r^*_l$, with probability at least $1 - e^{-\zeta}$, there holds

$$
\sup_{f\in\mathcal{F}} \mathbb{E}_D G_{f,r} < 10\psi_n(r)/r + \sqrt{-\log t\zeta/(nr)} + (-14\zeta\log t)/(3nr).
$$

By the definition of $g_{f_D,r}$, we have

$$
\mathbb{E}_P g^l_{p^t_{f_D}} - \mathbb{E}_D g^l_{p^t_{f_D}}
$$
$$
< \big(\lambda\|f_D\|^2_H + \mathbb{E}_P g^l_{p^t_{f_D}}\big)\big(10\psi_n(r)/r + \sqrt{-\zeta\log t/(nr)} + (-14\zeta\log t)/(3nr)\big)
$$
$$
+ 10\psi_n(r) + \sqrt{-\zeta r\log t/n} + (-14\zeta\log t)/(3n) \qquad (41)
$$

with probability at least $1 - e^{-\zeta}$. Subsequently, we estimate the term $\mathbb{E}_D h^u_{p^t_{f_0}} - \mathbb{E}_P h^u_{p^t_{f_0}}$ in Eq. (37). For any $f \in \mathcal{F}$, we observe that $\|h^u_{p^t_f} - \mathbb{E}_P h^u_{p^t_f}\|_\infty \leq -2\log t$. Using the variance bound in Lemma D.4 and $t \leq 1/(2M) \leq 1/e$, we get

$$
\mathbb{E}_P(h^u_{p^t_f} - \mathbb{E}_P h^u_{p^t_f})^2 \leq \mathbb{E}_P(h^u_{p^t_f})^2 \leq \mathbb{E}_P(h_{p^t_f})^2 \leq (-2\log t + 2) \cdot \mathbb{E}_P(h_{p^t_f}) \leq -4\log t\mathbb{E}_P(h_{p^t_f}).
$$

Then by applying Bernstein's inequality in (Steinwart & Christmann, 2008, Theorem 6.12) to $\{h^u_{p^t_{f_0}}(X_i, Y_i) - \mathbb{E}_P h^u_{p^t_{f_0}} : i \in [n]\}$ and $2ab \le a^2 + b^2$, we obtain

$$
\begin{aligned}
\mathbb{E}_D(h^u_{p^t_{f_0}} - \mathbb{E}_P h^u_{p^t_{f_0}}) &\le \sqrt{2\zeta(-4\log t)\mathbb{E}_P h_{p^t_{f_0}}/n} + 4(-2\log t)\zeta/(3n) \\
&\le -2\zeta\log t/n + \mathbb{E}_P h_{p^t_{f_0}} + (-8\zeta\log t)/(3n) = \mathbb{E}_P h_{p^t_{f_0}} - 14\zeta\log t/(3n).
\end{aligned}
\tag{42}
$$

To estimate the term $\mathbb{E}_P h^u_{p^t_{f_D}} - \mathbb{E}_D h^u_{p^t_{f_D}}$ in Eq. (37), we define the function

$$
H_{f,r} := \frac{\mathbb{E}_P h^u_{p^t_f} - h^u_{p^t_f}}{\lambda\|f\|^2_H + \mathbb{E}_P h_{p^t_f} + r}, \qquad f \in \mathcal{F},\ r > r^*.
$$

Then we have $\|H_{f,r}\|_\infty \le -2\log t/r$ and the variance bound in Lemma D.4 yields

$$
\mathbb{E}_P H^2_{f,r} \le \frac{\mathbb{E}_P(h^u_{p^t_f})^2}{(\mathbb{E}_P h_{p^t_f} + r)^2} \le \frac{\mathbb{E}_P(h_{p^t_f})^2}{2r\mathbb{E}_P h_{p^t_f}} \le \frac{1}{r}(-\log t + 1) \le \frac{-2\log t}{r}.
$$

Let $\mathcal{G}^u_r := \{h^u_{p^t_f} : f \in \mathcal{F}_r\}$ and $\mathcal{F}_r := \{f \in \mathcal{F} : \lambda\sum_{m\in[M-1]}\|f_m\|^2_H + \mathbb{E}_P h_{p^t_f} \le r\}$. Symmetrization in Proposition 7.10 of Steinwart & Christmann (2008) yields

$$
\mathbb{E}_{D\sim P^n} \sup_{f\in\mathcal{F}_r} |\mathbb{E}_D(\mathbb{E}_P h^u_{p^t_f} - h^u_{p^t_f})| \le 2\mathbb{E}_{D\sim P^n}\mathrm{Rad}_D(\mathcal{G}^u_r, n) \le 2\psi_n(r),
$$

where the second inequality can be proved in a similar way as in proving Eq. (40). Peeling in Theorem 7.7 of Steinwart & Christmann (2008) together with $\mathcal{F}_r$ hence gives

$$
\mathbb{E}_{D\sim P^n} \sup_{f\in\mathcal{F}} |\mathbb{E}_D H_{f,r}| \le 8\psi_n(r)/r.
$$

Applying Talagrand's inequality in the form of Theorem 7.5 of Steinwart & Christmann (2008) applied to $\gamma := 1/4$, we therefore obtain for any $r > r^*$,

$$
\sup_{f\in\mathcal{F}} \mathbb{E}_D H_{f,r} < 10\psi_n(r)/r + \sqrt{2(-2\log t)\zeta/(nr)} - 14\zeta\log t/(3nr)
$$

holds with probability at least $1 - e^{-\zeta}$. Using the definition of $H_{f_D,r}$, we obtain

$$
\begin{aligned}
&\mathbb{E}_P h^u_{p^t_{f_D}} - \mathbb{E}_D h^u_{p^t_{f_D}} \\
&< \big(\lambda\|f_D\|^2_H + \mathbb{E}_P h_{p^t_{f_D}}\big)\big(10\psi_n(r)/r + \sqrt{2(-2\log t)\zeta/(nr)} - 14\zeta\log t/(3nr)\big) \\
&\quad + 10\psi_n(r) + \sqrt{2(-2\log t)r\zeta/n} - 14\zeta\log t/(3n)
\end{aligned}
\tag{43}
$$

with probability at least $1 - e^{-\zeta}$. Combining Eq. (37), Eq. (39), Eq. (41), Eq. (42) and Eq. (43), we obtain

$$
\begin{aligned}
&\lambda\|f_D\|^2_H + \mathbb{E}_P h_{p^t_{f_D}} \le \lambda\|f_0\|^2_H + 2\mathbb{E}_P h_{p^t_{f_0}} + \mathbb{E}_P g^l_{p^t_{f_0}} - 8\zeta\log t/n \\
&\quad + \big(2\lambda\|f_D\|^2_H + \mathbb{E}_P g^l_{p^t_{f_D}} + \mathbb{E}_P h_{p^t_{f_D}}\big)\big(10\psi_n(r)/r + \sqrt{-4\zeta\log t/(nr)} - 14\zeta\log t/(3nr)\big) \\
&\quad + 20\psi_n(r) + 3\sqrt{-r\zeta\log t/n} - 28\zeta\log t/(3n)
\end{aligned}
$$

with probability at least $1 - 4e^{-\zeta}$.

Now, it suffices to bound the various terms. If we take $r \ge 900C^2\lambda^{-\xi}\gamma^{-d}(-\log t)^{1-\xi}n^{-1}$, then by elementary calculation, we get $\psi_n(r) \le r/30$. Moreover, let $r \ge -2304\zeta\log t/n$ and thus we get $\sqrt{-r\zeta\log t/n} \le r/48$, $-28\zeta\log t/(3nr) \le 1/60$, and $8\zeta\log t/n \le r/288$. By Lemma D.5, we have $\mathbb{E}_P g^l_{p^t_{f_0}} \le -Mt\log t$. Therefore, for any $r \ge 900C^2\lambda^{-\xi}\gamma^{-d}(-\log t)^{1-\xi}n^{-1} \vee -2304\zeta\log t/n \vee r^* \vee r^*_l$, we get

$$
\begin{aligned}
&\lambda\|f_D\|^2_H + \mathbb{E}_P h_{p^t_{f_D}} \le 2(\lambda\|f_0\|^2_H + \mathbb{E}_P h_{p^t_{f_0}}) - Mt\log t + r/288 \\
&\quad + \big(2\lambda\|f_D\|^2_H + \mathbb{E}_P h_{p^t_{f_D}} - Mt\log t\big)(1/3 + 1/24 + 1/120) + 2r/3 + r/16 + r/60
\end{aligned}
$$

$$\leq 2(\lambda\|f_0\|_H^2 + \mathbb{E}_P h_{p_{f_0}^t}) + (23/30)\big(\lambda\|f_D\|_H^2 + \mathbb{E}_P h_{p_{f_D}^t}\big) - (83/60)Mt\log t + (9/10)r$$

with probability at least $1 - 4e^{-\zeta}$. By the definition of $r_l^*$ and $r^*$, and Lemma D.5, we have $r_l^* \leq \lambda\|f_0\|_H^2 + \mathbb{E}_P g_{p_{f_0}^l}^l \leq r^* - Mt\log t$ and $r^* \leq \lambda\|f_0\|_H^2 + \mathbb{E}_P h_{p_{f_0}^t}$. By some elementary calculations and taking $r := 900C^2\lambda^{-\xi}\gamma^{-d}(-\log t)^{1-\xi}n^{-1} - 2304\zeta\log t/n + r^* + r_l^*$, we get

$$\begin{aligned}
\lambda\|f_D\|_H^2 + \mathbb{E}_P h_{p_{f_D}^t} &\leq 10(\lambda\|f_0\|_H^2 + \mathbb{E}_P h_{p_{f_0}^t}) - 7Mt\log t + 5r \\
&\leq 10(\lambda\|f_0\|_H^2 + \mathbb{E}_P h_{p_{f_0}^t}) - 7Mt\log t + 4500C^2\lambda^{-\xi}\gamma^{-d}(-\log t)^{1-\xi}n^{-1} \\
&\quad - 11520\zeta\log t/n + 5r^* + 5r_l^* \\
&\leq 20(\lambda\|f_0\|_H^2 + \mathbb{E}_P h_{p_{f_0}^t}) + C_0(-\log t) \cdot (t + \lambda^{-\xi}\gamma^{-d}n^{-1} + \zeta/n)
\end{aligned}$$

with probability at least $1 - 4e^{-\zeta}$, where $C_0 := (5 + 12M) \vee 4500C^2 \vee 11520$. This proves the assertion. $\qquad\square$

### D.1.2 Proofs Related to Section 4.2.2

The goal of this section is to derive an upper bound for the approximation error of KLR, i.e. an upper bound for

$$\inf_{f\in\mathcal{F}} \lambda\|f\|_H^2 + \mathcal{R}_{L_{\mathrm{CE}},P}(p_f^t(y|x)) - \mathcal{R}_{L_{\mathrm{CE}},P}^*.$$

To construct a function that approximates the bounded function $f_m^{*\tau}$ in Eq. (26), for a fixed bandwidth $\gamma > 0$ entering the Gaussian kernel, we define the function $K : \mathbb{R}^d \to \mathbb{R}$ by

$$K(x) := \big(2/(\gamma^2\pi)\big)^{d/2} \exp\big(-2\|x\|_2^2/\gamma^2\big). \tag{44}$$

Then we define the convolution of $f_m^{*\tau}$ and $K$ as

$$\widetilde{f}_m^\tau(x) := (K * f_m^{*\tau})(x) := \int_{\mathbb{R}^d} K(x-z) f_m^{*\tau}(z)\,dz, \qquad m \in [M], \tag{45}$$

and the score function $\widetilde{f}^\tau := (\widetilde{f}_m^\tau)_{m\in[M]}$.

Let us denote

$$g_m^\tau(x) := \log p^\tau(m|x). \tag{46}$$

Then $f_m^{*\tau}(x)$ in Eq. (26) can be expressed as

$$f_m^{*\tau}(x) = g_m^\tau(x) - g_M^\tau(x). \tag{47}$$

To analyze the approximation error of $\widetilde{f}^\tau$, we first need the following lemma.

**Lemma D.7.** *Let Assumption 3.1 hold. Moreover, let $g_m^\tau$ be defined as in Eq. (46). Then for any $x, x' \in \mathcal{X}$, we have*

$$\big|g_m^\tau(x) - g_m^\tau(x')\big| \leq \frac{\log\big(1 + c_L\|x - x'\|_2^\alpha\big)}{p^\tau(m|x) \wedge p^\tau(m|x')}.$$

To prove Lemma D.7, we need the following Lemmas D.8 and D.9.

**Lemma D.8.** *For any $c > 1$ and $z > 0$, we have $\log(1 + cz) \leq c\log(1 + z)$.*

*Proof of Lemma D.8.* For any $c > 1$ and $z > 0$, define $h(z) := \log(1 + cz) - c\log(1 + z)$. Then we have

$$h'(z) = \frac{c}{1 + cz} - \frac{c}{1 + z} = \frac{c(1 - c)z}{(1 + cz)(1 + z)} < 0$$

and thus $h$ is decreasing on $[0, \infty]$. Therefore, for any $z > 0$, there holds $h(z) < h(0) = 0$. Then the definition of $h$ yields the conclusion. $\qquad\square$

**Lemma D.9.** *Let Assumption 3.1 hold. Moreover, let $p^\tau(m|x)$ be defined as in Eq. (8) with $p_f$ replaced by $p$. Then for any $x, x' \in \mathcal{X}$, there holds*

$$\left| p^\tau(m|x) - p^\tau(m|x') \right| \le (4M+1)|p(m|x) - p(m|x')| \le c_L \|x - x'\|_2^\alpha,$$

*where $c_L := (4M+1)c_\alpha$.*

*Proof of Lemma D.9.* Given $\tau \in (0, 1/(2M))$, we define the label set $\mathcal{M}_{p,x}^\tau := \{m \in [M] : p_f(m|x) < \tau\} \subset [M]$. By Eq. (8) and Assumption 3.1 *(i)*, for any $m \in \mathcal{M}_{p,x}^\tau$, there holds

$$\left| p^\tau(m|x) - p^\tau(m|x') \right| = \begin{cases} 0, & \text{if } m \in \mathcal{M}_{p,x'}^\tau, \\ p(m|x') - \tau < p(m|x') - p(m|x) \le c_\alpha \|x - x'\|_2^\alpha, & \text{if } m \notin \mathcal{M}_{p,x'}^\tau. \end{cases}$$

For any $m \notin \mathcal{M}_{p,x}^\tau$ and $m \in \mathcal{M}_{p,x'}^\tau$, by using Assumption 3.1 *(i)*, we get

$$|p^\tau(m|x) - p^\tau(m|x')| = p^\tau(m|x) - \tau \le p(m|x) - p(m|x') \le c_\alpha \|x - x'\|_2^\alpha.$$

Otherwise, for any $m \notin \mathcal{M}_{p,x}^\tau$ and $m \notin \mathcal{M}_{p,x'}^\tau$, since $\tau \in (0, 1/(2M))$, then for any $x \in \mathcal{X}$, we have

$$\sum_{\ell \notin \mathcal{M}_{p,x}^\tau} (p(\ell|x) - \tau) - \sum_{j \in \mathcal{M}_{p,x}^\tau} (\tau - p(j|x)) = \sum_{\ell=1}^M (p(\ell|x) - \tau) = 1 - M\tau > 1/2. \tag{48}$$

Therefore, by the triangle inequality and Assumption 3.1 *(i)*, there holds

$$\left| p^\tau(m|x) - p^\tau(m|x') \right| = \left| \left( p^\tau(m|x) - \tau \right) - \left( p^\tau(m|x') - \tau \right) \right|$$

$$= \left| \left( p(m|x) - \tau \right) \left( 1 - \frac{\sum_{j \in \mathcal{M}_{p,x}^\tau} (\tau - p(j|x))}{\sum_{\ell \notin \mathcal{M}_{p,x}^\tau} (p(\ell|x) - \tau)} \right) - \left( p(m|x') - \tau \right) \left( 1 - \frac{\sum_{j \in \mathcal{M}_{p,x'}^\tau} (\tau - p(j|x'))}{\sum_{\ell \notin \mathcal{M}_{p,x'}^\tau} (p(\ell|x') - \tau)} \right) \right|$$

$$= \left| \left( p(m|x) - \tau \right) \cdot \frac{1 - M\tau}{\sum_{\ell \notin \mathcal{M}_{p,x}^\tau} (p(\ell|x) - \tau)} - \left( p(m|x') - \tau \right) \cdot \frac{1 - M\tau}{\sum_{\ell \notin \mathcal{M}_{p,x'}^\tau} (p(\ell|x') - \tau)} \right|$$

$$= \left| \left( p(m|x) - \tau \right) \cdot \frac{1 - M\tau}{\sum_{\ell \notin \mathcal{M}_{p,x}^\tau} (p(\ell|x) - \tau)} - \left( p(m|x') - \tau \right) \cdot \frac{1 - M\tau}{\sum_{\ell \notin \mathcal{M}_{p,x'}^\tau} (p(\ell|x) - \tau)} \right.$$

$$\left. + \left( p(m|x') - \tau \right) \cdot \frac{1 - M\tau}{\sum_{\ell \notin \mathcal{M}_{p,x}^\tau} (p(\ell|x) - \tau)} - \left( p(m|x') - \tau \right) \cdot \frac{1 - M\tau}{\sum_{\ell \notin \mathcal{M}_{p,x'}^\tau} (p(\ell|x') - \tau)} \right|$$

$$\le \left| p(m|x) - p(m|x') \right| + \left| \left( p(m|x') - \tau \right) \cdot \frac{\sum_{\ell \notin \mathcal{M}_{p,x'}^\tau} (p(\ell|x') - \tau) - \sum_{\ell \notin \mathcal{M}_{p,x}^\tau} (p(\ell|x) - \tau)}{\left( \sum_{\ell \notin \mathcal{M}_{p,x}^\tau} (p(\ell|x) - \tau) \right) \cdot \left( \sum_{\ell \notin \mathcal{M}_{p,x'}^\tau} (p(\ell|x') - \tau) \right)} \right|$$

$$\le c_\alpha \|x - x'\|_2^\alpha + \frac{\left| \sum_{\ell \notin \mathcal{M}_{p,x'}^\tau} (p(\ell|x') - \tau) - \sum_{\ell \notin \mathcal{M}_{p,x}^\tau} (p(\ell|x) - \tau) \right|}{\left( \sum_{\ell \notin \mathcal{M}_{p,x}^\tau} (p(\ell|x) - \tau) \right) \cdot \left( \sum_{\ell \notin \mathcal{M}_{p,x'}^\tau} (p(\ell|x') - \tau) \right)}. \tag{49}$$

By the triangle inequality, we have

$$\left| \sum_{\ell \notin \mathcal{M}_{p,x'}^\tau} (p(\ell|x') - \tau) - \sum_{\ell \notin \mathcal{M}_{p,x}^\tau} (p(\ell|x) - \tau) \right|$$

$$\le \sum_{\ell \notin (\mathcal{M}_{p,x}^\tau \cup \mathcal{M}_{p,x'}^\tau)} \left| p(\ell|x') - p(\ell|x) \right| + \sum_{\ell \notin \mathcal{M}_{p,x}^\tau, \ell \in \mathcal{M}_{p,x'}^\tau} \left| p(\ell|x') - \tau \right| + \sum_{\ell \in \mathcal{M}_{p,x}^\tau, \ell \notin \mathcal{M}_{p,x'}^\tau} \left| p(\ell|x) - \tau \right|. \tag{50}$$

For $\ell \in \mathcal{M}_{p,x'}^\tau$ and $\ell \notin \mathcal{M}_{p,x}^\tau$, by Assumption 3.1 *(i)*, we have

$$|p(\ell|x') - \tau| = \tau - p(\ell|x') \le p(\ell|x) - p(\ell|x') \le c_\alpha \|x - x'\|_2^\alpha.$$

Similarly, for $\ell \in \mathcal{M}_{p,x}^\tau$ and $\ell \notin \mathcal{M}_{p,x'}^\tau$, by Assumption 3.1 *(i)*, we have

$$|p(\ell|x) - \tau| = \tau - p(\ell|x) \le p(\ell|x') - p(\ell|x) \le c_\alpha \|x - x'\|_2^\alpha.$$

Therefore, combining Eq. (50) and Assumption 3.1 *(i)*, we obtain

$$\left| \sum_{\ell \notin \mathcal{M}^\tau_{p,x'}} \big( p(\ell|x') - \tau \big) - \sum_{\ell \notin \mathcal{M}^\tau_{p,x}} \big( p(\ell|x) - \tau \big) \right| \leq c_\alpha \sum_{\ell \notin \left( \mathcal{M}^\tau_{p,x'} \cup \mathcal{M}^\tau_{p,x} \right)} \|x - x'\|_2^\alpha \leq c_\alpha M \|x - x'\|_2^\alpha.$$

This together with Eq. (48) and Eq. (49) yields

$$
\begin{aligned}
\big| p^\tau(m|x) - p^\tau(m|x') \big| &\leq c_\alpha \|x - x'\|_2^\alpha + \frac{c_\alpha M \|x - x'\|_2^\alpha}{(1/2) \cdot (1/2)} \\
&= (4M + 1) c_\alpha \|x - x'\|_2^\alpha =: c_L \|x - x'\|_2^\alpha,
\end{aligned}
\tag{51}
$$

where $c_L := (4M + 1) c_\alpha$. Therefore, we finish the proof of the second inequality.

Now, if we assume that $|p(m|x) - p(m|x')| = c_L \|x - x'\|_2^\alpha$ holds for any $m \in [M]$, then similar to the analysis of Eq. (51), we can prove that

$$\big| p^\tau(m|x) - p^\tau(m|x') \big| \leq (4M + 1) c_\alpha \|x - x'\|_2^\alpha = (4M + 1) |p(m|x) - p(m|x')|,$$

which yields the first inequality. Thus, we finish the proof. $\qquad \square$

*Proof of Lemma D.7.* Lemma D.9 yields that for any $x, x' \in \mathcal{X}$, there holds

$$\big| p^\tau(m|x) - p^\tau(m|x') \big| \leq c_L \|x - x'\|_2^\alpha. \tag{52}$$

Moreover, Lemma D.8 together with Eq. (52) implies that if $p^\tau(m|x) \geq p^\tau(m|x')$, then

$$
\begin{aligned}
\big| g_m^\tau(x) - g_m^\tau(x') \big| &= \log p^\tau(m|x) - \log p^\tau(m|x') = \log \left( 1 + \frac{p^\tau(m|x) - p^\tau(m|x')}{p^\tau(m|x')} \right) \\
&\leq \frac{\log \big( 1 + p^\tau(m|x) - p^\tau(m|x') \big)}{p^\tau(m|x')} \leq \frac{\log \big( 1 + c_L \|x - x'\|_2^\alpha \big)}{p^\tau(m|x')}.
\end{aligned}
$$

Otherwise if $p^\tau(m|x) < p^\tau(m|x')$, then by using Lemma D.8 and Eq. (52) again, we get

$$
\begin{aligned}
\big| g_m^\tau(x) - g_m^\tau(x') \big| &= \log p^\tau(m|x') - \log p^\tau(m|x) = \log \left( 1 + \frac{p^\tau(m|x') - p^\tau(m|x)}{p^\tau(m|x)} \right) \\
&\leq \frac{\log \big( 1 + p^\tau(m|x') - p^\tau(m|x) \big)}{p^\tau(m|x)} \leq \frac{\log \big( 1 + c_L \|x - x'\|_2^\alpha \big)}{p^\tau(m|x)}.
\end{aligned}
$$

Therefore, we have

$$\big| g_m^\tau(x) - g_m^\tau(x') \big| \leq \frac{\log \big( 1 + c_L \|x - x'\|_2^\alpha \big)}{p^\tau(m|x) \wedge p^\tau(m|x')},$$

which yields the assertion. $\qquad \square$

With the aid of Lemma D.7, we are able to present the pointwise bound for the distance between $\widetilde{f}_m^\tau(x)$ and $f_m^{*\tau}(x)$, which is crucial to establish the approximation error bound of $p_{\widetilde{f}^\tau}(m|x)$.

**Proposition D.10.** *Let Assumption 3.1 hold. Furthermore, let $H$ be the Gaussian RKHS with the bandwidth parameter $\gamma \in \big( 0, 2^{-1/\alpha} \big)$. Moreover, let $\tau \in (0, 1/(2M))$ and let $p^\tau(m|x)$, $f_m^{*\tau}$ and $\widetilde{f}_m^\tau$ be defined as in Eq. (8), Eq. (26), and Eq. (45), respectively. Then, for any $m \in [M]$, there holds $\widetilde{f}_m^\tau \in H$ and*

$$\big| \widetilde{f}_m^\tau(x) - f_m^{*\tau}(x) \big| \leq c_1 (p^\tau(m|x)^{-1} + p^\tau(M|x)^{-1})(\gamma^\alpha \vee \tau^{-1} \gamma^{2\alpha}), \qquad m \in [M], \tag{53}$$

*where $c_1$ is a constant which will be specified in the proof.*

Proposition D.10 demonstrates that the approximation error to the target function depends on $p^\tau(m|x)^{-1}$ and $p^\tau(M|x)^{-1}$, highlighting that smaller CCP values are more challenging to approximate. Notably, the upper bound in Eq. (53) adaptively varies for different $x$ and is tighter than the universal bound that replaces both $p^\tau(m|x)^{-1}$ and $p^\tau(M|x)^{-1}$ in Eq. (53) by their upper bound $\tau^{-1}$, especially when $p(m|x)$ is not close to zero. This is crucial for us to obtain the minimal approximation error bound later by choosing a proper value for $\tau$.

*Proof of Proposition D.10.* Let the function $K : \mathbb{R}^d \to \mathbb{R}$ be defined as in Eq. (44). Then for any $x \in \mathcal{X}$ and $m \in [M]$, there holds

$$K * g_m^\tau(x) = \int_{\mathbb{R}^d} \left(\frac{2}{\gamma^2 \pi}\right)^{d/2} \exp\left(-\frac{2\|x - z\|_2^2}{\gamma^2}\right) g_m^\tau(z) \, dz$$

$$= \int_{\mathbb{R}^d} \left(\frac{2}{\gamma^2 \pi}\right)^{d/2} \exp\left(-\frac{2\|h\|_2^2}{\gamma^2}\right) g_m^\tau(x + h) \, dh.$$

Since the functions $g_m^\tau$, $m \in [M]$, have a compact support and are bounded, we have $g_m^\tau \in L_2(\mathbb{R}^d)$. This together with Proposition 4.46 in Steinwart & Christmann (2008) yields

$$K * g_m^\tau \in H. \tag{54}$$

Moreover, we have

$$g_m^\tau(x) = \int_{\mathbb{R}^d} \left(\frac{2}{\gamma^2 \pi}\right)^{d/2} \exp\left(-\frac{2\|h\|_2^2}{\gamma^2}\right) g_m^\tau(x) \, dh.$$

Then for any $x \in \mathcal{X}$, there holds

$$\left|K * g_m^\tau(x) - g_m^\tau(x)\right| = \left|\int_{\mathbb{R}^d} \left(\frac{2}{\gamma^2 \pi}\right)^{d/2} \exp\left(-\frac{2\|h\|_2^2}{\gamma^2}\right) \left(g_m^\tau(x + h) - g_m^\tau(x)\right) dh\right|$$

$$\leq \int_{\mathbb{R}^d} \left(\frac{2}{\gamma^2 \pi}\right)^{d/2} \exp\left(-\frac{2\|h\|_2^2}{\gamma^2}\right) \left|g_m^\tau(x + h) - g_m^\tau(x)\right| dh.$$

For $m \in [M]$, let $A_{m,x} := \{h \in \mathbb{R}^d : g_m^\tau(x + h) \geq g_m^\tau(x)\}$. Then by using Lemma D.7 and the fact that $\log(1 + x) \leq x$, $x > 0$, we get

$$\left|K * g_m^\tau(x) - g_m^\tau(x)\right| = \int_{A_{m,x}} \left(\frac{2}{\gamma^2 \pi}\right)^{d/2} \exp\left(-\frac{2\|h\|_2^2}{\gamma^2}\right) \left(g_m^\tau(x + h) - g_m^\tau(x)\right) dh$$

$$+ \int_{\mathbb{R}^d \backslash A_{m,x}} \left(\frac{2}{\gamma^2 \pi}\right)^{d/2} \exp\left(-\frac{2\|h\|_2^2}{\gamma^2}\right) \left(g_m^\tau(x) - g_m^\tau(x + h)\right) dh$$

$$\leq \int_{A_{m,x}} \left(\frac{2}{\gamma^2 \pi}\right)^{d/2} \exp\left(-\frac{2\|h\|_2^2}{\gamma^2}\right) \frac{\log\left(1 + c_L \|h\|_2^\alpha\right)}{p^\tau(m|x)} dh$$

$$+ \int_{\mathbb{R}^d \backslash A_{m,x}} \left(\frac{2}{\gamma^2 \pi}\right)^{d/2} \exp\left(-\frac{2\|h\|_2^2}{\gamma^2}\right) \frac{\log\left(1 + c_L \|h\|_2^\alpha\right)}{p^\tau(m|x + h)} dh$$

$$\leq \int_{\mathbb{R}^d} \left(\frac{2}{\gamma^2 \pi}\right)^{d/2} \exp\left(-\frac{2\|h\|_2^2}{\gamma^2}\right) \frac{c_L \|h\|_2^\alpha}{p^\tau(m|x)} dh$$

$$+ \int_{\mathbb{R}^d} \left(\frac{2}{\gamma^2 \pi}\right)^{d/2} \exp\left(-\frac{2\|h\|_2^2}{\gamma^2}\right) \frac{c_L \|h\|_2^\alpha}{p^\tau(m|x + h)} dh =: (I) + (II). \tag{55}$$

For the first term $(I)$ in Eq. (55), using the rotation invariance of $x \mapsto \exp(-2\|x\|_2^2/\gamma^2)$ and $\Gamma(1 + c) = c\Gamma(c)$, $c > 0$, we get

$$(I) = \frac{c_L}{p^\tau(m|x)} \left(\frac{\gamma}{\sqrt{2}}\right)^\alpha \int_{\mathbb{R}^d} \left(\frac{1}{\pi}\right)^{d/2} \exp(-\|h\|_2^2)\|h\|_2^\alpha \, dh$$

$$= \frac{c_L}{p^\tau(m|x)} \left(\frac{\gamma}{\sqrt{2}}\right)^\alpha \frac{2}{\Gamma(d/2)} \int_0^\infty e^{-r^2} r^{\alpha + d - 1} dr$$

$$= \frac{c_L}{p^\tau(m|x)} \Gamma(d/2)^{-1} \Gamma\left(\frac{d + \alpha}{2}\right) 2^{-\alpha/2} \gamma^\alpha. \tag{56}$$

Using Eq. (52) and $p^\tau(m|x + h) \geq \tau$, for any $x \in \mathcal{X}$ and $h \in \mathbb{R}^d$, we get

$$p^\tau(m|x + h)^{-1} \leq p^\tau(m|x)^{-1} + \left|p^\tau(m|x + h)^{-1} - p^\tau(m|x)^{-1}\right|$$

$$\le p^{\tau}(m|x)^{-1} + \frac{\left|p^{\tau}(m|x+h) - p^{\tau}(m|x)\right|}{p^{\tau}(m|x+h)p^{\tau}(m|x)} \le p^{\tau}(m|x)^{-1} + \frac{c_L\|h\|_2^{\alpha}}{\tau p^{\tau}(m|x)}. \quad (57)$$

For the second term $(II)$ in Eq. (55), using Eq. (56) and Eq. (57), we obtain

$$(II) \le c_L \int_{\mathbb{R}^d} \left(p^{\tau}(m|x)^{-1} + \frac{c_L\|h\|_2^{\alpha}}{\tau p^{\tau}(m|x)}\right) \left(\frac{2}{\gamma^2\pi}\right)^{d/2} \exp\left(-\frac{2\|h\|_2^2}{\gamma^2}\right)\|h\|_2^{\alpha} \, dh$$

$$= \int_{\mathbb{R}^d} \frac{c_L}{p^{\tau}(m|x)} \left(\frac{2}{\gamma^2\pi}\right)^{d/2} \exp\left(-\frac{2\|h\|_2^2}{\gamma^2}\right)\|h\|_2^{\alpha} \, dh$$

$$+ \int_{\mathbb{R}^d} \frac{c_L^2}{\tau p^{\tau}(m|x)} \left(\frac{2}{\gamma^2\pi}\right)^{d/2} \exp\left(-\frac{2\|h\|_2^2}{\gamma^2}\right)\|h\|_2^{2\alpha} \, dh$$

$$= \frac{c_L}{p^{\tau}(m|x)} \cdot \Gamma(d/2)^{-1}\Gamma\left(\frac{d+\alpha}{2}\right)2^{-\alpha/2}\gamma^{\alpha}$$

$$+ \frac{c_L^2}{\tau p^{\tau}(m|x)} \left(\frac{\gamma}{\sqrt{2}}\right)^{2\alpha} \int_{\mathbb{R}^d} \pi^{-d/2} \exp(-\|h\|_2^2)\|h\|_2^{2\alpha} \, dh.$$

The rotation invariance of $x \mapsto \exp(-2\|x\|_2^2/\gamma^2)$ together with $\Gamma(1+c) = c\Gamma(c)$, $c > 0$, yields

$$\int_{\mathbb{R}^d} \pi^{-d/2} \exp(-\|h\|_2^2)\|h\|_2^{2\alpha} \, dh = \int_0^{\infty} \frac{2}{\Gamma(d/2)} \exp(-r^2)r^{2\alpha+d-1} \, dr$$

$$= \Gamma(d/2)^{-1} \int_0^{\infty} \exp(-r)r^{\alpha+d/2-1} \, dr = \frac{\Gamma(\alpha+d/2)}{\Gamma(d/2)}$$

and consequently we have

$$(II) \le \frac{c_L}{p^{\tau}(m|x)}\Gamma(d/2)^{-1}\Gamma\left(\frac{d+\alpha}{2}\right)2^{-\alpha/2}\gamma^{\alpha} + \frac{c_L^2}{\tau p^{\tau}(m|x)}\left(\frac{\gamma}{\sqrt{2}}\right)^{2\alpha}\frac{\Gamma(\alpha+d/2)}{\Gamma(d/2)}. \quad (58)$$

Combining Eq. (56), Eq. (58) and Eq. (55), we obtain

$$\left|K * g_m^{\tau}(x) - g_m^{\tau}(x)\right|$$

$$\le \frac{c_L}{p^{\tau}(m|x)}\Gamma(d/2)^{-1}\Gamma\left(\frac{d+\alpha}{2}\right)2^{1-\alpha/2}\gamma^{\alpha} + \frac{c_L^2}{\tau p^{\tau}(m|x)}\frac{\Gamma(\alpha+d/2)}{\Gamma(d/2)}\left(\frac{\gamma}{\sqrt{2}}\right)^{2\alpha}$$

$$\le c_1(p^{\tau}(m|x))^{-1}(\gamma^{\alpha} \vee \tau^{-1}\gamma^{2\alpha}), \quad (59)$$

where $c_1 := 2c_L\Gamma(d/2)^{-1}\Gamma((d+\alpha)/2) + c_L^2\Gamma(\alpha+d/2)\Gamma(d/2)^{-1}$. By the definition of $\widetilde{f}_m^{\tau}$, we have

$$\widetilde{f}_m^{\tau} = K * g_m^{\tau} - K * g_M^{\tau} = K * f_m^{*\tau}. \quad (60)$$

Then by Eq. (54) and the linearity of the RKHS, we have $\widetilde{f}_m^{\tau} \in H$. Using the triangle inequality Eq. (59), we obtain that for any $x \in \mathcal{X}$, there holds

$$\left|\widetilde{f}_m^{\tau}(x) - f_m^{*\tau}(x)\right| = \left|(K * g_m^{\tau} - K * g_M^{\tau}) - (g_m^{\tau} - g_M^{\tau})\right|$$

$$\le \left|K * g_m^{\tau}(x) - g_m^{\tau}(x)\right| + \left|K * g_M^{\tau}(x) - g_M^{\tau}(x)\right|$$

$$\le c_1\left(p^{\tau}(m|x)^{-1} + p^{\tau}(M|x)^{-1}\right)(\gamma^{\alpha} \vee \tau^{-1}\gamma^{2\alpha}),$$

which finishes the proof. $\qquad\square$

Based on the upper bound of the pointwise distance between $\widetilde{f}_m^{\tau}(x)$ and $f_m^{*\tau}(x)$, we are able to derive the approximation error bound for the CCP estimator $p_{\widetilde{f}^{\tau}}(\cdot|x)$ in Proposition D.11.

**Proposition D.11.** *Let Assumption 3.1 hold. Let $\gamma \in \left(0, 2^{-1/\alpha}\right)$ be the bandwidth of the Gaussian kernel, $\widetilde{f}^{\tau} := (\widetilde{f}_m^{\tau})_{m\in[M]}$ with $\tau \in [0., 1/(2M)]$ be as in Eq. (45) and $c_1 > 1$ the constant as in Proposition D.10. Then, its induced estimator $p_{\widetilde{f}^{\tau}}(m|x)$ in Eq. (7) satisfies*

*(i) $p_{\widetilde{f}^{\tau}}(m|x) \ge \tau/M$;*

*(ii)* $|p(m|x) - p_{\widetilde{f}^\tau}(m|x)| \leq 4M \exp\left(6c_1(1 \vee (\gamma^\alpha/\tau)^2)\right)\tau$. *This upper bound achieves its minimum at $\tau := \gamma^\alpha$ and the resulting order is of $\gamma^\alpha$.*

*Proof of Proposition D.11.* Using the definition of $p_{\widetilde{f}^\tau}(m|x)$ and Eq. (47), we get

$$p_{\widetilde{f}^\tau}(m|x) := \frac{\exp(\widetilde{f}_m^\tau)}{\sum_{j=1}^M \exp(\widetilde{f}_m^\tau)} = \frac{\exp(K * f_m^{*\tau})}{\sum_{j=1}^M \exp(K * f_m^{*\tau})} = \frac{\exp(K * g_m^\tau)}{\sum_{j=1}^M \exp(K * g_m^\tau)}.$$

Using Eq. (46) and $p^\tau(m|x) \in [\tau, (1-(M-1)\tau)]$, we get $g_m^\tau \in [\log \tau, \log(1-(M-1)\tau)]$ and thus $K * g_m^\tau \in [\log \tau, \log(1-(M-1)\tau)]$. Consequently we have

$$p_{\widetilde{f}^\tau}(m|x) \geq \frac{\tau}{\sum_{j=1}^M (1-(M-1)\tau)} \geq \frac{\tau}{M},$$

which proves the first assertion *(i)*.

Using the definitions of $p_{\widetilde{f}^\tau}(m|x)$ and $f_m^{*\tau}$ in Eq. (26), we get

$$
\begin{aligned}
p^\tau(m|\cdot) - p_{\widetilde{f}^\tau}(m|\cdot) &= \frac{\exp(f_m^{*\tau})}{\sum_{j=1}^M \exp(f_j^{*\tau})} - \frac{\exp(\widetilde{f}_m^\tau)}{\sum_{j=1}^M \exp(\widetilde{f}_j^\tau)} \\
&= \frac{\exp(f_m^{*\tau}) \cdot \sum_{j=1}^M \exp(\widetilde{f}_j^\tau) - \exp(\widetilde{f}_m^\tau) \cdot \sum_{j=1}^M \exp(f_j^{*\tau})}{\sum_{j=1}^M \exp(f_j^{*\tau}) \cdot \sum_{j=1}^M \exp(\widetilde{f}_j^\tau)} \\
&= \frac{\exp(f_m^{*\tau}) \cdot \sum_{j=1}^M \exp(\widetilde{f}_j^\tau) - \exp(f_j^{*\tau}) + (\exp(f_j^{*\tau}) - \exp(\widetilde{f}_m^\tau)) \cdot \sum_{j=1}^M \exp(f_j^{*\tau})}{\sum_{j=1}^M \exp(f_j^{*\tau}) \cdot \sum_{j=1}^M \exp(\widetilde{f}_j^\tau)} \\
&= p_{\widetilde{f}^\tau}(M|\cdot)\left(\sum_{j=1}^M \left(e^{\widetilde{f}_j^\tau} - e^{f_j^{*\tau}}\right)p^\tau(m|\cdot) + e^{f_m^{*\tau}} - e^{\widetilde{f}_m^\tau}\right) \\
&= p_{\widetilde{f}^\tau}(M|\cdot)\left(\sum_{j \neq m} \left(e^{\widetilde{f}_j^\tau} - e^{f_j^{*\tau}}\right)p^\tau(m|\cdot) + \left(e^{f_m^{*\tau}} - e^{\widetilde{f}_m^\tau}\right)\left(1 - p^\tau(m|\cdot)\right)\right) \\
&= p_{\widetilde{f}^\tau}(M|\cdot)\left(\sum_{j \neq m} \left(e^{\widetilde{f}_j^\tau - f_j^{*\tau}} - 1\right)e^{f_j^{*\tau}}p^\tau(m|\cdot) + \left(1 - e^{\widetilde{f}_m^\tau - f_m^{*\tau}}\right)e^{f_m^{*\tau}}\left(1 - p^\tau(m|\cdot)\right)\right) \\
&= p_{\widetilde{f}^\tau}(M|\cdot)\left(\sum_{j \neq m} \left(e^{\widetilde{f}_j^\tau - f_j^{*\tau}} - 1\right)e^{f_m^{*\tau}}p^\tau(j|\cdot) + \left(1 - e^{\widetilde{f}_m^\tau - f_m^{*\tau}}\right)e^{f_m^{*\tau}}\left(1 - p^\tau(m|\cdot)\right)\right) \\
&= p_{\widetilde{f}^\tau}(M|\cdot)\left(\sum_{j \neq m} (\exp(\widetilde{f}_j^\tau - f_j^{*\tau}) - \exp(\widetilde{f}_m^\tau - f_m^{*\tau}))\exp(f_m^{*\tau})p^\tau(j|\cdot)\right) \\
&= \frac{p_{\widetilde{f}^\tau}(M|\cdot)}{p^\tau(M|\cdot)}\left(\sum_{j \neq m} (\exp(\widetilde{f}_j^\tau - f_j^{*\tau}) - \exp(\widetilde{f}_m^\tau - f_m^{*\tau}))p^\tau(m|\cdot)p^\tau(j|\cdot)\right) \\
&= \frac{p_{\widetilde{f}^\tau}(M|\cdot)}{p^\tau(M|\cdot)}\left(\sum_{j \neq m} (\exp(\widetilde{f}_j^\tau - f_j^{*\tau} - (\widetilde{f}_m^\tau - f_m^{*\tau})) - 1)\exp(\widetilde{f}_m^\tau - f_m^{*\tau})p^\tau(m|\cdot)p^\tau(j|\cdot)\right).
\end{aligned}
$$

By the triangle inequality, we have

$$|p^\tau(m|\cdot) - p_{\widetilde{f}^\tau}(m|\cdot)|$$
$$\leq \frac{p_{\widetilde{f}^\tau}(M|\cdot)}{p^\tau(M|\cdot)}\left(\sum_{j \neq m} \left|\exp(\widetilde{f}_j^\tau - f_j^{*\tau} - (\widetilde{f}_m^\tau - f_m^{*\tau})) - 1\right|p^\tau(m|\cdot)p^\tau(j|\cdot)\right)\exp(\widetilde{f}_m^\tau - f_m^{*\tau}). \quad (61)$$

Using Eq. (47) and $\widetilde{f}_m^\tau = K * f_m^{*\tau}$, we get

$$\widetilde{f}_j^\tau - f_j^{*\tau} - (\widetilde{f}_m^\tau - f_m^{*\tau}) = K * g_j^\tau - g_j^\tau - (K * g_m^\tau - g_m^\tau). \quad (62)$$

Then by using the triangle inequality and Eq. (59), for any $m \in [M]$, we obtain

$$|\widetilde{f}_j^\tau - f_j^{*\tau} - (\widetilde{f}_m^\tau - f_m^{*\tau})| \le |K * g_j^\tau - g_j^\tau| + |K * g_m^\tau - g_m^\tau|$$

$$\le c_1(p^\tau(j|\cdot)^{-1} + p^\tau(m|\cdot)^{-1})(\gamma^\alpha \vee \tau^{-1}\gamma^{2\alpha}) \tag{63}$$

$$\le 2c_1(\tau^{-1}\gamma^\alpha \vee \tau^{-2}\gamma^{2\alpha}) =: 2c_{\tau,\gamma}, \tag{64}$$

where we denote $c_{\tau,\gamma} := c_1(\tau^{-1}\gamma^\alpha \vee \tau^{-2}\gamma^{2\alpha})$. For any function $h_1$ and $h_2$ satisfying $|h_1 - h_2| \le 2c_{\tau,\gamma}$, if $h_1 > h_2$, then by using the Lagrange mean value theorem, there exists $p \in (0, h_1 - h_2)$ such that

$$|\exp(h_1 - h_2) - 1| = \exp(h_1 - h_2) - 1 = e^p(h_1 - h_2)$$

$$\le \exp(h_1 - h_2) \cdot (h_1 - h_2) \le e^{2c_{\tau,\gamma}}(h_1 - h_2). \tag{65}$$

Otherwise if $h_1 < h_2$, then by using the Lagrange mean value theorem once again, there exists $p \in (h_1 - h_2, 0)$ such that

$$|\exp(h_1 - h_2) - 1| = 1 - \exp(h_1 - h_2) = e^p(h_2 - h_1) \le h_2 - h_1. \tag{66}$$

Combining Eq. (65) and Eq. (66), we find

$$|\exp(h_1 - h_2) - 1| \le e^{2c_{\tau,\gamma}}|h_1 - h_2|. \tag{67}$$

Applying Eq. (67) with $h_1 := \widetilde{f}_j^\tau - f_j^{*\tau}$ and $h_2 := \widetilde{f}_m^\tau - f_m^{*\tau}$, we obtain

$$\left|\exp(\widetilde{f}_j^\tau - f_j^{*\tau} - (\widetilde{f}_m^\tau - f_m^{*\tau})) - 1\right| \le e^{2c_{\tau,\gamma}}|\widetilde{f}_j^\tau - f_j^{*\tau} - (\widetilde{f}_m^\tau - f_m^{*\tau})|$$

$$\le e^{2c_{\tau,\gamma}}c_{\tau,\gamma}\tau(p^\tau(j|\cdot)^{-1} + p^\tau(m|\cdot)^{-1}), \tag{68}$$

where the last inequality is due to Eq. (63). Similar to the analysis in Eq. (62) and Eq. (64), we have

$$|\widetilde{f}_m^\tau - f_m^{*\tau}| = |K * g_m^\tau - K * g_M^\tau - (g_m^\tau - g_M^\tau)| \le |K * g_m^\tau - g_m^\tau| + |K * g_M^\tau - g_M^\tau| \le 2c_{\tau,\gamma}.$$

Applying Eq. (67) once again with $h_1 := \widetilde{f}_m^\tau$ and $h_2 := f_m^{*\tau}$, we obtain

$$|\exp(\widetilde{f}_m^\tau - f_m^{*\tau}) - 1| \le e^{2c_1(\tau^{-1}\gamma^\alpha \vee \tau^{-2}\gamma^{2\alpha})}|\widetilde{f}_m^\tau - f_m^{*\tau}| \le 2c_{\tau,\gamma}e^{2c_{\tau,\gamma}},$$

which implies

$$\exp(\widetilde{f}_m^\tau - f_m^{*\tau}) \le 1 + 2c_{\tau,\gamma}e^{2c_{\tau,\gamma}}. \tag{69}$$

Combining Eq. (61), Eq. (68) and Eq. (69), we obtain

$$|p^\tau(m|x) - p_{\widetilde{f}^\tau}(m|x)|$$

$$\le (1 + 2c_{\tau,\gamma}e^{2c_{\tau,\gamma}})\frac{p_{\widetilde{f}^\tau}(M|x)}{p^\tau(M|x)} \sum_{j \ne m} \left(c_{\tau,\gamma}e^{2c_{\tau,\gamma}}\tau(p^\tau(j|x)^{-1} + p^\tau(m|x)^{-1})\right)p^\tau(m|x)p^\tau(j|x)$$

$$= (1 + 2c_{\tau,\gamma}e^{2c_{\tau,\gamma}})c_{\tau,\gamma}e^{2c_{\tau,\gamma}}\tau \sum_{j \ne m}\left(p^\tau(m|x) + p^\tau(j|x)\right)$$

$$\le M(1 + 2c_{\tau,\gamma}e^{2c_{\tau,\gamma}})c_{\tau,\gamma}e^{2c_{\tau,\gamma}}\tau.$$

By the definition of $p^\tau(m|x)$ in Eq. (8), we have $|p^\tau(m|x) - p(m|x)| \le M\tau$. Using the triangle inequality, we then get

$$|p_{\widetilde{f}^\tau}(m|x) - p(m|x)| \le |p_{\widetilde{f}^\tau}(m|x) - p^\tau(m|x)| + |p^\tau(m|x) - p(m|x)|$$

$$\le M(1 + 2c_{\tau,\gamma}e^{2c_{\tau,\gamma}})c_{\tau,\gamma}e^{2c_{\tau,\gamma}}\tau + M\tau$$

$$= M\left(1 + (1 + 2c_{\tau,\gamma}e^{2c_{\tau,\gamma}})c_{\tau,\gamma}e^{2c_{\tau,\gamma}}\right)\tau$$

$$\le 4M\left(1 \vee c_{\tau,\gamma}^2 e^{4c_{\tau,\gamma}}\right)\tau \le 4M\left(1 \vee c_{\tau,\gamma}\right)^2 \exp\left(4(1 \vee c_{\tau,\gamma})\right)\tau$$

$$\le 4M\exp\left(6(1 \vee c_{\tau,\gamma})\right)\tau \le 4M\exp\left(6c_1(1 \vee (\gamma^\alpha/\tau)^2)\right)\tau,$$

where the second last inequality follows from $e^{2x} \ge x^2$, $x \ge 1$. This finishes the proof. $\qquad\square$

Before we present the proof of Proposition 4.2, we need the following proposition that gives an upper bound of the excess CE risk for any estimator $p_f(\cdot|x)$.

**Proposition D.12.** *Let $P$ be the probability distribution on $\mathcal{X} \times \mathcal{Y}$. Moreover, let $f := (f_m)_{m \in [M]}$ with $f_m : \mathcal{X} \to \mathbb{R}$ be the score function and its corresponding conditional probability estimator be $p_f(m|\cdot)$ as in Eq. (7). Then we have*

$$\mathcal{R}_{L_{\mathrm{CE}},P}(p_f(\cdot|x)) - \mathcal{R}^*_{L_{\mathrm{CE}},P} \leq \mathbb{E}_{X \sim p} \sum_{m=1}^{M} \frac{(p(m|X) - p_f(m|X))^2}{p_f(m|X)}.$$

*Proof of Proposition D.12.* By the definition of $L_{\mathrm{CE}}$ and $p_f(m|\cdot)$, we have

$$\mathcal{R}_{L_{\mathrm{CE}},P}(p_f(\cdot|x)) = - \int_{\mathcal{X}} \sum_{m=1}^{M} p(m|x) \log p_f(m|x) \, dP_X(x).$$

Then we have $\mathcal{R}^*_{L_{\mathrm{CE}},P} = - \int_{\mathcal{X}} \sum_{m=1}^{M} p(m|x) \log p(m|x) \, dP_X(x)$. Consequently, we obtain

$$\mathcal{R}_{L_{\mathrm{CE}},P}(p_f(\cdot|x)) - \mathcal{R}^*_{L_{\mathrm{CE}},P} = \mathbb{E}_{X \sim p} \sum_{m=1}^{M} p(m|X) \log \frac{p(m|X)}{p_f(m|X)}.$$

Using Lemma 2.7 in Tsybakov (2008), we get

$$\mathbb{E}_{X \sim p} \sum_{m=1}^{M} p(m|X) \log \frac{p(m|X)}{p_f(m|X)} \leq \mathbb{E}_{X \sim p} \sum_{m=1}^{M} \frac{(p(m|X) - p_f(m|X))^2}{p_f(m|X)},$$

which finishes the proof. $\qquad\square$

**Proposition D.13.** *Let the probability distribution $P$ satisfy Assumption 3.1 (ii). Then for any $s \in (0, 1]$ and any $\beta \geq 0$, we have*

$$\int_{\{p(m|x) \geq s\}} \frac{1}{p(m|x)} \, dP_X(x) \leq \begin{cases} c_\beta (1 - \beta)^{-1} s^{\beta-1}, & \text{for } 0 \leq \beta < 1; \\ c_\beta (1 + \log s^{-1}), & \text{for } \beta \geq 1. \end{cases}$$

*Proof of Proposition D.13.* Since $p(m|x)$ is a probability, we have $p(m|x) \leq 1$ and consequently $C \geq 1$. For any nonnegative function $h$ and random variable $Z \sim P_Z$, there holds $\int h(Z) \, dP(Z) = E[h(Z)] = \int_0^\infty P_Z(h_Z \geq u) \, du$. Hence we have

$$\int_{\{p(m|x) \geq s\}} \frac{1}{p(m|x)} \, dP_X(x) = \int_0^\infty P_X \left( \frac{\mathbf{1}\{p(m|x) \geq s\}}{p(m|x)} \geq u \right) du$$

$$\leq \int_0^{1/s} P_X(p(m|x) \leq 1/u) \, du,$$

where the last inequality follows from the fact that $\mathbf{1}\{p(m|x) \geq s\}/p(m|x) \geq u$ implies $s < p(m|x) \leq 1/u$ and $u \leq 1/s$. By Assumption 3.1 *(ii)* with $0 < \beta < 1$, we have

$$\int_0^{1/s} P_X\big(p(m|x) \leq 1/u\big) \, du \leq c_\beta \int_0^{1/s} u^{-\beta} \, du = \frac{c_\beta s^{\beta-1}}{1 - \beta}. \qquad (70)$$

Since $P_X(p(m|x) \leq t) \leq 1$, we have for all $t \in [0, 1]$,

$$\int_0^{1/s} P_X(p(m|x) \leq 1/u) \, du \leq \int_0^{1/s} 1 \, du = 1/s \leq c_\beta s^{-1}.$$

Therefore, Eq. (70) also holds if $\beta = 0$ and thus we obtain the first assertion.

For $\beta > 1$, we have $P_X(p(x|k) \leq t) \leq c_\beta t^\beta \leq c_\beta t$, $t \in [0, 1]$. If $C \leq s^{-1}$, then we have

$$\int_0^{1/s} P_X(p(m|x) \leq 1/u) \, du = \int_0^{c_\beta} P_X(p(m|x) \leq 1/u) \, du + \int_{c_\beta}^{1/s} P_X(p(m|x) \leq 1/u) \, du$$

$$\leq \int_0^{c_\beta} 1 \, du + \int_{c_\beta}^{1/s} c_\beta/u \, du = c_\beta + c_\beta(\log s^{-1} - \log c_\beta) \leq c_\beta(1 + \log s^{-1}),$$

which finishes the proof. $\qquad\square$

Now, with all the above preparations, we are able to establish the excess CE risk of the approximator $p_{\widetilde{f}^\tau}^t(y|x)$.

*Proof of Proposition 4.2.* Let $c_1$ be the constant as in Propositions D.10. By Proposition D.11, we have

$$
\begin{aligned}
|p(m|x) - p_{\widetilde{f}^\tau}^t(m|x)| &\leq |p(m|x) - p_{\widetilde{f}^\tau}(m|x)| + |p_{\widetilde{f}^\tau}(m|x) - p_{\widetilde{f}^\tau}^t(m|x)| \\
&\leq 4M \exp\left(6c_1(1 \vee (\gamma^\alpha/\tau)^2)\right)\tau + t \\
&\leq 4Me^{6c_1}\left(\tau \vee \tau \exp\left(6c_1((\gamma^\alpha/\tau)^2)\right) \vee t\right) =: a_1,
\end{aligned}
\tag{71}
$$

and

$$
p_{\widetilde{f}^\tau}^t(m|x) \geq \tau/M \vee t \geq (\tau \vee t)/M =: a_2.
\tag{72}
$$

By Proposition D.12, for the truncated conditional probability estimator $p_{\widetilde{f}^\tau}^t$, we have

$$
\begin{aligned}
\mathcal{R}_{L_{\mathrm{CE}},P}(p_{\widetilde{f}^\tau}^t(y|x)) - \mathcal{R}_{L_{\mathrm{CE}},P}^* &\leq \mathbb{E}_{x\sim p} \sum_{m=1}^M \frac{(p(m|x) - p_{\widetilde{f}^\tau}^t(m|x))^2}{p_{\widetilde{f}^\tau}^t(m|x)} \\
&= \sum_{m=1}^M \mathbb{E}_{x\sim p}\left(\frac{(p(m|x) - p_{\widetilde{f}^\tau}^t(m|x))^2}{p_{\widetilde{f}^\tau}^t(m|x)} \cdot \mathbf{1}\{p(m|x) \leq a_1 + a_2\}\right) \\
&\quad + \sum_{m=1}^M \mathbb{E}_{x\sim p}\left(\frac{(p(m|x) - p_{\widetilde{f}^\tau}^t(m|x))^2}{p_{\widetilde{f}^\tau}^t(m|x)} \cdot \mathbf{1}\{p(m|x) \geq a_1 + a_2\}\right) =: (I) + (II).
\end{aligned}
\tag{73}
$$

Thus for the first term $(I)$ in Eq. (73), by Eq. (71), Eq. (72) and Assumption 3.1 *(ii)*, we have

$$
(I) \leq (a_1^2/a_2) \cdot P_X\left(p(m|x) \leq (a_1 + a_2)\right) \leq (a_1^2/a_2)c_\beta(a_1 + a_2)^\beta.
\tag{74}
$$

For the second term $(II)$ in Eq. (73), if $p(m|x) \geq a_1 + a_2$, then we have $a_1 \leq (a_1/(a_1+a_2))p(m|x)$ and thus $p(m|x) - a_1 \geq p(m|x) - (a_1/(a_1 + a_2))p(m|x) = (a_2/(a_1 + a_2))p(m|x)$. This together with Eq. (71) yields

$$
p_{\widetilde{f}^\tau}^t(m|x) \geq p(m|x) - |p(m|x) - p_{\widetilde{f}^\tau}^t(m|x)| \geq p(m|x) - a_1 \geq (a_2/(a_1 + a_2))p(m|x).
$$

This together with Eq. (71), Eq. (72) and Proposition D.13 yields

$$
\begin{aligned}
(II) &\leq (a_1^2(a_1 + a_2)/a_2) \cdot \int_{\{p(m|x)\geq(a_1+a_2)\}} \frac{1}{p(m|x)} \, dP_X(x) \\
&\leq (a_1^2(a_1 + a_2)/a_2)c_\beta\left(\frac{(a_1 + a_2)^{\beta-1}}{1 - \beta} \cdot \mathbf{1}_{\{\beta<1\}} + \left(1 + \frac{1}{\log(a_1 + a_2)}\right)\mathbf{1}_{\{\beta\geq1\}}\right).
\end{aligned}
\tag{75}
$$

Combining Eq. (73), Eq. (74), Eq. (75) and $a_1 \geq a_2$, we obtain

$$
\begin{aligned}
\mathcal{R}_{L_{\mathrm{CE}},P}(p_{\widetilde{f}^\tau}^t(y|x)) - \mathcal{R}_{L_{\mathrm{CE}},P}^* &\lesssim a_1^{2+\beta}/a_2 + \log(a_2)^{-1}(a_1^3/a_2)\mathbf{1}_{\{\beta\geq1\}} \lesssim \log(a_2)^{-1}(a_1^{2+\beta\wedge1}/a_2) \\
&\lesssim \log(\tau \vee t)^{-1}\left(\tau \vee \tau \exp\left(6c_1((\gamma^\alpha/\tau)^2)\right) \vee t\right)^{2+\beta\wedge1}/(\tau \vee t).
\end{aligned}
$$

Notice that the right-hand side of the above inequality attains the minimal order at $\tau := \gamma^\alpha$. Therefore, we choose $f_0 := \widetilde{f}^\tau$ with $\tau = \gamma^\alpha$ as the approximator to get

$$
\mathcal{R}_{L_{\mathrm{CE}},P}(p_{\widetilde{f}^\tau}^t(y|x)) - \mathcal{R}_{L_{\mathrm{CE}},P}^* \lesssim \log(\gamma^\alpha)^{-1}\left(\gamma^\alpha \vee t\right)^{1+\beta\wedge1}.
\tag{76}
$$

In addition, the definition of $\widetilde{f}_m^t \in H$ in Eq. (60) together with Proposition 4.46 in Steinwart & Christmann (2008) yields

$$
\begin{aligned}
\|f_0\|_H^2 &\leq \pi^{-d/2}\gamma^{-d}\|f_m^{*\tau}\mathbf{1}_\mathcal{X}\|_{L_2}^2 \leq \pi^{-d/2}\gamma^{-d}\log^2\left((1 - \tau)/\tau\right) \\
&\leq \pi^{-d/2}\gamma^{-d}\log^2(1/\tau) \lesssim \gamma^{-d}\log^2(\gamma^{-\alpha}).
\end{aligned}
$$

This together with Eq. (76) proves the assertion. □

### D.1.3 Proofs of Minimax Convergence Rates for KLR

**Theorem D.14.** *Let Assumption 3.1 hold and let the CCP estimator $\widehat{p}(y|x)$ be defined as in Eq. (10). If we choose $\lambda \asymp n^{-1}$, $\gamma \asymp n^{-1/((1+\beta\wedge1)\alpha+d)}$, and $t \asymp n^{-\theta}$ with $\theta \geq 1$, then there exists some $N \in \mathbb{N}$ such that for any $n \geq N$ and for any $\xi \in (0, 1/2)$, there holds*

$$\mathcal{R}_{L_{\mathrm{CE}},P}(\widehat{p}(y|x)) - \mathcal{R}^*_{L_{\mathrm{CE}},P} \lesssim n^{-\frac{(1+\beta\wedge1)\alpha}{(1+\beta\wedge1)\alpha+d}+\xi}$$

*with probability $P^n$ at least $1 - 1/n$.*

*Proof of Theorem D.14.* By combining Proposition 4.2 and Proposition 4.1, we obtain

$$\lambda\|f_D\|_H^2 + \mathcal{R}_{L_{\mathrm{CE}},P}(\widehat{p}(y|x)) - \mathcal{R}^*_{L_{\mathrm{CE}},P}$$
$$\lesssim \lambda\gamma^{-d}\log^2(\gamma^{-\alpha}) + \log(\gamma^\alpha \vee t)^{-1} \cdot (\gamma^\alpha \vee t)^{1+\beta\wedge1} + (-\log t) \cdot (t + \lambda^{-\xi}\gamma^{-d}n^{-1} + \zeta/n)$$

with probability at least $1 - 4e^{-\zeta}$. In order to minimize the right-hand side with respect to $\gamma$, $t$ and $\lambda$, we choose $\lambda = n^{-1}$, $\zeta = 4\log n$, $\gamma = n^{-1/((1+\beta\wedge1)\alpha+d)}$ and $t = n^{-\theta}$ with $\theta \geq 1$. Then we obtain

$$\lambda\|f_D\|_H^2 + \mathcal{R}_{L_{\mathrm{CE}},P}(\widehat{p}(y|x)) - \mathcal{R}^*_{L_{\mathrm{CE}},P}$$
$$\lesssim n^{-\frac{(1+\beta\wedge1)\alpha}{(1+\beta\wedge1)\alpha+d}}\log^2 n + \log n\Big(n^{-1} + n^{-\frac{(1+\beta\wedge1)\alpha}{(1+\beta\wedge1)\alpha+d}}n^\xi + n^{-1}\log n\Big).$$

For any $n \geq N$, there exists an $N \in \mathbb{N}$ such that $\log^2 n \leq n^\xi$. Thus we get

$$\lambda\|f_D\|_H^2 + \mathcal{R}_{L_{\mathrm{CE}},P}(\widehat{p}(y|x)) - \mathcal{R}^*_{L_{\mathrm{CE}},P} \lesssim n^{-\frac{(1+\beta\wedge1)\alpha}{(1+\beta\wedge1)\alpha+d}+2\xi}$$

with probability $P^n$ at least $1 - 1/n$. Replacing $2\xi$ by $\xi$, we obtain the assertion. $\square$

The proof of the lower bound in Theorem 3.5 is based on the construction of two families of distribution $P^\sigma$ and $Q^\sigma$ as well as Proposition D.15 (Tsybakov, 2008, Theorem 2.5) and the Varshamov-Gilbert bound in Lemma D.16 (Varshamov, 1957).

**Proposition D.15.** *Let $\{\Pi_h\}_{h\in H}$ be a family of distributions indexed over a subset $H$ of a semi-metric $(\mathcal{F}, \rho)$. Assume that there exist $h_0, \ldots, h_L \in H$ such that for some $L \geq 2$,*

   *(i) $\rho(h_j, h_i) \geq 2s > 0$ for all $0 \leq i < j \leq L$;*

   *(ii) $\Pi_{h_j} \ll \Pi_{h_0}$ for all $j \in [L]$;*

   *(iii) the average KL divergence to $\Pi_{h_0}$ satisfies $\frac{1}{L}\sum_{j=1}^L \mathrm{KL}(\Pi_{h_j}, \Pi_{h_0}) \leq \kappa\log L$ for some $\kappa \in (0, 1/8)$.*

*Let $Z \sim \Pi_h$, and let $\widehat{h}: Z \to \mathcal{F}$ denote any improper learner of $h \in H$. Then we have*

$$\sup_{h\in H} \Pi_h\big(\rho(\widehat{h}(Z), h) \geq s\big) \geq \big(\sqrt{L}/(1+\sqrt{L})\big)\big(1 - 2\kappa - 2\kappa/\log L\big) \geq (3 - 2\sqrt{2})/8.$$

**Lemma D.16** (Varshamov-Gilbert Bound). *Let $\ell \geq 8$ and $L \geq 2^{\ell/8}$. For all $0 \leq i < j \leq L$, let $\overline{\rho}_H(\sigma^i, \sigma^j) := \#\{\ell \in [L] : \sigma_\ell^i \neq \sigma_\ell^j\}$ be the Hamming distance. Then there exists a subset $\{\sigma^0, \ldots, \sigma^L\}$ of $\{-1, 1\}^\ell$ such that $\overline{\rho}_H(\sigma^i, \sigma^j) \geq \ell/8$, where $\sigma^0 := (1, \ldots, 1)$.*

**Theorem D.17.** *Let $\mathcal{F}$ be the set of all measurable predictors $f : \mathcal{X} \to \Delta^{M-1}$ and let $\mathcal{P}$ be a collection of all distributions $P$ which satisfies Assumption 3.1. In addition, let a learning algorithm that accepts data $D$ and outputs a predictor, be denoted as $\mathcal{A} : (\mathcal{X} \times \mathcal{Y})^n \to \mathcal{F}$. Then, we have*

$$\inf_{\mathcal{A}} \sup_{P\in\mathcal{P}} \mathcal{R}_{L_{\mathrm{CE}},P}(\mathcal{A}(D)) - \mathcal{R}^*_{L_{\mathrm{CE}},P} \gtrsim n^{-\frac{(1+\beta\wedge1)\alpha}{(1+\beta\wedge1)\alpha+d}}$$

*with probability $P^n$ at least $(3 - 2\sqrt{2})/8$.*

Theorem D.17 together with Theorem D.14 illustrates that the convergence rates of KLR shown in Theorem D.14 is minimax optimal up to an arbitrary small order $\xi$.

*Proof of Theorem D.17.* Without loss of generality, we investigate the binary classification, i.e., $M = 2$. Let the input space $\mathcal{X} := [0,1]^d$ and the output space as $\mathcal{Y} = \{-1,1\}$. Define $r := c_r n^{-1/((\beta \wedge 1+1)\alpha+d)}$ with the constant $c_r > 0$ to be determined later. In the unit cube $\mathcal{X}$, we find a grid of points with radius parameter $r$,

$$\mathcal{G} := \{(2k_1 r, 2k_2 r, \ldots, 2k_d r) : k_i = 1, 2, \ldots, (2r)^{-1} - 1, i = 1, 2, \ldots, d\}.$$

Denote $\ell := |\mathcal{G}| = ((2r)^{-1} - 1)^d$ and $\mathcal{G} = \{x_i\}_{i=1}^{\ell}$. Without loss of generality, we let $(6r)^{-1} - 1/3$ be an integer. Define the set of grid points $\mathcal{G}_1 := \{(2k_1 r, 2k_2 r, \ldots, 2k_d r) : k_i = 1, \ldots, (6r)^{-1} - 1/3, i \in [d]\} \subset \mathcal{G}$ and $\mathcal{G}_2 := \{(2k_1 r, 2k_2 r, \ldots, 2k_d r) : k_i = (r^{-1} - 2)/3, \ldots, (2r)^{-1} - 1, i \in [d]\} \subset \mathcal{G}$. Then we have $|\mathcal{G}_1| = |\mathcal{G}_2| = ((6r)^{-1} - 1/3)^d = 3^{-d}\ell$.

*Construction of the Conditional Probability Distribution $p(y|x)$.* Since we consider the binary classification case $\mathcal{Y} = \{-1,1\}$, we denote the conditional probability of the positive class as $p(1|x) := p(y = 1|x)$ and the nagative class as $p(-1|x) := 1 - p(1|x)$. Let the function $g_r(\cdot)$ on $[0, \infty]$ be defined by

$$g_r(z) := \begin{cases} 1 - z/r & \text{if } 0 \le z < r, \\ 0 & \text{if } z > r. \end{cases}$$

Moreover, let $a_U := 1/3 + r/3$ and $a_L := 2/3 - 7r/3$, which are close to $1/3$ and $2/3$, respectively. Given $\sigma \in \{-1,1\}^{\ell}$ and $c_{\alpha} > 0$, we define

$$p^{\sigma}(1|x) := \begin{cases} 1 - c_{\alpha} r^{\alpha} + c_{\alpha} \mathbf{1}\{\sigma_i = 1\} r^{\alpha} g_r^{\alpha}(\|x - x_i\|_2) & \text{if } x \in \bigcup_{x_i \in \mathcal{G}_2} B(x_i, r), \\ 1 - c_{\alpha} r^{\alpha} & \text{if } x \in [a_L, 1]^d \setminus \bigcup_{x_i \in \mathcal{G}_2} B(x_i, r), \\ 1/2 + c_{\alpha} \sigma_i r^{\alpha} g_r^{\alpha}(\|x - x_i\|_2) & \text{if } x \in \bigcup_{x_i \in \mathcal{G}_1} B(x_i, r), \\ 1/2 & \text{if } x \in [0, a_U]^d \setminus \bigcup_{x_i \in \mathcal{G}_1} B(x_i, r), \\ \in [1/2, 1 - c_{\alpha} r^{\alpha}] & \text{otherwise.} \end{cases}$$

*Construction of the Marginal Distribution $p(x)$.* First, we define the marginal density function $p(x)$ by

$$p(x) := \begin{cases} r^{d+(\alpha-1)\beta} \|x - x_i\|_2^{\beta-d} & \text{if } x \in \bigcup_{x_i \in \mathcal{G}_2} B(x_i, r) \setminus \{x_i\}, \\ \left(1 - \sum_{x_i \in \mathcal{G}_2} P(B(x_i, r))\right) \big/ \sum_{x_i \in \mathcal{G}_1} \mu(B(x_i, r)) & \text{if } x \in \bigcup_{x_i \in \mathcal{G}_1} B(x_i, r), \\ 0 & \text{otherwise.} \end{cases}$$

Let us verify that $p$ is a density function by proving $\int_{\mathcal{X}} p(x)\,dx = 1$. To be specific,

$$\begin{aligned} \int_{\mathcal{X}} p(x)\,dx &= \int_{\bigcup_{x_i \in \mathcal{G}_2} B(x_i, r)} p(x)\,dx + \int_{\bigcup_{x_i \in \mathcal{G}_1} B(x_i, r)} p(x)\,dx \\ &= |\mathcal{G}_2| \cdot P(B(x_1, r)) + \left(1 - |\mathcal{G}_2| \cdot P(B(x_1, r))\right) \\ &= 3^{-d}\ell \cdot P(B(x_1, r)) + 1 - 3^{-d}\ell P(B(x_1, r)) = 1, \end{aligned}$$

where $x_1 \in \mathcal{G}_2$. Finally, for any $\sigma^j \in \{-1, 1\}^{\ell}$, we write $P_X^{\sigma^j} := P_X$.

*Verification of the Hölder Smoothness.* First, $g_r$ satisfies the Lipschitz continuity with $|g(x) - g(x')| \le r^{-1}|x - x'|$. Moreover, using the inequality $|a^{\alpha} - b^{\alpha}| \le |a - b|^{\alpha}$, $\alpha \in (0, 1)$, we obtain that for any $x, x' \in B(x_i, r)$, $x_i \in \mathcal{G}_1 \cup \mathcal{G}_2$, there holds

$$\begin{aligned} |p^{\sigma}(1|x) - p^{\sigma}(1|x')| &= c_{\alpha} r^{\alpha} \left| g_r^{\alpha}(\|x - x_i\|_2) - g_r^{\alpha}(\|x' - x_i\|_2) \right| \\ &\le c_{\alpha} r^{\alpha} \left| g_r(\|x - x_i\|_2) - g_r(\|x' - x_i\|_2) \right|^{\alpha} \\ &\le c_{\alpha} r^{\alpha} \left| \|x - x_i\|_2/r - \|x' - x_i\|_2/r \right|^{\alpha} \le c_{\alpha} \|x' - x\|_2^{\alpha}. \end{aligned}$$

Therefore, $p^{\sigma}(y|x)$ satisfies the Hölder smoothness assumption.

*Verification of the SVB Condition.* Using the inequality $1 - (1 - x)^{1/\alpha} \le 1 - (1 - \alpha^{-1}x) = \alpha^{-1}x$ for any $x \in (0, 1)$ and $\alpha \in (0, 1)$, we obtain that for any $0 < t \le c_{\alpha} r^{\alpha}$,

$$P_X\left(p^{\sigma}(1|X) \ge 1 - t\right)$$

$$= \sum_{x_i \in \mathcal{G}_2} P_X\big(\{x \in B(x_i, r) : 1 - c_\alpha r^\alpha + c_\alpha r^\alpha \mathbf{1}\{\sigma_i = 1\} g_r^\alpha(\|x - x_i\|_2) \geq 1 - t\}\big)$$

$$\leq |\mathcal{G}_2| \cdot P_X\big(\{x \in B(x_1, r) : 1 - c_\alpha r^\alpha + c_\alpha r^\alpha g_r^\alpha(\|x - x_1\|_2) \geq 1 - t\}\big)$$

$$= 3^{-d} \ell \cdot P_X\big(\{x \in B(x_1, r) : r^\alpha - (r - \|x - x_1\|_2)^\alpha \leq c_\alpha^{-1} t\}\big)$$

$$= 3^{-d} \ell \cdot P_X\big(\{x \in B(x_1, r) : \|x - x_1\|_2 \leq r\big(1 - (1 - c_\alpha^{-1} r^{-\alpha} t)^{1/\alpha}\big)\}\big)$$

$$\leq 3^{-d} \ell \cdot P_X\big(\{x \in B(x_1, r) : \|x - x_1\|_2 \leq \alpha^{-1} c_\alpha^{-1} r^{1-\alpha} t\}\big)$$

$$= 3^{-d} \ell \cdot P_X\big(B(x_1, \alpha^{-1} c_\alpha^{-1} r^{1-\alpha} t)\big) = 3^{-d} \ell \int_{B(x_1, \alpha^{-1} c_\alpha^{-1} r^{1-\alpha} t)} p(x)\, dx$$

$$= \frac{2\pi^{d/2} \ell r^{d+(\alpha-1)\beta}}{3^d \Gamma(d/2)} \int_0^{\alpha^{-1} c_\alpha^{-1} r^{1-\alpha} t} \rho^{d-1} \rho^{\beta-d}\, d\rho$$

$$= \frac{2\pi^{d/2}}{3^d \Gamma(d/2) \beta (c_\alpha \alpha)^\beta} \ell r^d t^\beta \leq \frac{2\pi^{d/2}}{6^d \Gamma(d/2) \beta (c_\alpha \alpha)^\beta} t^\beta.$$

Choosing $c_\beta \geq 2\pi^{d/2}/(\Gamma(d/2)\beta(c_\alpha\alpha)^\beta 6^d) \vee 1$, the $\beta$-SVB in Assumption 3.1 *(ii)* is satisfied.

*Verification of the Conditions in Proposition D.15.* Let $L = 2^\ell - 1$. For the sake of convenience, for any $\sigma^j \in \{-1, 1\}^\ell$, $j = 0, \ldots, L$, we write $P^j := P^{\sigma^j}$ and $Q^j := Q^{\sigma^j}$. Denote $\sigma^0 := (-1, \ldots, -1)$ and $P^0 = P^{\sigma^0}$. Define the full sample distribution by

$$\Pi_j := P^{j \otimes n}, \qquad j = 0, \ldots, L.$$

Moreover, we define the semi-metric $\rho$ in Proposition D.15 by

$$\rho(p^i(\cdot|x), p^j(\cdot|x)) := \int_{\mathcal{X}} \left( p^i(1|x) \log \frac{p^i(1|x)}{p^j(1|x)} + p^i(-1|x) \log \frac{p^i(-1|x)}{p^j(-1|x)} \right) p(x)\, dx = \mathrm{KL}(P^i, P^j).$$

Therefore, for any predictor $\widehat{p}(y|x)$, we have $\mathcal{R}_{L_{\mathrm{CE}}, P}(\widehat{p}(y|x)) - \mathcal{R}^*_{L_{\mathrm{CE}}; P} = \rho(p(\cdot|x), \widehat{p}(\cdot|x))$. Now, we verify the first condition in Proposition D.15. For sufficient large $n$, we have $2\pi^{d/2} r^{\alpha\beta}/(6^d \Gamma(d/2)\beta) \leq 1/2$. For any $x \in \bigcup_{x_k \in \mathcal{G}_1} B(x_k, r)$, there holds

$$p(x) = \frac{1 - \sum_{x_i \in \mathcal{G}_2} P(B(x_i, r))}{\sum_{x_i \in \mathcal{G}_1} \mu(B(x_i, r))} = \frac{1 - 2\pi^{d/2}\Gamma(d/2)^{-1}\beta^{-1} 3^{-d} \ell r^{d+\alpha\beta}}{3^{-d}\ell\pi^{d/2} r^d / \Gamma(d/2+1)} \geq \frac{6^d \Gamma(d/2+1)}{2\pi^{d/2}}.$$

Denote the Hellinger distance between $P^i$ and $P^j$ as $H(P^i, P^j) := \int (\sqrt{dP^i} - \sqrt{dP^j})^2$. Using the inequality $\mathrm{KL}(P^i, P^j) \geq 2H^2(P^i, P^j)$, $\sqrt{a} - \sqrt{b} = (a - b)/(\sqrt{a} + \sqrt{b})$ and Lemma D.16, we obtain that for any $0 \leq i < j \leq L$, there holds

$$\rho\big(p^j(\cdot|x), p^i(\cdot|x)\big) = \mathrm{KL}(P^i, P^j) \geq 2H^2(P^i, P^j)$$

$$= 2 \int_{\mathcal{X}} \Big( \big(p^i(1|x)^{\frac{1}{2}} - p^j(1|x)^{\frac{1}{2}}\big)^2 + \big(p^i(-1|x)^{\frac{1}{2}} - p^j(-1|x)^{\frac{1}{2}}\big)^2 \Big) p(x)\, dx$$

$$= 2 \int_{\mathcal{X}} \big(p^j(1|x) - p^i(1|x)\big)^2 \Big( \big(p^j(1|x)^{\frac{1}{2}} + p^i(1|x)^{\frac{1}{2}}\big)^{-2} + \big(p^j(-1|x)^{\frac{1}{2}} + p^i(-1|x)^{\frac{1}{2}}\big)^{-2} \Big) p(x)\, dx$$

$$\geq \int_{\bigcup_{x_k \in \mathcal{G}_1} B(x_k, r)} \big(p^j(1|x) - p^i(1|x)\big)^2 \big(p^j(1|x) \vee p^i(1|x)\big)^{-1} p(x)\, dx$$

$$+ \int_{\bigcup_{x_k \in \mathcal{G}_2} B(x_k, r)} \big(p^j(1|x) - p^i(1|x)\big)^2 \big(p^j(-1|x) \vee p^i(-1|x)\big)^{-1} p(x)\, dx$$

$$\geq 2\rho_H(\sigma^i, \sigma^j) \int_{B(x_1, r)} (c_\alpha(r - \|x - x_1\|_2)^\alpha)^2 \cdot \frac{6^d \Gamma(d/2+1)}{2\pi^{d/2}}\, dx$$

$$+ \rho_H(\sigma^i, \sigma^j) \int_{B(x_1, r)} (c_\alpha(r - \|x - x_1\|_2)^\alpha)^2 (c_\alpha r^\alpha)^{-1} \cdot r^{d+(\alpha-1)\beta} \|x - x_1\|_2^{\beta-d}\, dx$$

$$\geq \rho_H(\sigma^i, \sigma^j) \left( 6^d c_\alpha^2 d \int_0^r (r - \rho)^{2\alpha} \rho^{d-1}\, d\rho + \frac{2\pi^{d/2} c_\alpha}{\Gamma(d/2)} r^{d+(\alpha-1)\beta-\alpha} \int_0^r (r - \rho)^{2\alpha} \rho^{\beta-d} \rho^{d-1}\, d\rho \right)$$

$$= \rho_H(\sigma^i, \sigma^j) \left( 6^d c_\alpha^2 dr^{2\alpha+d} \int_0^1 (1-t)^{2\alpha} t^{d-1} \, dt + \frac{2\pi^{d/2} c_\alpha}{\Gamma(d/2)} r^{d+(\beta+1)\alpha} \int_0^1 (1-t)^{2\alpha} t^{\beta-1} \, dt \right)$$

$$\geq \frac{\ell}{8} \left( 6^d c_\alpha^2 dr^{2\alpha+d} \text{Beta}(2\alpha+1, d) + \frac{2\pi^{d/2} c_\alpha}{\Gamma(d/2)} \text{Beta}(2\alpha+1, \beta) r^{d+\alpha(1+\beta)} \right)$$

$$\geq 2^{-d-3} \left( 6^d c_\alpha^2 dr^{2\alpha} \text{Beta}(2\alpha+1, d) + \frac{2\pi^{d/2} c_\alpha}{\Gamma(d/2)} \text{Beta}(2\alpha+1, \beta) r^{\alpha(1+\beta)} \right) \geq C_4 r^{\alpha(1+\beta\wedge1)},$$

where $C_4 := 2^{-d-3} \big( 6^d c_\alpha^2 d\text{Beta}(2\alpha+1, d) \wedge 2\pi^{d/2} c_\alpha \Gamma(d/2)^{-1} \text{Beta}(2\alpha+1, \beta) \big)$. By taking

$$s := 2^{-1} C_4 r^{\alpha(1+\beta\wedge1)} = 2^{-1} C_4 c_r^{\alpha(1+\beta\wedge1)} n^{-\frac{(1+\beta\wedge1)\alpha}{(1+\beta\wedge1)\alpha+d}},$$

we obtain $\rho(p^j(\cdot|x), p^i(\cdot|x)) \geq 2s$. The second condition of Proposition D.15 holds obviously. Therefore, it suffices to verify the third condition in Proposition D.15, which requires to consider the KL divergence between $P^j$ and $P^0$. Using Lemma 2.7 in Tsybakov (2008) and $1 - c_\alpha r^\alpha \geq 7/8$, we get

$$\text{KL}(P^j, P^0) \leq \int_{\mathcal{X}} \frac{(p^j(1|x) - p^0(1|x))^2}{p^0(1|x)p^0(-1|x)} p(x) \, dx$$

$$\leq \sum_{x_k \in \mathcal{G}_1} \mathbf{1}\{\sigma_k^j = 1\} \int_{B(x_k, r)} \frac{4c_\alpha^2 (r - \|x - x_k\|_2)^{2\alpha}}{(1/2 + c_\alpha r^\alpha)(1/2 - c_\alpha r^\alpha)} \cdot \frac{\Gamma(d/2+1)3^d}{\pi^{d/2}} (mr^d)^{-1} \, dx$$

$$+ \sum_{x_k \in \mathcal{G}_2} \mathbf{1}\{\sigma_k^j = 1\} \int_{B(x_k, r)} \frac{4c_\alpha^2 (r - \|x - x_k\|_2)^{2\alpha}}{c_\alpha r^\alpha (1 - c_\alpha r^\alpha)} \cdot r^{d+(\alpha-1)\beta} \|x - x_k\|_2^{\beta-d} \, dx$$

$$\leq \frac{20c_\alpha^2 \Gamma(d/2+1)3^d}{\pi^{d/2} r^d} \int_{B(x_k, r)} (r - \|x - x_k\|_2)^{2\alpha} \, dx$$

$$+ \frac{4c_\alpha m r^{d+(\alpha-1)\beta}}{r^\alpha (1 - c_\alpha r^\alpha)} \int_{B(x_k, r)} (r - \|x - x_k\|_2)^{2\alpha} \cdot \|x - x_k\|_2^{\beta-d} \, dx$$

$$\leq \frac{20c_\alpha^2 d3^d}{r^d} \int_0^r (r - \rho)^{2\alpha} \rho^{d-1} \, dx + \frac{8c_\alpha \pi^{d/2}}{\Gamma(d/2)} \frac{m r^{d+(\alpha-1)\beta}}{r^\alpha (1 - c_\alpha r^\alpha)} \int_0^r (r - \rho)^{2\alpha} \cdot \rho^{\beta-d} \rho^{d-1} \, dx$$

$$= 20c_\alpha^2 d3^d r^{2\alpha} \int_0^1 (1-t)^{2\alpha} t^{d-1} \, dt + \frac{8c_\alpha \pi^{d/2} c_\alpha \cdot m r^{d+\alpha(1+\beta)}}{\Gamma(d/2)(1 - r^\alpha)} \int_0^1 (1-t)^{2\alpha} \cdot t^{\beta-1} \, dt$$

$$\leq 20c_\alpha^2 d3^d \text{Beta}(2\alpha+1, d) r^{2\alpha} + \frac{8\pi^{d/2} c_\alpha \cdot \text{Beta}(2\alpha+1, \beta)}{2^d \Gamma(d/2)(1 - c_\alpha r^\alpha)} \cdot r^{\alpha(1+\beta)} \leq C_3 r^{\alpha(1+\beta\wedge1)}, \quad (77)$$

where $C_3 := 20c_\alpha^2 d3^d \text{Beta}(2\alpha+1, d) + 8\pi^{d/2} c_\alpha \text{Beta}(2\alpha+1, \beta)/(2^d \Gamma(d/2))$. By the independence of samples and Eq. (77), we have for any $j \in \{0, 1, \ldots, L\}$,

$$\text{KL}(\Pi_j, \Pi_0) = n\text{KL}(P^j, P^0) \leq C_3 n r^{(1+\beta\wedge1)\alpha}$$

$$= C_3 c_r^{(1+\beta\wedge1)\alpha+d} r^{-d} \leq C_3 c_r^{(1+\beta\wedge1)\alpha+d} 4^d \ell \leq 2(\log 2)^{-1} C_3 c_r^{(1+\beta\wedge1)\alpha+d} 4^d \log L.$$

By choosing a sufficient small $c_r$ such that $2(\log 2)^{-1} C_3 c_r^{(1+\beta\wedge1)\alpha+d} 4^d = 1/16$, we verify the third condition. Applying Proposition D.15, we obtain that for any estimator $\widehat{p}(y|x)$ built on $D$, with probability $P^n$ at least $(3 - 2\sqrt{2})/8$, there holds

$$\sup_{P \in \mathcal{P}} \mathcal{R}_{L_{\text{CE}}, P}(\widehat{p}(y|x)) - \mathcal{R}_{L_{\text{CE}}, P}^* \geq (C_4 c_r^{\alpha(1+\beta\wedge1)}/2) \cdot n^{-\frac{(1+\beta\wedge1)\alpha}{(1+\beta\wedge1)\alpha+d}},$$

which finishes the proof. $\qquad\square$

## D.2 PROOFS RELATED TO SECTION 3

### D.2.1 PROOFS RELATED TO SECTION 3.1

*Proof of Theorem 3.2.* Applying Bernstein's inequality in (Steinwart & Christmann, 2008, Theorem 6.12) to $\xi_i := \mathbf{1}\{Y_i = m\} - p(m), i \in [n_p]$, we get

$$|\widehat{p}(m) - p(m)| = \left| \frac{1}{n_p} \sum_{i=1}^{n_p} \xi_i \right| \leq \sqrt{\frac{2p(m)\tau}{n_p}} + \frac{2\tau}{3n_p}$$

with probability at least $1 - 2e^{-\tau}$. Using the union bound and $(a + b)^2 \leq 2(a^2 + b^2)$, we obtain

$$\sum_{m=1}^{M} |\widehat{p}(m) - p(m)|^2 \leq \sum_{m=1}^{M} \left( \sqrt{\frac{2p(m)\tau}{n_p}} + \frac{2\tau}{3n_p} \right)^2 \leq \sum_{m=1}^{M} \left( \frac{4p(m)\tau}{n_p} + \frac{4\tau^2}{9n_p^2} \right)$$

with probability at least $1 - 2Ke^{-\tau}$. Taking $\tau := \log(2Kn_p)$, we get

$$\sum_{m=1}^{M} |\widehat{p}(m) - p(m)|^2 \leq \frac{4\log(2Mn_p)}{n_p} + \frac{4\log^2(2Mn_p)}{9n_p^2} \lesssim \frac{\log n_p}{n_p} \tag{78}$$

with probability at least $1 - 1/n_p$. Combining Inequality (33), Propositions B.7 and B.9 in Wen et al. (2024), we get

$$\mathcal{R}_{L_{\mathrm{CE}}, Q}(\widehat{q}(y|x)) - \mathcal{R}^*_{L_{\mathrm{CE}}, Q} \lesssim \mathcal{R}_{L_{\mathrm{CE}}, P}(\widehat{p}(y|x)) - \mathcal{R}^*_{L_{\mathrm{CE}}, P} + \sum_{y \in [M]} (1/p(y) - 1/\widehat{p}(y))^2$$

$$\lesssim \mathcal{R}_{L_{\mathrm{CE}}, P}(\widehat{p}(y|x)) - \mathcal{R}^*_{L_{\mathrm{CE}}, P} + \sum_{y \in [M]} (p(y) - \widehat{p}(y))^2$$

$$\lesssim \mathcal{R}_{L_{\mathrm{CE}}, P}(\widehat{p}(y|x)) - \mathcal{R}^*_{L_{\mathrm{CE}}, P} + \log n_p / n_p$$

$$\lesssim n_p^{-\frac{(1+\beta \wedge 1)\alpha}{(1+\beta \wedge 1)\alpha + d} + \xi} + \log n_p / n_p \lesssim n_p^{-\frac{(1+\beta \wedge 1)\alpha}{(1+\beta \wedge 1)\alpha + d} + \xi}$$

with probability at least $1 - 2/n_p$, where the third last inequality is due to Eq. (78) and the second last inequality follows from Theorem D.14. $\qquad\square$

*Proof of Theorem 3.3.* Obviously, we have

$$\mathcal{T} := \{(P, Q) : P \text{ and } Q \text{ satisfy Eq. (2)}, P \in \mathcal{P}\} \supset \mathcal{T}' := \{(P, Q) : P = Q \in \mathcal{P}\}.$$

Theorem D.17 then yields that

$$\inf_{\mathcal{A}} \sup_{(P,Q) \in \mathcal{T}} \mathcal{R}_{L_{\mathrm{CE}}, Q}(\mathcal{A}(D_p)) - \mathcal{R}^*_{L_{\mathrm{CE}}, Q} \geq \inf_{\mathcal{A}} \sup_{(P,Q) \in \mathcal{T}'} \mathcal{R}_{L_{\mathrm{CE}}, Q}(\mathcal{A}(D_p)) - \mathcal{R}^*_{L_{\mathrm{CE}}, Q}$$

$$= \inf_{\mathcal{A}} \sup_{P \in \mathcal{P}} \mathcal{R}_{L_{\mathrm{CE}}, P}(\mathcal{A}(D_p)) - \mathcal{R}^*_{L_{\mathrm{CE}}, P} \gtrsim n_p^{-\frac{(1+\beta \wedge 1)\alpha}{(1+\beta \wedge 1)\alpha + d}}$$

holds with probability $P^{n_p}$ at least $(3 - 2\sqrt{2})/8$. This finishes the proof. $\qquad\square$

### D.2.2 PROOFS RELATED TO SECTION 3.2

*Proof of Theorem 3.4.* Combining Inequality (33), Propositions B.7 and B.9 in Wen et al. (2024), we get

$$\mathcal{R}_{L_{\mathrm{CE}}, Q}(\widehat{q}(y|x)) - \mathcal{R}^*_{L_{\mathrm{CE}}, Q} \lesssim \mathcal{R}_{L_{\mathrm{CE}}, P}(\widehat{p}(y|x)) - \mathcal{R}^*_{L_{\mathrm{CE}}, P} + \|w^* - \widehat{w}\|_2^2.$$

Using Proposition B.1 in Wen et al. (2024) and Theorem D.14, we obtain that for any $\xi \in (0, 1/2)$, there holds

$$\mathcal{R}_{L_{\mathrm{CE}}, Q}(\widehat{q}(y|x)) - \mathcal{R}^*_{L_{\mathrm{CE}}, Q} \lesssim \mathcal{R}_{L_{\mathrm{CE}}, P}(\widehat{p}(y|x)) - \mathcal{R}^*_{L_{\mathrm{CE}}, P} + \log n_q / n_q + \log n_p / n_p$$

$$\lesssim n_p^{-\frac{(1+\beta)\alpha}{(1+\beta)\alpha + d} + \xi} + \log n_q / n_q$$

with probability at least $1 - 1/n_p - 1/n_q$. $\qquad\square$

*Proof of Theorem 3.5.* Note that the lower bound of the excess risk consists of two parts depending on $n_p$ and $n_q$, respectively. Thus in the following, we prove the excess risk is larger than the two parts, respectively. First, we prove that the excess risk is larger than the first part related to $n_p$. To this end, we construct a sequence of the probability distribution $P$ as in Theorem D.17, and then we construct the probability distribution $Q$. To satisfy the label shift assumption, for any $\sigma^j \in \{-1, 1\}^\ell$, $j = 0, \ldots, 2^\ell - 1$, we let $Q^{\sigma^j} := P^{\sigma^j}$. For the sake of convenience, we write

$P^j := P^{\sigma^j}$ and $Q^j := Q^{\sigma^j}$. Correspondingly, we write $p^j(y|x) := P^{\sigma^j}(Y = y|X = x)$ and $q^j(y|x) := Q^{\sigma^j}(Y = y|X = x)$.

*Verification of the Conditions in Proposition D.15.* Let $L = 2^\ell - 1$, and we define the full sample distribution by $\Pi_j := P^{j\otimes n_p} \otimes Q_X^{j\otimes n_q}$, $j = 0, \ldots, L$. Moreover, we define the semi-metric $\rho$ in in Proposition D.15 by

$$\rho(q^i(\cdot|x), q^j(\cdot|x)) := \int_{\mathcal{X}} \left( q^T V i(1|x) \log \frac{q^i(1|x)}{q^j(1|x)} + q^i(-1|x) \log \frac{q^i(-1|x)}{q^j(-1|x)} \right) q(x)\, dx = \mathrm{KL}(Q^i, Q^j).$$

Therefore, for any predictor $\widehat{q}(y|x)$, we have $\mathcal{R}_{L_{\mathrm{CE}},Q}(\widehat{q}(y|x)) - \mathcal{R}^*_{L_{\mathrm{CE}},Q} = \rho(q(\cdot|x), \widehat{q}(\cdot|x))$. Since $Q^j = P^j$, the first and second conditions in Proposition D.15 can be verified in the same way as in Theorem D.17. Thus it suffices to verify the third condition in Proposition D.15. By the independence of samples, $Q_X^j = Q_X^0 = P_X$ and Eq. (77), we have for any $j \in \{0, 1, \ldots, L\}$,

$$\mathrm{KL}(\Pi_j, \Pi_0) = n_p \mathrm{KL}(P^j, P^0) + n_q \mathrm{KL}(Q_X^j, Q_X^0) = n_p \mathrm{KL}(P^j, P^0) \leq C_3 n_p r^{(1+\beta \wedge 1)\alpha}$$
$$= C_3 c_r^{(1+\beta \wedge 1)\alpha + d} r^{-d} \leq C_3 c_r^{(1+\beta \wedge 1)\alpha + d} 4^d \ell \leq 2(\log 2)^{-1} C_3 c_r^{(1+\beta \wedge 1)\alpha + d} 4^d \log L,$$

where the constant $C_3$ is defined in Eq. (77). By choosing a sufficient small $c_r$ such that $2(\log 2)^{-1} C_3 c_r^{(1+\beta \wedge 1)\alpha + d} 4^d = 1/16$, we verify the third condition. Apply Proposition D.15, we obtain that for any estimator $\widehat{q}(y|x)$ built on $D_p \cup D_q^u$, with probability $P^{n_p} \otimes Q_X^{n_q}$ at least $(3 - 2\sqrt{2})/8$, there holds

$$\sup_{(P,Q) \in \mathcal{T}} \mathcal{R}_{L_{\mathrm{CE}},Q}(\widehat{q}(y|x)) - \mathcal{R}^*_{L_{\mathrm{CE}},Q} \geq (C_4/2) \cdot n_p^{-\frac{(1+\beta \wedge 1)\alpha}{(1+\beta \wedge 1)\alpha + d}}. \tag{79}$$

Next, we construct a new class of probability to prove the second part $n_Q^{-1}$ of the lower bound. Let $w := 1/16$ and $\delta > 0$. Define the 1-dimension class conditional densities:

$$q(x|1) := \begin{cases} 4w\delta & \text{if } x \in [0, 1/4], \\ 4(1-\delta) & \text{if } x \in [3/8, 5/8], \\ 4(1-w)\delta & \text{if } x \in [3/4, 1], \\ 0 & \text{otherwise}; \end{cases} \qquad q(x|-1) := \begin{cases} 4(1-w)\delta & \text{if } x \in [0, 1/4], \\ 4(1-\delta) & \text{if } x \in [3/8, 5/8], \\ 4w\delta & \text{if } x \in [3/4, 1], \\ 0 & \text{otherwise}. \end{cases} \tag{80}$$

Let $\sigma \in \{-1, 1\}$ and $\delta$ will be chosen later. We specify the class probabilities in the following way: $p(1) := p(y = 1) := 1/2$, $q^\sigma(1) := q^\sigma(y = 1) := (1 + \sigma\theta)/2$. Then we compute the conditional probability function $q^\sigma(1|x)$. By the Bayes formula, we have

$$q^\sigma(1|x) = \frac{q^\sigma(1)q(x|1)}{q^\sigma(1)q(x|1) + q^\sigma(-1)q(x|-1)}$$
$$= \begin{cases} w(1 + \sigma\theta)/[w(1 + \sigma\theta) + (1 - w)(1 - \sigma\theta)] & \text{if } x \in [0, 1/4], \\ (1 + \sigma\theta)/2 & \text{if } x \in [3/8, 5/8], \\ (1 - w)(1 + \sigma\theta)/[(1 - w)(1 + \sigma\theta) + w(1 - \sigma\theta)] & \text{if } x \in [3/4, 1], \\ 1/2 & \text{otherwise}. \end{cases}$$

*Verification of the SVB Condition.* Denote

$$t_1 := \frac{w(1 - \theta)}{w(1 - \theta) + (1 - w)(1 + \theta)}, \qquad t_2 := \frac{w(1 + \theta)}{w(1 + \theta) + (1 - w)(1 - \theta)}.$$

For $t < t_1$, we have $Q^\sigma(q^\sigma(1|x) < t) = 0$. For $t \in [t_1, t_2)$, by taking $\theta := 1/(16\sqrt{n_q})$ and $\delta := c_\beta t_1^\beta$, we have

$$Q^\sigma(q^\sigma(1|x) < t) = \mathbf{1}\{\sigma = -1\}Q([0, 1/4])$$
$$= \mathbf{1}\{\sigma = -1\}\left((1 - \theta)w\delta/2 + (1 + \theta)(1 - w)\delta/2\right)$$
$$= \mathbf{1}\{\sigma = -1\}(1 + \theta - 2\theta w)\delta/2 \leq \delta = c_\beta t_1^\beta \leq c_\beta t^\beta.$$

Moreover, for $t \in [t_2, (1-\theta)/2)$, we have

$$Q^\sigma(q^\sigma(1|x) < t) = Q([0, 1/4]) = (1 + \theta - 2\theta w)\delta/2 \leq \delta \leq c_\beta t_1^\beta \leq c_\beta t^\beta.$$

Otherwise if $t \in [(1-\theta)/2, 1/2]$, by taking $c_\beta := 4^\beta$, there holds

$$Q^\sigma(q^\sigma(1|x) < t) = Q([0, 1/4] \cup [3/8, 5/8]) = (1 + \theta - 2\theta w)\delta/2 + 1 - \delta \leq 1 \leq c_\beta t^\beta.$$

We define our distribution class $\mathcal{K} := \{\Pi^\sigma : \sigma \in \{-1, 1\}\}$, where $\Pi^\sigma$ is defined as $\Pi^\sigma := P^{n_p} \otimes (Q_X^\sigma)^{n_q}$. Then using the inequality $\log((1+x)/(1-x)) \leq 3x$ for $0 \leq x \leq 1/2$, the Kullback-Leibler divergence between $\Pi^{-1}$ and $\Pi^1$ is

$$\begin{aligned}
\mathrm{KL}(\Pi^{-1}|\Pi^1) &= n_q \mathrm{KL}(Q_X^1|Q_X^{-1}) \leq n_q \mathrm{KL}(Q^1|Q^{-1}) \\
&= n_q\big(\log\big((1+\theta)/(1-\theta)\big)(1+\theta)/2 + \log\big((1-\theta)/(1+\theta)\big)(1-\theta)/2\big) \\
&= 2\theta n_q \log\big((1+\theta)/(1-\theta)\big) \leq 6\theta^2 n_q = 3/128.
\end{aligned}$$

Since $|\mathcal{K}| = 2$, $\Pi^1 \ll \Pi^{-1}$ and $|\mathcal{K}|^{-1}\mathrm{KL}(\Pi^{-1}|\mathcal{K}) = 3/256 < (\log 2)/8$. Then we calculate the excess risk. Define $q^\sigma(y|x) := Q^\sigma(Y = y|X = x)$ and $q^\sigma(y) := Q^\sigma(Y = y)$. The semi-metric $\rho$ is defined by

$$\begin{aligned}
\rho(q^1(y|x), q^{-1}(y|x)) &:= \int_\mathcal{X} \left( q^1(1|x) \log \frac{q^1(1|x)}{q^{-1}(1|x)} + q^1(-1|x) \log \frac{q^1(-1|x)}{q^{-1}(-1|x)} \right) q(x)\, dx \\
&= \int_{[0,1/4]} \log(t_2/t_1) q^1(1) q(x|1) + \log((1-t_2)/(1-t_1)) q^1(-1) q(x|-1)\, dx \\
&\quad + \int_{[3/8,5/8]} \log((1+\theta)/(1-\theta)) q^1(1) q(x|1) + \log((1-\theta)/(1+\theta)) q^1(-1) q(x|-1)\, dx \\
&\quad + \int_{[3/4,1]} \log(t_2/t_1) q^1(1) q(x|1) + \log((1-t_2)/(1-t_1)) q^1(-1) q(x|-1)\, dx \\
&= \theta \log((1+\theta)/(1-\theta)) + \theta t(-2w+1) \log((1+\theta-2\theta w)/(1-\theta+2\theta w)) \\
&\geq \frac{\theta^2}{1-\theta} + \frac{\theta^2 t(1-2w)^2}{1-\theta+2\theta w} \geq 256^{-1}(1 + (7/8)^2 4^\beta) n_q^{-1} =: 2c_2 n_q^{-1},
\end{aligned} \tag{81}$$

where the second last inequality is due to $\log(1+x) \geq x/2$ for $x \in (0, 1)$ and $c_2 := 512^{-1}(1 + (7/8)^2 4^\beta)$. By Proposition D.15, we then obtain that with probability $\Pi$ at least $(3 - 2\sqrt{2})/8$, there holds

$$\sup_{\Pi \in \mathcal{K}} \mathcal{R}_{L_{\mathrm{CE}}, Q}(\widehat{q}(y|x)) - \mathcal{R}_{L_{\mathrm{CE}}, Q}^* \geq c_2 n_q^{-1}. \tag{82}$$

Combining Eq. (79) and Eq. (82), we obtain that for any $\widehat{q}(y|x)$ built on $D_p \cup D_q$, with probability $P^{n_p} \otimes Q_X^{n_q}$ at least $(3 - 2\sqrt{2})/8$, there holds

$$\sup_{(P,Q) \in \mathcal{T}} \mathcal{R}_{L_{\mathrm{CE}}, Q}(\widehat{q}(y|x)) - \mathcal{R}_{L_{\mathrm{CE}}, Q}^* \geq c_\ell \left( n_p^{-\frac{(1+\beta \wedge 1)\alpha}{(1+\beta \wedge 1)\alpha + d}} + n_q^{-1} \right), \tag{83}$$

where $c_\ell := C_4/2 \wedge c_2$. This finishes the proof. $\qquad\square$

### D.2.3 PROOFS RELATED TO SECTION 3.3

*Proof of Theorem 3.6.* First, let $\bar{C}$ be the true matrix, i.e., $\bar{C}_{kj} := \mathbb{E}_{P_s(X|Y=j)} p(k|X)$. By the triangle inequality, we have

$$\begin{aligned}
|C_{kj} - \bar{C}_{kj}| &\leq |C_{kj} - \mathbb{E}_{P_s(X|Y=j)}\widehat{p}(k|X)| + |\mathbb{E}_{P_s(X|Y=j)}\widehat{p}(k|X) - \bar{C}_{kj}| \\
&\leq \left| \frac{1}{n_{s,j}} \sum_{i \in [n_s], Y_i = j} (\widehat{p}(k|X) - \mathbb{E}_{P_s(X|Y=j)}\widehat{p}(k|X)) \right| + \mathbb{E}_{P_s(X|Y=j)}|\widehat{p}(k|X) - p(k|X)|.
\end{aligned} \tag{84}$$

Applying Bernstein's inequality, we get

$$\left| \frac{1}{n_{s,j}} \sum_{i \in [n_s], Y_i = j} (\widehat{p}(k|X) - \mathbb{E}_{P_s(X|Y=j)}\widehat{p}(k|X)) \right| \leq \sqrt{\frac{2\tau}{n_{s,j}}} + \frac{2\tau}{3n_{s,j}} \leq 2\sqrt{\frac{\tau}{n_{s,j}}} \leq 2c_{s,1}\sqrt{\frac{\tau}{n_s}}$$

with probability at least $1 - 2e^{-\tau}$, where $c_{s,1} := (\bigwedge_{j \in [M]} p_s(j))^{-1}$. By the union bound, we have

$$\max_{j,k} \left| \frac{1}{n_{s,j}} \sum_{i \in [n_s], Y_i = j} \left( \widehat{p}(k|X) - \mathbb{E}_{P_s(X|Y=j)} \widehat{p}(k|X) \right) \right| \leq 2c_{s,1} \sqrt{\frac{3 \log(n_s M)}{n_s}} \qquad (85)$$

with probability at least $1 - 1/n_s$. Moreover, using the Bayes formula and the label shift assumption, we get

$$\mathbb{E}_{P_s(X|Y=j)} \left| \widehat{p}(k|X) - p(k|X) \right|$$

$$= \mathbb{E}_{P(X|Y=j)} \left| \widehat{p}(k|X) - p(k|X) \right| = \int_{\mathcal{X}} \left| \widehat{p}(k|x) - p(k|x) \right| \cdot p(x|j) \, dx$$

$$= \int_{\mathcal{X}} \left| \widehat{p}(k|x) - p(k|x) \right| \cdot \frac{p(x)p(j|x)}{p(j)} \, dx \leq p(j)^{-1} \int_{\mathcal{X}} \left| \widehat{p}(k|x) - p(k|x) \right| \cdot p(x) \, dx$$

$$\lesssim \left( \int_{\mathcal{X}} \left| \widehat{p}(k|x) - p(k|x) \right|^2 \cdot p(x) \, dx \right)^{1/2} \lesssim \left( \mathcal{R}_{L_{\mathrm{CE}},P}(\widehat{p}(y|x)) - \mathcal{R}^*_{L_{\mathrm{CE}},P} \right)^{1/2}$$

where the last two inequalities follow from $(\mathbb{E}X)^2 \leq \mathbb{E}X^2$ and Lemma B.4 in Wen et al. (2024), respectively. This together with Eq. (84) and Eq. (85) yields

$$\max_{j,k} |C_{kj} - \bar{C}_{kj}| \lesssim \left( \log n_s / n_s \right)^{1/2} + \left( \mathcal{R}_{L_{\mathrm{CE}},P}(\widehat{p}(y|x)) - \mathcal{R}^*_{L_{\mathrm{CE}},P} \right)^{1/2}$$

with probability at least $1 - 1/n_s - 1/n_p$. Using Theorem 4.2 in Meyer (1980), we get

$$\|\widehat{p} - p\|_1 \leq \|(C - \bar{C})A\|_1 \leq \|C - \bar{C}\|_1 \|A\|_1 \lesssim (\log n_s / n_s)^{1/2} + \left( \mathcal{R}_{L_{\mathrm{CE}},P}(\widehat{p}(y|x)) - \mathcal{R}^*_{L_{\mathrm{CE}},P} \right)^{1/2},$$

where $A$ is the group inverse of the matrix $Id - \bar{C}$. Combining Inequality (33), Propositions B.7 and B.9 in Wen et al. (2024), we obtain

$$\mathcal{R}_{L_{\mathrm{CE}},Q}(\widehat{q}(y|x)) - \mathcal{R}^*_{L_{\mathrm{CE}},Q} \lesssim \mathcal{R}_{L_{\mathrm{CE}},P}(\widehat{p}(y|x)) - \mathcal{R}^*_{L_{\mathrm{CE}},P} + \sum_{y \in [M]} (1/p(y) - 1/\widehat{p}(y))^2$$

$$\lesssim \mathcal{R}_{L_{\mathrm{CE}},P}(\widehat{p}(y|x)) - \mathcal{R}^*_{L_{\mathrm{CE}},P} + \sum_{y \in [M]} (p(y) - \widehat{p}(y))^2$$

$$\lesssim \mathcal{R}_{L_{\mathrm{CE}},P}(\widehat{p}(y|x)) - \mathcal{R}^*_{L_{\mathrm{CE}},P} + \|p - \widehat{p}\|_1^2$$

$$\lesssim n_p^{-\frac{(1+\beta \wedge 1)\alpha}{(1+\beta \wedge 1)\alpha + d} + \xi} + \log n_s / n_s$$

with probability at least $1 - 1/n_s - 1/n_p$, where the last two inequalities follow from $\|p - \widehat{p}\|_2^2 \leq \|p - \widehat{p}\|_1^2$ and Theorem D.14. This finishes the proof. $\qquad \square$

*Proof of Theorem 3.7.* Note that the lower bound of the excess risk consists of two parts. Thus in the following, we prove the excess risk is larger than the two parts, respectively. First, we prove that the excess risk is larger than the first part. Similar to the first part of proof of Theorem 3.5, we let $Q^{\sigma^j} = P^{\sigma^j} = S^{\sigma^j}$, where $j \in \{0, \ldots, 2^\ell - 1\}$. Then similar analysis as in Eq. (79) yields the lower bound

$$(n_p + n_s)^{-\frac{(1+\beta \wedge 1)\alpha}{(1+\beta \wedge 1)\alpha + d}} \asymp (n_p \vee n_s)^{-\frac{(1+\beta \wedge 1)\alpha}{(1+\beta \wedge 1)\alpha + d}}.$$

For the second part, the construction of distribution $P$ and $Q^\sigma$ totally follow the second part as in Eq. (80) and we let $S^\sigma = Q^\sigma$. Similar arguments for proving Eq. (83) show the lower bound $n_s^{-1}$. This together with the first part yields the conclusion. $\qquad \square$

# E   CONVERGENCE RATES FOR THE CLASSIFICATION LOSS

In this section, we present the convergence rates with respect to the classification loss for the three complex scenarios. To this end, we first combine the calibration inequality in Eq. (1) and the convergence rates with respect to the CE loss in Eq. (11), Eq. (13) and Eq. (15) yields the following results.

*(a)* In the long-tailed learning, we have

$$\mathcal{R}_{L_{\text{class}},Q}(\widehat{q}(y|x)) - \mathcal{R}^*_{L_{\text{class}},Q} \lesssim n_p^{-\frac{(1+\beta\wedge1)\alpha}{2(1+\beta\wedge1)\alpha+2d}+\xi/2}. \tag{86}$$

*(b)* In the label shift adaptation, we have

$$\mathcal{R}_{L_{\text{class}},Q}(\widehat{q}(y|x)) - \mathcal{R}^*_{L_{\text{class}},Q} \lesssim n_p^{-\frac{(1+\beta\wedge1)\alpha}{2(1+\beta\wedge1)\alpha+2d}+\xi/2} + (\log n_q/n_q)^{1/2}. \tag{87}$$

*(c)* In the transfer learning, we have

$$\mathcal{R}_{L_{\text{class}},Q}(\widehat{q}(y|x)) - \mathcal{R}^*_{L_{\text{class}},Q} \lesssim n_p^{-\frac{(1+\beta\wedge1)\alpha}{2(1+\beta\wedge1)\alpha+2d}+\xi/2} + (\log n_s/n_s)^{1/2}. \tag{88}$$

To derive the lower bound for the classification loss, we follow the analysis for the constructed probability distribution in Theorem 4.1 of Audibert & Tsybakov (2007). Since the constructed CCP lies within a range $[c_\phi, 1 - c_\phi]$ with a constant $c_\phi \in (0, 1/2)$, the distribution satisfies the SVB in Assumption 3.1 *(ii)* with any $\beta \in [0, \infty]$ and therefore their established lower bound $n^{-\alpha/(2\alpha+d)}$ holds under Assumption 3.1 for the standard classification.

*(a)* In the long-tailed learning, we let $Q := P$, then the lower bound would be

$$\inf_{\mathcal{A}} \sup_{P,Q} \mathcal{R}_{L_{\text{class}},Q}(\mathcal{A}(D_p)) - \mathcal{R}^*_{L_{\text{class}},Q} \gtrsim n_p^{-\frac{\alpha}{2\alpha+d}}. \tag{89}$$

*(b)* In the label shift adaptation, we have two parts of lower bounds. For the first part, we let $Q := P$ and thus get the lower bound $n_p^{-\alpha/(2\alpha+d)}$ by applying the conclusion in the standard classification. For the second part, we follow the constructed probability distribution in Theroerm 3.5 and obtain that the excess classification error

$$\rho(q^1(y|x), q^{-1}(y|x)) := \int_{\mathcal{X}} |2q^1(1|x) - 1| \cdot \mathbf{1}\left\{\max_y q^1(y|x) \neq \max_y q^{-1}(y|x)\right\} q^1(x)\, dx$$

$$= \int_{[3/8,5/8]} \theta\, q^1(x)\, dx = (1 - \delta)\theta = (1 - \delta)/(16\sqrt{n_q})$$

holds with probability at least a constant $c \in (0, 1)$. Similar analysis of Eq. (82) yields the lower bound of the order $n_q^{-1/2}$. Combining these two parts, we get

$$\inf_{\mathcal{A}} \sup_{P,Q} \mathcal{R}_{L_{\text{class}},Q}(\mathcal{A}(D)) - \mathcal{R}^*_{L_{\text{class}},Q} \gtrsim n_p^{-\frac{\alpha}{2\alpha+d}} + n_q^{-1/2} \tag{90}$$

with probability at least a constant $c \in (0, 1)$.

*(c)* In the transfer learning, similar analysis of Eq. (90) yields the lower bound

$$\inf_{\mathcal{A}} \sup_{P,Q,S} \mathcal{R}_{L_{\text{class}},Q}(\mathcal{A}(D)) - \mathcal{R}^*_{L_{\text{class}},Q} \gtrsim n_p^{-\frac{\alpha}{2\alpha+d}} + n_s^{-1/2} \tag{91}$$

holds with probability at least a constant $c \in (0, 1)$.

If $\beta \geq 1$, then the upper bounds in Eq. (86), Eq. (87) and Eq. (88) match the lower bounds in Eq. (89), Eq. (90) and Eq. (91), respectively. This demonstrates that the KLR-based approaches in Section 2 are minimax optimal with respect to the classification loss when the probability that CCP is less than a threshold increases sub-linearly as the threshold increases from zero, i.e. $\beta \geq 1$.

## F  EXPERIMENTAL DETAILS

Using the benchmark datasets, we construct the labeled data $D_p$ and the unlabeled test data $D_t^u := (X_i^{(t)})_{i=1}^{n_t}$ from the distribution $Q$ to evaluate performance. Additionally, we construct $D_q^u$ for label shift adaptation and $D_s$ for transfer learning. To achieve this, we first generate label distributions using either a uniform or Dirichlet distribution. Based on these distributions, we randomly resample data from the original benchmark datasets to create the required datasets. Specifically, we generate

10 labeled datasets $D_p$, and for each $D_p$, we create 10 test datasets $D_t^u$ and 10 additional datasets (e.g., $D_q^u$ or $D_s$) using different random seeds, resulting in a total of 100 repeated experiments.

*(a)* In the context of long-tailed learning, the label distribution $q(y)$ is set to be uniform, while the long-tailed label distribution $p(y)$ is generated using a Dirichlet distribution with parameter $\alpha = 1$. Based on $p(y)$, we resample $n_p$ samples from the original benchmark dataset to construct the labeled dataset $D_p$. Similarly, we resample $n_q$ samples according to $q(y)$ to generate the unlabeled test dataset $D_t^u$, which will be used for prediction, along with their corresponding true labels for evaluation.

*(b)* In domain adaptation under label shift, following the setup in Lipton et al. (2018), the source label distribution $p(y)$ is uniform, while the target label distribution $q(y)$ is generated using a Dirichlet distribution with $\alpha = 1$. Using these distributions, we construct $D_p$, $D_q^u$, and $D_t^u$ from the benchmark data, sampled according to $p(y)$, $q(y)$, and $q(y)$, respectively. The CCP estimator $\widehat{q}(y|x)$ is trained on the combined dataset $(D_p, D_q^u)$ and evaluated on the test dataset $D_t^u$.

*(c)* In transfer learning, while the labeled dataset $D_p$ is unavailable, we do have access to the CCP estimator $\widehat{p}(y|x)$, pre-trained on $D_p$ using the KLR method. Additionally, we utilize auxiliary data $D_s$, constructed by resampling $n_s$ samples from the remaining benchmark data based on a label distribution $s(y)$. This distribution is generated using a Dirichlet distribution with the parameter $\alpha = 10$.

Table 2: Data Descriptions in Three Complex Classification Scenarios

| Dataset | $n$ | $d$ | $M$ | Long-tailed | | Domain Adaptation | | | Transfer Learning | | |
|---|---|---|---|---|---|---|---|---|---|---|---|
| | | | | $n_p$ | $n_t$ | $n_p$ | $n_q$ | $n_t$ | $n_p$ | $n_s$ | $n_t$ |
| Dionis | 416188 | 61 | 355 | 14200 | 7100 | 14200 | 7100 | 14200 | 14200 | 5000 | 7100 |
| Gas Sensor | 13910 | 128 | 6 | 3000 | 1000 | 3000 | 1000 | 3000 | 3000 | 1000 | 1000 |
| Satimage | 6430 | 36 | 6 | 1500 | 1000 | 2400 | 1200 | 1200 | 1500 | 1000 | 1000 |

When fitting the KLR model to the source domain $D_p$, we select two key hyperparameters: the regularization parameter $C$ and the kernel coefficient $\gamma$. These are determined using 5-fold cross-validation. Unlike the commonly used classification loss, we use the CE loss as the criterion for cross-validation, as our primary objective is to accurately estimate $p(y|x)$. Specifically, $C$ is chosen from seven values evenly spaced on a logarithmic scale between $10^{-6}$ and $10^0$, while $\gamma$ is selected from seven values evenly spaced on a logarithmic scale between $2^{-6}$ and $2^0$.

