# OpenReview forum: "Optimal Learning of Kernel Logistic Regression for Complex Classification Scenarios"
_ICLR.cc/2025/Conference — ICLR 2025 Poster_

### Official Review · Reviewer_rSR9 · 2024-10-20

**Soundness:** 3
**Presentation:** 3
**Contribution:** 2
**Rating:** 6
**Confidence:** 3

**Summary:**

This paper explores the theoretical properties of kernel logistic regression (KLR) in addressing complex classification challenges, such as long-tailed learning, domain adaptation, and transfer learning. The study highlights KLR's strengths in conditional class probability (CCP) estimation and demonstrates that KLR-based algorithms can achieve minimax optimal convergence rates for cross-entropy (CE) loss under mild conditions.

This paper presents the relevant theory of kernel logistic regression in complex classification scenarios and provides thorough justification. While it is a solid theoretical work, it somewhat lacks practical guidance for real-world classification problems. Readers may find it difficult to extract feasible methods for addressing issues such as long-tailed distributions from the presented arguments.

**Strengths:**

1. This paper provides a theoretical analysis of optimal learning for kernel logistic regression in various complex scenarios, including long-tailed learning, domain adaptation, and transfer learning.
2. A detailed proof process is provided for the theories related to KLR and cross-entropy (CE).
3. Exploring and analyzing complex classification scenarios from a theoretical perspective is insightful and innovative.

**Weaknesses:**

1. Currently, kernel-based methods cannot compete with mainstream neural network methods in real-world classification tasks, such as ImageNet and iNaturalist 2018. As a result, the practical significance of this work may be significantly diminished.
2. To determine whether the theory is truly applicable in complex classification scenarios, experimental validation is needed. This paper lacks experimental results to verify whether the proposed theory aligns with real-world situations. By conducting experiments, further analysis could be done to identify in which cases the theory holds and in which cases it does not. This would provide readers with more valuable insights and practical references.

**Questions:**

See Weaknesses.

1. Based on these theories, how should we design classification models? Could you provide some examples explicitly addressing issues like long-tailed distributions? Alternatively, could you analyze existing models, such as LDAM-DRW [1] and Logit Adjustment, and suggest possible improvements?

[1] Kaidi Cao, Colin Wei, Adrien Gaidon, Nikos Arechiga,and Tengyu Ma. Learning imbalanced datasets with labeldistribution-aware margin loss. In NeurIPS, pages 1567–1578, 2019.

[2] Aditya Krishna Menon, Sadeep Jayasumana, Ankit Singh Rawat, Himanshu Jain, Andreas Veit, and Sanjiv Kumar. Long-tail learning via logit adjustment. In ICLR, 2021.

2. Data, models, and loss functions are key considerations in classification tasks. This paper has already provided analysis from the perspective of the model (kernel methods) and the loss function (CE). Typically, when dealing with complex tasks, data augmentation techniques such as rotation, flipping, crop, and noise addition are applied. How would these operations impact the proposed framework?

---

> ### Author Response · Authors · 2024-11-25
>
> **W1.** Currently, kernel-based methods cannot compete with mainstream neural network methods in real-world classification tasks, such as ImageNet and iNaturalist 2018. As a result, the practical significance of this work may be significantly diminished.
>
> **Response to W1.** While we acknowledge that kernel-based methods may not match the performance of neural networks on high-profile benchmark tasks such as ImageNet or iNaturalist 2018, their practical significance should not be overlooked.
>
> Kernel-based methods offer distinct advantages in smaller-scale, theoretically grounded, and domain-specific problems, making them a vital component of the machine learning toolbox. Specifically, kernel methods often outperform neural networks on small datasets, where overfitting is a concern, as they do not require the extensive training or large volumes of data that neural networks typically need. The regularization terms in kernel methods (e.g., the RKHS norm) provide explicit control over model complexity, facilitating a clear trade-off between fitting the data and generalizing to unseen examples. Moreover, KLR uses a linear combination of kernel basis functions to predict class conditional probabilities (CCPs), resulting in a convex optimization problem that guarantees convergence to a global minimum, ensuring predictable and stable solutions.
>
> Furthermore, kernel methods enable the explicit incorporation of domain-specific knowledge through tailored kernels, which define how data points are compared or measured for similarity. This feature is particularly valuable in specialized tasks, where leveraging domain expertise can significantly enhance performance. Additionally, the strong theoretical foundations of kernel methods make them especially well-suited for applications where interpretability and guarantees on generalization are essential.
>
> In summary, kernel methods are particularly effective in scenarios such as small-sample learning, high-dimensional low-sample-size problems, and tasks that require interpretability, integration of domain knowledge, robustness, and strong theoretical guarantees. On the other hand, neural networks are better suited for large-scale or unstructured data (such as images, videos, and audio) and highly complex tasks. Both approaches have distinct strengths, and the choice between them should be based on the specific characteristics of the problem and the practical requirements at hand.

---

> ### Author Response · Authors · 2024-11-25
>
> **W2.** To determine whether the theory is truly applicable in complex classification scenarios, experimental validation is needed. This paper lacks experimental results to verify whether the proposed theory aligns with real-world situations. By conducting experiments, further analysis could be done to identify in which cases the theory holds and in which cases it does not. This would provide readers with more valuable insights and practical references.
>
> **Response to W2.** To validate the theoretical results presented in Section 3, we examine the impact of varying sample sizes, such as $n_p$, $n_q$, and $n_s$, on the performance of CCP-based methods. Our experiments are conducted on the *Dionis* dataset from the OpenML science platform, selected for its large number of classes, which provides a challenging scenario for evaluating the methods.
>
> *(a)*
> In long-tailed learning, as shown in Figure 1(a) of the revised manuscript, the accuracy on the test data from $Q$ improves as $n_p$ increases. This trend is consistent with Theorem 3.2, which asserts that the excess CE risk on $Q$ decreases with larger $n_p$. According to the calibration inequality in Eq.(1), this reduction in excess CE risk leads to the classification error approaching its minimal possible value. Thus, Figure 1(a) offers empirical evidence supporting the theoretical result presented in Theorem 3.2.
>
> *(b)*
> In label shift adaptation, Figure 1(b) of the revised manuscript demonstrates that for a fixed $n_p$, the accuracy improves as $n_q$ increases from $500$ to $3000$. However, when $n_q$ is further increased from $3000$ to $5000$, the performance stabilizes, showing minimal improvement. This trend suggests that, given the labeled source domain data $D_p$ (i.e., a fixed $n_p$), a certain number of unlabeled target domain samples $n_q$ is sufficient for achieving efficient performance. Additionally, Figure 1(b) of the revised manuscript reveals that when $n_q$ is sufficiently large, increasing $n_p$ leads to higher accuracy. This observation validates the convergence rate in Theorem 3.4 and aligns with the subsequent discussion.
>
> *(c)*
> In transfer learning with label bias, Figure 1(c) of the revised manuscript shows that the effects of $n_p$ and $n_s$ on accuracy exhibit trends similar to those observed for $n_p$ and $n_q$ in label shift adaptation (as shown in Figure 1(b) of the revised manuscript). This similarity suggests that the findings in Figure 1(c) of the revised manuscript empirically validate the convergence rates established in Theorem 3.6 with respect to $n_p$ and $n_s$.
>
> Therefore, the empirical results demonstrate that the CCP-based methods effectively validate our theoretical findings, confirming their applicability to real-world datasets across the three complex classification scenarios.

---

> ### Author Response · Authors · 2024-11-25
>
> **Q1.** Based on these theories, how should we design classification models? Could you provide some examples explicitly addressing issues like long-tailed distributions? Alternatively, could you analyze existing models, such as LDAM-DRW [1] and Logit Adjustment, and suggest possible improvements?
>
> [1] Cao, Kaidi, et al. Learning imbalanced datasets with label-distribution-aware margin loss. In NeurIPS, 2019.
>
> [2] Menon, Aditya Krishna, et al. Long-tail learning via logit adjustment. In ICLR, 2021.
>
> **A1.** Eq.(7) and Eq.(8) in [2] establish that the optimal classifier $h_q^*$ under the distribution $Q$ can be expressed as
> \begin{align*}
> h_q^*(x) &= \arg\max_{m \in [M]} M^{-1} p(x|m) = \arg\max_{m \in [M]} p(x|m) = \arg\max_{m \in [M]} p(m|x) p(x) / p(m)
> \\\\
> &= \arg\max_{m \in [M]} p(m|x)/p(m) = \arg\max_{m \in [M]} \log p(m|x) -\log p(m),
> \end{align*}
> where $M$ denotes the number of classes in the considered classification task.
> Since the posterior probability $p(m|x)$ can be estimated as
> \begin{align*}
> \widehat{p}(m|x) := \frac{\exp(f_m(x))}{\sum_{j \in [M]} \exp(f_j(x))},
> \end{align*}
> it follows that $\log \widehat{p}(m|x) \propto f_m(x)$. Using this insight, [2] proposed the *Logit Adjustment classifier* defined as
> \begin{align*}
> \widehat{h}\_{\text{LA}}(x) = \arg\max_{m \in [M]} \log \widehat{p}(m|x) - \log \widehat{p}(m) = \arg\max\_{m \in [M]} f_m(x) - \log \widehat{p}(m).
> \end{align*}
> This formulation adjusts the *logits* $f_m(x)$ by subtracting the logarithm of the estimated class prior $\widehat{p}(m)$, effectively mitigating the influence of class imbalances on the classification decision.
>
> In our manuscript, leveraging Eq.(3) with $\widehat{w}(y) = 1/(M\widehat{p}(y))$, the CCP-based classifier can be expressed as
> \begin{align*}
> \widehat{h}\_q(x) &= \arg \max\_{m\in [M]} \widehat{q}(m|x) = \arg \max\_{m\in [M]} \frac{\widehat{p}(m|x)/
> \widehat{p}(m)}{\sum\_{j\in[M]} \widehat{p}(j|x)/ \widehat{p}(j)}
> \\\\
> &= \arg \max_{m\in [M]} \widehat{p}(m|x)/\widehat{p}(m) = \arg \max_{m\in [M]} \log \widehat{p}(m|x) - \log \widehat{p}(m).
> \end{align*}
>
> Therefore, the classifier $\widehat{h}\_q(x)$ induced by our CCP-based estimator $\widehat{q}(y|x)$ from Eq.(3) aligns with the logit adjustment classifier $\widehat{h}_{\text{LA}}$ introduced in [2].
> Consequently, our Theorem 3.2 provides a theoretical guarantee for the logit adjustment method as well, reinforcing its validity and effectiveness.
>
> The equivalence mentioned above has been included in the revised manuscript.
>
> [1] derives the ideal margin $\Delta_y$ for each class based on margin-based generalization bounds for the balanced error. Using this insight, to ensure the model $f$ to achieve the derived margins, [1] designs a new margin-based loss function
> \begin{align*}
> L_{\text{LDAM-HG}}(y, f(x)) := \max \Bigl( 0, \max_{j\neq y} f_j(x) - f_y(x) + \Delta_y \Bigr),
> \end{align*}
> where $f := (f_m)_{m \in [M]}$ represents the score function with each element $f_j: \mathbb{R}^d \to \mathbb{R}$. The ideal margin is given by $\Delta_y:= C \cdot n_y^{-1/4}$, where $C$ is a constant and $n_y$ is the sample size for the corresponding class $y \in [M]$.
>
>
> To facilitate the application of optimization techniques commonly used in neural networks—such as backpropagation (BP) and parameter tuning (e.g., adjusting $C$)—the authors introduce a smoothed version of the loss function, referred to as the LDAM loss. This loss is defined as
> \begin{align*}
> L_{\text{LDAM}}(y, f(x))
> & := - \log \frac{\exp(f_y(x) - \Delta_y)}{\exp(f_y(x)-\Delta_y) + \sum_{j \neq y} \exp(f_j(x))}
> \\\\
> & = - \log \frac{\exp(-\Delta_y) p_f(y|x)}{\exp(-\Delta_y) p_f(y|x) + \sum_{j \neq y} p_f(j|x)}.
> \end{align*}
> Notice that the transition from $L_{\text{LDAM-HG}}$ to $L_{\text{LDAM}}$ also involves normalizing the scores $f_j(x)$ to convert them into probabilities. Additionally, the maximum term $\exp(\max\_{j \neq y} f(j|x))$ is replaced by the summation $\sum_{j \neq y} \exp(f_j(x))$, further smoothing the loss function.
> By minimizing the LDAM risk, that is, $\mathbb{E}\_{(X, Y) \sim P} L_{\text{LDAM}}(Y, f(X))$, [1] derives a classifier given by $\arg \max_y f^*_y(x)$.
> However, a key limitation in [1] is the lack of theoretical guarantees regarding the minimizer $f^*$ and its alignment with the optimal classifier $\arg \max_y q(y|x)$. We believe that ensuring convergence within this framework is inherently challenging.
>
> In contrast, our CCP-based approach not only addresses this limitation but also provides robust and efficient theoretical guarantees for convergence. This makes it a more reliable and practical solution. Specifically, the score function $f^*_{\text{CE}}$, which minimizes the CE risk, satisfies $p_{f^*_{\text{CE}}}(y|x) = p(y|x)$. Furthermore, given the CCP function $p(y|x)$ under the distribution $P$, the conditional probability $q(y|x)$ can be recovered using $p(y|x)$ and $p(y)$, as detailed in Lines 140–144 of our original manuscript.

---

> ### Author Response · Authors · 2024-11-25
>
> **Q2.** Data, models, and loss functions are key considerations in classification tasks. This paper has already provided analysis from the perspective of the model (kernel methods) and the loss function (CE). Typically, when dealing with complex tasks, data augmentation techniques such as rotation, flipping, crop, and noise addition are applied. How would these operations impact the proposed framework?
>
> **A2.** To investigate the effectiveness of data augmentation for KLR on real-world datasets, we conduct numerical experiments comparing the performance of CCP-based methods on both the original and augmented data. For each augmentation technique—such as rotation, flipping, cropping, and noise—we randomly apply five different variations. Specifically, we generate five different orthogonal rotation matrices, or random 20\% features for flipping or cropping, or five random Gaussian noise with the scale parameter $\sigma = 0.1$. The augmented dataset is then constructed by concatenating the original samples with their augmented counterparts. With a sample size of $n_p=3000$ for *Dionis* and *Gas Sensor* datasets, and $n_p = 1500$ for *satimage* dataset, the augmented dataset of $D_p$ consists of a total of $6 n_p$ samples, where the *Dionis* and *Satimage* datasets from the OpenML Science Platform, and the *Gas Sensor* dataset from the UCI ML Repository.
> The following tables present the mean and standard deviation of the accuracy of the CCP-based methods on three classification datasets, under four different augmentation ways as well as without augmentation over 50 repeated experiments.
>
> Long-tailed Learning:
> |        |  Dionis | Gas Sensor  | Satimage |
> |---|---|---|---|
> | No augmentation  | 71.84 (2.25)  | 90.38 (3.36)  |  84.37 (1.31) |
> | Rotation  |  36.12 (7.30) |  80.20 (3.10) | 81.35 (1.87) |
> | Flipping  |  73.90 (1.30) |  89.42 (4.26) | 83.72 (1.10) |
> | Cropping  |  69.33 (2.74) |  87.77 (5.25) | 83.67 (1.03) |
> | Noise     |  73.72 (1.25) |  88.87 (3.55) | 84.45 (1.17) |
>
> Label shift adaptaiton:
> |        |  Dionis | Gas Sensor  | Satimage |
> |---|---|---|---|
> | No augmentation  | 62.46 (2.48)  | 96.62 (1.41)  |  92.41 (1.69) |
> | Rotation  |  14.98 (3.83) |  88.39 (4.29) | 91.32 (2.18) |
> | Flipping  |  63.11 (3.44) |  94.42 (2.32) | 92.33 (1.80) |
> | Cropping  |  55.70 (3.07) |  95.06 (2.23) | 92.54 (1.84) |
> | Noise     |  61.37 (3.85) |  93.55 (3.16) | 92.41 (1.73) |
>
> Transfer Learning:
> |        |  Dionis | Gas Sensor  | Satimage |
> |---|---|---|---|
> | No augmentation  | 73.63 (1.02)  | 90.60 (3.69)  | 84.24 (2.00)  |
> | Rotation  |  50.06 (2.72) |  81.83 (3.47) | 81.63 (1.23) |
> | Flipping  |  74.72 (0.97) |  88.02 (4.59) | 83.31 (1.59) |
> | Cropping  |  69.07 (2.38) |  88.47 (5.16) | 83.16 (1.33) |
> | Noise     |  74.28 (1.29) |  87.85 (4.61) | 83.75 (2.03) |
>
> From the above empirical results, we observe that the augmentation fails to improve the performance of the KLR-based methods. We conjecture that augmentation operations, such as rotation, flipping, and cropping (or their equivalents for non-image data), disrupt the inherent structure of the data, preventing it from providing the same benefits as in image-based tasks. For noise addition, the effect is less intuitive. It seems to depend on the signal-to-noise ratio: smaller amounts of noise may lead to improvements, while excessive noise could result in a deterioration of accuracy.

---

### Official Review · Reviewer_ygYV · 2024-10-27

**Soundness:** 3
**Presentation:** 3
**Contribution:** 3
**Rating:** 6
**Confidence:** 2

**Summary:**

The existing literature offers limited theoretical exploration of conditional class probabilities-based algorithms for classification in complex scenarios, particularly concerning the cross-entropy loss. In this paper, authors investigated the convergence rates of algorithms based on conditional class probabilities estimation using kernel logistic regression in complex classification scenarios, such as long-tailed learning, domain adaptation, and transfer learning.

**Strengths:**

1. A new oracle inequality for kernel logistic regression is established that holds with high probability.

2. The approximation error of kernel logistic regression w.r.t. the cross-entropy loss is derived.

3. The optimal convergence rates w.r.t. the cross-entropy loss for complex variants of classification problems is established.

**Weaknesses:**

1. As stated in the 3rd paragraph, the main motivation is "The existing literature offers limited theoretical exploration of conditional class probabilities-based algorithms for classification in complex scenarios, particularly concerning the cross-entropy loss.” It can be better to show the connection of the findings in this paper (i.e., kernel logistic regression for complex classification scenarios) and existing findings for standard classification scenario.

2. It can be better to show the connection of the three contributions in introduction. In the current version, it is even hard to find if there is some connection them with the title of this paper.

3. The main novelty is to restrict applications into three kinds of scenarios, i.e., long-tailed learning, domain adaptation, and transfer learning, which are called complex classification scenarios. I know all of them, but I don't know why they are put together in this paper (especially for long-tailed learning). Is there any logic to that? Why does the author focus on these three scenarios? The current version feels pieced together.

**Questions:**

Please explain the 3rd weakness mentioned above.

**Details Of Ethics Concerns:**

No.

---

> ### Author Response · Authors · 2024-11-25
>
> **W1.** As stated in the 3rd paragraph, the main motivation is ``The existing literature offers limited theoretical exploration of conditional class probabilities-based algorithms for classification in complex scenarios, particularly concerning the cross-entropy loss." It can be better to show the connection of the findings in this paper (i.e., kernel logistic regression for complex classification scenarios) and existing findings for standard classification scenarios.
>
> **Response to W1.** Although the KLR algorithm has been successfully applied to tasks such as long-tailed classification [1] and credit risk classification [2], and has been further developed into a more efficient classification method known as the import vector machine [3], the generalization bounds of KLR, particularly with respect to CE loss, remain largely unexplored. This gap exists for both standard and complex classification tasks.
>
> The limited theoretical study of KLR for both standard and complex classification problems primarily stems from the challenges associated with the CE loss, which KLR minimizes. The CE loss is unbounded, complicating error analysis in standard classification tasks. Specifically, as discussed in the introduction, the unbounded nature of the CE loss invalidates many standard oracle inequalities typically used for sample error analysis under bounded losses. Furthermore, the target function that minimizes the CE risk is also unbounded, which makes it difficult to construct a bounded approximation function with small approximation errors.
>
> In our manuscript, we tackle these challenges for standard classification in Section 4.2. Additionally, Section 4.1 establishes a connection between the error analysis in standard classification and that in complex classification tasks. This unified approach allows us to derive theoretical results for KLR-based methods that apply to both standard and complex scenarios, bridging the gap and advancing the theoretical understanding in both contexts.
>
> [1] Miho Ohsaki, Peng Wang, Kenji Matsuda, Shigeru Katagiri, Hideyuki Watanabe, and Anca Ralescu. Confusion-matrix-based kernel logistic regression for imbalanced data classification. TKDE, pages 1806-1819, 2017.
>
> [2] S. P. Rahayu, Jasni Mohammad Zain, A. Embong, and S.W. Purnami. Credit risk classification using Kernel Logistic Regression with optimal parameter. In ISSPA, pages 602-605, 2010.
>
> [3] Ji Zhu and Trevor Hastie. Kernel logistic regression and the import vector machine. In NeurIPS, pages 1081-1088, 2001.

---

> ### Author Response · Authors · 2024-11-25
>
> **W2.** It can be better to show the connection of the three contributions in introduction. In the current version, it is even hard to find if there is some connection them with the title of this paper.
>
> **Response to W2.** The main contribution of our work is establishing the minimax-optimality of the proposed CCP-based method. "Minimax-optimal" refers to estimators whose maximum risk, or worst-case error across all possible distributions, is the smallest achievable among all estimators. To demonstrate minimax-optimality of the KLR-based methods across the three complex classification scenarios, we aim to establish  matching upper and lower bounds for the excess CE risk of these methods.
>
> In the revised manuscript, we describe our contributions as follows.
>
> *(i)*
> We propose a novel decomposition framework that separates the CE loss into upper and lower components,  facilitating the decomposition of the excess CE risk of KLR into approximation and sample error terms. This innovative approach effectively tackles the analytical challenges posed by the unbounded nature of the CE loss, paving the way for the derivation of a new oracle inequality.
>
> *(ii)*
> We establish an upper bound for the approximation error terms of KLR with respect to the CE loss. To achieve this, we design a bounded approximation function that closely approximates the unbounded true CCP function while remaining above a specified positive threshold. By carefully selecting this threshold, the constructed approximation minimizes the resulting error.
>
> *(iii)*
> We derive an overall upper bound for the excess CE risk of KLR by integrating the results from *(i)* and *(ii)*. By judiciously selecting the appropriate KLR parameters, we further establish the convergence rates for KLR in the aforementioned complex classification scenarios.
>
> *(iv)*
> We derive lower bounds for the excess CE risk, which align with the convergence rates of KLR established in *(iii)*. This consistency demonstrates the minimax optimality of these rates, underscoring the effectiveness of existing approaches based on CCP estimation.
>
> *(v)*
> We conduct numerical experiments to demonstrate the effectiveness of CCP-based algorithms and to empirically validate the minimax optimal convergence rates established in *(iii)* and *(iv)*.
>
> Against this background, the connection between our contributions and the title of the article becomes evident. Specifically, in *(iii)* and *(iv)*, we demonstrate that, based on the theoretical results from *(i)* and *(ii)*, we can achieve "optimal learning" for "complex classification scenarios." Furthermore, we validate this "optimality" through real-world experiments, as outlined in *(v)*.
>
> **W3.** The main novelty is to restrict applications into three kinds of scenarios, i.e., long-tailed learning, domain adaptation, and transfer learning, which are called complex classification scenarios. I know all of them, but I don't know why they are put together in this paper (especially for long-tailed learning). Is there any logic to that? Why does the author focus on these three scenarios? The current version feels pieced together.
>
> **Response to W3.** While standard classification has been extensively studied, complex classification tasks have recently gained increasing importance. However, establishing theoretical guarantees for all complex classification tasks is clearly unrealistic. Therefore, we focus on a class of complex scenarios in which knowledge from labeled data under distribution $P$ on $\mathcal{X} \times \mathcal{Y}$ is transferred to address classification on a different distribution $Q$ over the same space, without access to any labeled data from $Q$. Crucially, we assume that the distributions $P$ and $Q$ satisfy the "label-shift" assumption, meaning the class-conditional probabilities $q(x|y)$ are assumed to be identical to $p(x|y)$. Key examples of this scenario include long-tailed learning, domain adaptation under label shift, and transfer learning under label bias.
>
> In future research, we expect that additional learning scenarios falling within the framework of transferring knowledge from the distribution $P$ to $Q$ under the "label-shift" setting can also be incorporated into our framework, along with corresponding design guidelines for CCP-based algorithms and theoretical guarantees.

---

> > ### Comment · Reviewer_ygYV · 2024-11-28
> >
> > Thanks for your great efforts to clarify my concerns.

---

### Official Review · Reviewer_Xy6p · 2024-11-02

**Soundness:** 3
**Presentation:** 3
**Contribution:** 3
**Rating:** 6
**Confidence:** 4

**Summary:**

This paper proposes a theoretical framework using Kernel Logistic Regression (KLR) to address the Conditional Class Probability (CCP) estimation problem in complex classification tasks. By introducing error decomposition and a new oracle inequality, the researchers demonstrate KLR’s optimal convergence rates in scenarios like long-tail learning, label shift, and transfer learning, underscoring its effectiveness in both theoretical and practical applications.

**Strengths:**

1. The authors decompose the unbounded CE loss into upper and lower bounded components, ensuring the boundedness of the sample error term. This decomposition enhances the model's interpretability and strengthens the robustness of using KLR for CCP estimation in complex tasks.
2. The proposed framework is versatile, offering potential applicability across various complex classification task scenarios and showing promise for real-world applications in long-tail distributions, label shift, and transfer learning.

**Weaknesses:**

1 The paper focuses on theoretical analysis without experimental validation on real-world datasets or applications. Although the theoretical results suggest that KLR has potential in complex classification tasks, without empirical evaluations, it’s difficult for readers to assess the method’s practical performance or compare it to other methods.
2 The paper highlights the theoretical advantages of KLR in complex classification tasks but does not extensively compare it with other commonly used kernel methods (e.g., Support Vector Machines, Gaussian Processes) or modern deep learning methods.
3 The theoretical proofs rely on several assumptions about the data distribution, such as smoothness of conditional probabilities and specific constraints on small values. These assumptions may not hold in all real-world applications, which could affect the generalizability of the method. For example, in long-tailed or multimodal data distributions, these theoretical assumptions might be difficult to satisfy, potentially resulting in degraded model performance.

**Questions:**

1. The experimental section is relatively limited, which limits validation of the framework's practical utility. Examples in the paper lack real-world data validation in complex classification settings, especially regarding long-tail learning and transfer learning. More comprehensive experiments with various real datasets are needed to demonstrate the theoretical findings effectively.
2. In Assumption 3.1, the authors assume smoothness of the conditional probability function and impose a lower-bound condition on probability values. However, these assumptions are challenging to satisfy in real-world data, where extreme nonlinearity or sparse data often preclude such smoothness, especially in high-dimensional settings. This limits the generalizability of the framework for practical applications.
3. Although kernel methods can theoretically enhance model expressiveness, the complexity of Kernel Logistic Regression increases significantly with sample size and feature dimensions. Particularly in high-dimensional datasets, memory consumption and computation time grow exponentially. While the authors suggest a Gaussian kernel-based approach, they do not provide further optimization strategies to alleviate computational costs. For large-scale datasets, it would be beneficial to explore additional methods for reducing computational burdens to enhance scalability.

---

> ### Author Response · Authors · 2024-11-25
>
> **W1.** The paper focuses on theoretical analysis without experimental validation on real-world datasets or applications. Although the theoretical results suggest that KLR has potential in complex classification tasks, without empirical evaluations, it’s difficult for readers to assess the method’s practical performance or compare it to other methods.
>
> **Response to W1.** We present the empirical results of the KLR-based methods on real-world datasets across three complex classification scenarios. Specifically, we compare the KLR-based methods, $\widehat{q}(y|x)$ in Eq.(3), with the baseline methods, $\widehat{p}(y|x)$ in Eq.(10), which are designed for standard classification tasks. As shown in Table 1 of the revised manuscript (or Answer A1 below), the KLR-based methods consistently achieve higher accuracy than the baseline methods. These results highlight the effectiveness of KLR-based algorithms in addressing complex learning challenges on real-world datasets.
>
>
> **W2.** The paper highlights the theoretical advantages of KLR in complex classification tasks but does not extensively compare it with other commonly used kernel methods (e.g., Support Vector Machines, Gaussian Processes) or modern deep learning methods.
>
> **Response to W2.** Since CCP-based methods for solving complex classification tasks require accurate CCP estimation, KLR is generally preferred over support vector machines (SVM) and Gaussian processes (GP) due to its straightforwardness and accuracy. KLR directly minimizes the CE loss, which targets conditional probability estimation. This makes it inherently suited for tasks that require precise conditional probabilities, as it optimizes for probability calibration within the model itself. On the other hand, GPs are ideal when uncertainty in the estimates is important, such as in Bayesian settings or when predictions benefit from a distribution over possible outputs. Although SVM is a widely used margin-based classifier, it is not specifically designed for CCP estimation. While it can incorporate post hoc techniques like Platt scaling for CCP estimation, it is generally less reliable for this purpose compared to KLR.
>
> Kernel methods such as KLR and neural networks are two mainstream approaches in machine learning, each with distinct advantages depending on the dataset type. KLR is often preferred for small-to-medium-sized, structured datasets where a kernelized feature space effectively captures the underlying patterns. In contrast, neural networks excel with large, unstructured datasets, such as images or audio, that require complex feature extraction. Looking ahead, we plan to extend our error analysis technique for KLR—particularly addressing how to theoretically handle unbounded CE loss—to the analysis of CCP-based algorithms using neural networks.
>
> **W3.** The theoretical proofs rely on several assumptions about the data distribution, such as smoothness of conditional probabilities and specific constraints on small values. These assumptions may not hold in all real-world applications, which could affect the generalizability of the method. For example, in long-tailed or multimodal data distributions, these theoretical assumptions might be difficult to satisfy, potentially resulting in degraded model performance.
>
> **Response to W3.** In long-tailed learning, the label distribution is highly imbalanced, resulting in some classes being underrepresented. Consequently, $p(y|x)$ may exhibit sharp variations in regions with very few samples, potentially reducing its smoothness. However, this does not entirely invalidate the smoothness condition. In other words, even with a long-tailed label distribution, Assumption 3.1(i) can still hold, as smoothness is not fundamentally incompatible with class imbalance. A decrease in smoothness simply means that the smoothness parameter $\alpha$ in Assumption 3.1(i), which quantifies the smoothness of $p(y|x)$, may become very small or even approach zero. Importantly, our theorems remain applicable in such scenarios, though this would result in a slower convergence rate.
>
> In the case of multimodal data distributions, $p(y|x)$ may exhibit multiple modes and varying levels of Hölder smoothness across different regions. To account for this variability, the global smoothness can be defined as the minimum of all local smoothness values, representing the lowest smoothness parameter $\alpha$ observed across the regions.
>
> Furthermore, the Small Value Bound in Assumption 3.1(ii) always holds with an exponent $\beta = 0$ for any distribution. This is because we can choose $c_0 \geq 1$ such that $P_X(p(m|X) \leq t) \leq 1 \leq c_0 t^0$ holds universally for any distribution $P$ and $t \in (0, 1]$. For further discussion, please refer to Answer A2 below.

---

> ### Author Response · Authors · 2024-11-25
>
> **Q1.** The experimental section is relatively limited, which limits validation of the framework's practical utility. Examples in the paper lack real-world data validation in complex classification settings, especially regarding long-tail learning and transfer learning. More comprehensive experiments with various real datasets are needed to demonstrate the theoretical findings effectively.
>
> **A1.** To demonstrate the effectiveness of the CCP-based algorithms in complex classification scenarios using real-world datasets, we conduct experiments on three multi-class classification datasets: the *Dionis* and *Satimage* datasets from the OpenML Science Platform, and the *Gas Sensor* dataset from the UCI ML Repository.
>
> We denote the CCP-based estimator $\widehat{q}(y|x)$ in Eq.(3), designed for complex classification tasks on distribution $Q$, as the *CCP* method. Similarly, we refer to the estimator $\widehat{p}(y|x)$ in Eq.(10), designed for standard classification on distribution $P$, as the *baseline* method. We compare the performance of these two methods to demonstrate the effectiveness of the CCP-based approach in complex classification tasks. The table below reports the mean and standard deviation of accuracy for the two methods over 100 repeated experiments.
>
> Long-tailed Learning:
> |        |  Dionis | Gassensor  | Satimage |
> |---|---|---|---|
> | Baseline  | 80.71 (0.87)  | 85.57 (5.97)  |  80.51 (3.70)  |
> |  CCP   |  83.67 (1.10) |  90.49 (4.47) | 84.56 (1.32) |
>
> Label shift:
> |        |  Dionis | Gassensor  | Satimage |
> |---|---|---|---|
> | Baseline  | 77.69 (1.58)  | 96.14 (0.89)  |  89.80 (3.90)  |
> |  CCP   |  82.73 (1.40) |  96.52 (1.30) | 96.46 (2.42) |
>
> Transfer Learning:
> |        |  Dionis | Gassensor  | Satimage |
> |---|---|---|---|
> | Baseline  | 80.72 (0.88)  |  85.97 (5.57)  | 80.51 (3.70) |
> |  CCP   |  84.22 (0.99) |  90.27 (4.78) | 84.47 (1.98) |
>
> The tables above display the label prediction accuracy of the CCP-based estimator $\widehat{q}(y|x)$ from Eq.(3) and the baseline estimator $\widehat{p}(y|x)$ from Eq.(10) on the distribution $Q$ across three complex classification scenarios. The consistently superior performance of the CCP-based method compared to the baseline underscores the effectiveness of CCP-based algorithms utilizing KLR in tackling these complex learning scenarios with real-world datasets.
>
> A detailed description of the data generation process and experimental setups for KLR can be found in the revised manuscript.
>
> Moreover, we provide Figure 1 in the revised manuscript to validate the theoretical results presented in Section 3 by investigating the effect of varying sample sizes ($n_p$, $n_q$, and $n_s$) on the performance of CCP-based methods. The following paragraphs provide a detailed discussion, which can also be found in Section 5 of the revised manuscript.
>
> *(a)*  In long-tailed learning, as shown in Figure 1(a) of the revised manuscript, the accuracy on the test data from $Q$ improves as $n_p$ increases. This trend is consistent with Theorem 3.2, which asserts that the excess CE risk on $Q$ decreases with larger $n_p$. According to the calibration inequality in Eq.(1), this reduction in excess CE risk leads to the classification error approaching its minimal possible value. Thus, Figure 1(a) offers empirical evidence supporting the theoretical result presented in Theorem 3.2.
>
> *(b)* In label shift adaptation, Figure 1(b) of the revised manuscript demonstrates that for a fixed $n_p$, the accuracy improves as $n_q$ increases from $500$ to $3000$. However, when $n_q$ is further increased from $3000$ to $5000$, the performance stabilizes, showing minimal improvement. This trend suggests that, given the labeled source domain data $D_p$ (i.e., a fixed $n_p$), a certain number of unlabeled target domain samples $n_q$ is sufficient for achieving efficient performance. Additionally, Figure 1(b) of the revised manuscript reveals that when $n_q$ is sufficiently large, increasing $n_p$ leads to higher accuracy. This observation validates the convergence rate in Theorem 3.4 and aligns with the subsequent discussion.
>
> *(c)*  In transfer learning with label bias, Figure 1(c) of the revised manuscript shows that the effects of $n_p$ and $n_s$ on accuracy exhibit trends similar to those observed for $n_p$ and $n_q$ in label shift adaptation (as shown in Figure 1(b) of the revised manuscript). This similarity suggests that the findings in Figure 1(c) of the revised manuscript empirically validate the convergence rates established in Theorem 3.6 with respect to $n_p$ and $n_s$.
>
> Therefore, the empirical results demonstrate that the CCP-based methods effectively validate our theoretical findings, confirming their applicability to real-world datasets across the three complex classification scenarios.

---

> ### Author Response · Authors · 2024-11-25
>
> **Q2.** In Assumption 3.1, the authors assume the smoothness of the conditional probability function and impose a lower-bound condition on probability values. However, these assumptions are challenging to satisfy in real-world data, where extreme nonlinearity or sparse data often preclude such smoothness, especially in high-dimensional settings. This limits the generalizability of the framework for practical applications.
>
> **A2.** The extreme nonlinearity suggests that $p(y|x)$ is not linear, while sparsity indicates a scarcity of data points in high-dimensional spaces. While both nonlinearity and sparsity may reduce the smoothness of $p(y|x)$, they do not entirely rule it out. In other words, despite the presence of nonlinearity or sparsity, Assumption 3.1 can still hold because smoothness is not inherently incompatible with these conditions. Even if the smoothness is substantially reduced, this simply means that the parameter $\alpha$ in Assumption 3.1(i), which quantifies the smoothness of $p(y|x)$, may become very small or even approach zero. Fortunately, our theorems remain applicable in such cases, leading to a slower convergence rate. Additionally, the Small Value Bound in Assumption 3.1(ii) always holds with an exponent $\beta = 0$ for any distribution. Even when $\beta = 0$, a valid convergence rate can still be derived as long as $\alpha > 0$. Notably, larger values of $\beta$ correspond to faster convergence rates.
>
>
> The Hölder smoothness of the conditional probability function $p(y|x)$ is a standard assumption commonly used in the analysis of classification algorithms within the non-parametric context [1,2,3,4]. By assuming the benign properties of $p(y|x)$ in Assumption 3.1, such as smoothness, we can derive upper bounds for various algorithms and a lower bound for the learning problem. If the orders of these bounds match, we can conclude that the algorithm is minimax optimal for the given learning problem, thus demonstrating its effectiveness. These standard assumptions provide a solid framework for fairly comparing different algorithms and assessing their theoretical effectiveness.
>
> [1] Kamalika Chaudhuri and Sanjoy Dasgupta. Rates of convergence for nearest neighbor classification. In NeurIPS, pages 3437–3445, 2014.
>
> [2] Maik Döring, Laszlo Gyorfi, and Harro Walk. Rate of convergence of k-nearest-neighbor classification rule. JMLR, pages 8485–8500, 2017.
>
> [3] Lirong Xue and Samory Kpotufe. Achieving the time of 1-NN, but the accuracy of k-NN. In AISTAT, pages 1628–1636. 2018.
>
> [4] Justin Khim, Ziyu Xu, and Shashank Singh. Multiclass classification via class-weighted nearest neighbors. arXiv:2004.04715, 2020.
>
>
>
> **Q3.** Although kernel methods can theoretically enhance model expressiveness, the complexity of Kernel Logistic Regression increases significantly with sample size and feature dimensions. Particularly in high-dimensional datasets, memory consumption and computation time grow exponentially. While the authors suggest a Gaussian kernel-based approach, they do not provide further optimization strategies to alleviate computational costs. For large-scale datasets, it would be beneficial to explore additional methods for reducing computational burdens to enhance scalability.
>
> **A3.** KLR can be memory-intensive and computationally expensive, particularly for large-scale datasets. However, several effective methods have been developed to address these challenges. For instance, approximate kernel methods like the Nyström approximation or random Fourier features replace the full kernel matrix with a lower-dimensional representation, significantly reducing both computational and storage requirements. Additionally, stochastic gradient descent (SGD) is commonly used for large-scale KLR, as it minimizes computational load by using a single or small batch of samples per gradient step iteration. To further enhance scalability, distributed learning methods [1,2,3,4] can be employed. These methods divide the dataset into subsets and aggregate models fitted on each subset, leveraging parallel computation on multi-core clusters to further reduce running time.
>
> [1] Zheng-Chu Guo, Lei Shi, and Qiang Wu. Learning Theory of Distributed Regression with Bias Corrected Regularization Kernel Network. JMLR, pages 1-25, 2017.
>
> [2] Haibin Cheng, Pang-Ning Tan, and Rong Jin. Efficient algorithm for localized support vector machine. TKDE, pages 537-549, 2010.
>
> [3] Mona Meister and Ingo Steinwart. Optimal learning rates for localized SVMs. JMLR, pages 1-44, 2016.
>
> [4] Cho-Jui Hsieh, Si Si, and Inderjit Dhillon. A divide-and-conquer solver for kernel support vector machines. In ICML, pages 566-574, 2014.

---

> > ### Comment · Reviewer_Xy6p · 2024-11-27
> >
> > The authors have carefully and thoughtfully addressed my comments and questions.

---

### Official Review · Reviewer_ZQYY · 2024-11-03

**Soundness:** 3
**Presentation:** 3
**Contribution:** 3
**Rating:** 6
**Confidence:** 3

**Summary:**

This paper provides a theoretical explanation for the empirical success of conditional class probability (CCP)-based method, particularly regarding the cross entropy (CE) loss. Specifically, this paper first truncates the CCP estimator to prevent large CE loss. Consequently, the authors discuss the properties of kernel logistic regression (KLR) in three complex classification scenarios. The findings indicate that KLR can achieve minimax optimal convergence rates for CE loss under mild conditions. Furthermore, the error analysis reveals that the excess CE risk for the complex classification scenarios can be reduced to that for the standard classification, establishing a new oracle inequality of KLR for CCP estimation.

**Strengths:**

-	This paper is well-written and organized. The theoretical analysis is detailed and coherent.
-	This paper has proposed a significant theoretical understanding of CCP-based approaches. The results are impressive.

**Weaknesses:**

Please see the **Question** section

**Questions:**

1. In Section 2.1, why is the label probability $q(y)$ set to be uniform while $p(y)$ is far from being uniform? Could you provide more context behind this choice?
2. The equation in the Theorem 3.5 is not labeled. Labeling this equation would improve readability and make it easier to reference later in the paper.
3. Note that the first item on the right-hand side of Equation (11), (13) and (14) is the same. The same pattern is observed in the results of lower bounds. Could you explain this phenomenon? What it might reveal about the fundamental nature of these classification tasks?
4. This paper presents important theoretical results. Could the authors explain how the results can help improve the practical solution of complex classification scenes? For example, what particular aspects of algorithm design or parameter selection that might be informed by these theoretical results? Including toy experiments might be beneficial to illustrate the practical impact.

---

> ### Author Response · Authors · 2024-11-25
>
> **Q1.** In Section 2.1, why is the label probability set to be uniform while is far from being uniform? Could you provide more context behind this choice?
>
> **A1.** Long-tailed learning addresses the challenge of training models on datasets with highly imbalanced label distributions. In such datasets, a *few* "head" classes contain *a large number* of samples, while *many* "tail" classes have *only a few* samples. These distributions, where class frequencies decrease sharply from head to tail, are referred to as "long-tailed." Consequently, it is often assumed that the label distribution in the training data is significantly non-uniform [1,2].
>
> In classification tasks on data following the "imbalanced" distribution $P$, the performance of a classifier $h$ is typically evaluated using the "overall" classification error, defined as
> $$
> \begin{align*}
> \text{Classification Error on } P
> & := P(h(X) \neq Y)
> \\\\
> & = \sum_{m\in [M]} p(m) \cdot P(h(X) \neq m | Y=m ).
> \end{align*}
> $$
> Here, $P(h(X) \neq m \mid Y = m)$ represents the class-specific error for class $m$. Consequently, the overall classification error can be expressed as a weighted average of class-specific errors, where the label probability $p(m)$ serves as the weight.
>
> In long-tailed scenarios, "tail" classes are associated with very small weights ($p(m) \ll 1$). As a result, a classifier that minimizes the overall classification error often prioritizes accuracy on "head" classes at the expense of performance on "tail" classes.
>
> In practice, "tail" classes are as important as "head" classes. Therefore, it is both necessary and meaningful to evaluate a classifier $h$ using the balanced error metric [3], which assigns equal weight $M^{-1}$ to each class, rather than relying on class proportions $(p(m))_{m \in [M]}$. The balanced error is defined as
> \begin{align*}
> \text{Balanced Error on } P
> := \sum\_{m\in [M]} M^{-1} \cdot P(h(X) \neq m | Y=m ).
> \end{align*}
>
>
> Now, consider the distribution $Q$ that satisfies a uniform label distribution $q(y) = 1/M$ and adheres to the label shift assumption $q(x|y) = p(x|y)$. Under these conditions, we have
> \begin{align*}
> \text{Balanced Error on } P
> &= \sum_{m\in [M]} M^{-1} \cdot P(h(X) \neq m | Y=m )
> \\\\
> &= \sum_{m\in [M]} Q(Y = m) P(h(X) \neq m | Y=m )
> \\\\
> &= \sum_{m\in [M]} Q(Y = m) Q(h(X) \neq m | Y=m )
> \\\\
> &= \sum_{m\in [M]} Q(h(X) \neq m, Y=m)
> \\\\
> &= Q(h(X) \neq Y) = \text{Classification Error on $Q$}.
> \end{align*}
> This demonstrates that minimizing the balanced error on the "imbalanced" distribution $P$ is equivalent to minimizing the classification error on the "balanced" distribution $Q$. Consequently, the uniform label distribution of $Q$ (i.e., $q(y) = 1/M$) aligns with the primary objective of long-tailed learning: ensuring equal importance is placed on performance across all classes.
>
> [1] Aditya Krishna Menon, Sadeep Jayasumana, Ankit Singh Rawat, Himanshu Jain, Andreas Veit, and Sanjiv Kumar. Long-tail learning via logit adjustment. In ICLR, 2021.
>
> [2] Kaidi Cao, Colin Wei, Adrien Gaidon, Nikos Arechiga, and Tengyu Ma. Learning imbalanced datasets with label-distribution-aware margin loss. In NeurIPS, pages 1567–1578, 2019.
>
> [3] Kay Henning Brodersen, Cheng Soon Ong, Klaas Enno Stephan, and Joachim M. Buhmann. The balanced accuracy and its posterior distribution. In ICPR, 2010.
>
>
>
>
> **Q2.** The equation in the Theorem 3.5 is not labeled. Labeling this equation would improve readability and make it easier to reference later in the paper.
>
> **A2.** Thank you for your suggestion. Following your advice, we have labeled the equation in Theorem 3.5 to improve readability and facilitate easier referencing.

---

> ### Author Response · Authors · 2024-11-25
>
> **Q3.** Note that the first item on the right-hand side of Equation (11), (13) and (14) is the same. The same pattern is observed in the results of lower bounds. Could you explain this phenomenon? What it might reveal about the fundamental nature of these classification tasks?
>
> **A3.** The right-hand side of Eq.(11), Eq.(13), and Eq.(15) (formerly Eq.(14)) corresponds to the convergence rates of CCP-based methods across three complex classification scenarios. All these rates share a common expression: $n_p^{-\frac{(1+\beta\wedge 1)\alpha}{(1+\beta\wedge 1)\alpha + d}}$. This expression represents the order of the upper and lower bounds of the excess CE risk $\mathcal{R}\_{L_{\mathrm{CE}},P}(\widehat{p}(y|x)) - \mathcal{R}\_{L_{\mathrm{CE}},P}^*$ with respect to the distribution $P$, which also appears in the error decomposition inequalities in Eq.(17), Eq.(18), and Eq.(19) for all three scenarios. This demonstrates that the CCP estimation error, $\mathcal{R}\_{L_{\mathrm{CE}},P}(\widehat{p}(y|x)) - \mathcal{R}\_{L_{\mathrm{CE}},P}^*$, is central to the performance of CCP-based algorithms across different classification settings. Even in standard classification, where $Q = P$ and $\widehat{q}(y|x) = \widehat{p}(y|x)$, this relationship still holds:
> \begin{align*}
> \mathcal{R}\_{L_{\mathrm{CE}},Q}(\widehat{q}(y|x)) - \mathcal{R}\_{L_{\mathrm{CE}},Q}^* = \mathcal{R}\_{L_{\mathrm{CE}},P}(\widehat{p}(y|x)) - \mathcal{R}\_{L_{\mathrm{CE}},P}^*.
> \end{align*}
>
> From the standard case, together with Eq.(17)-(19) for the complex scenarios, it is clear that the excess CE risk $\mathcal{R}\_{L_{\mathrm{CE}},P}(\widehat{p}(y|x)) - \mathcal{R}\_{L_{\mathrm{CE}},P}^*$ is always present. This reflects that the CCP estimation on $P$ is fundamental and indispensable to CCP-based algorithms for classification tasks on $Q$. In addition to this essential term, the additional terms in Eq.(17)-(19) reflect the missing information compared to standard classification problems where $Q = P$. Specifically:
>
> *(a)* In long-tailed learning, $p(y)$ is unknown and needs to be estimated. As a result, we incur a cost of $\log n_p / n_p$ for estimating $p(y)$, as shown in Eq.(17). Since $\log n_p / n_p$ decays faster than $n_p^{-\frac{(1+\beta\wedge 1)\alpha}{(1+\beta\wedge 1)\alpha + d}}$, the latter term dominates the convergence rate, and thus the former disappears in Eq.(11).
>
> *(b)* In label shift adaptation, both $p(y)$ and $q(y)$ are unknown. Therefore, extra errors $\log n_p / n_p$ and $\log n_q / n_q$ are introduced in Eq.(18).
>
> *(c)* In transfer learning, $p(y)$ is unknown and $D_p$ is inaccessible, so we must use the auxiliary set $D_s$ to estimate $p(y)$. This leads to the additional term $\log n_s / n_s$ in Eq.(19).
>
> In summary, the shared first term in Eq.(17)-(19), i.e., the excess CE risk with respect to $P$, highlights the common nature of these complex classification problems. The different additional terms in Eq.(17)-(19) reflect the distinctions in the problem settings.

---

> ### Author Response · Authors · 2024-11-25
>
> **Q4.** This paper presents important theoretical results. Could the authors explain how the results can help improve the practical solution of complex classification scenes? For example, what particular aspects of algorithm design or parameter selection might be informed by these theoretical results? Including toy experiments might be beneficial to illustrate the practical impact.
>
> **A4.** Our theoretical results provide guidance on collecting datasets $D_p$, $D_q^u$, and $D_s$ that are as small as possible while still ensuring strong performance.
>
> *(a)* In long-tailed learning, Theorem 3.2 demonstrates that the excess CE risk on $Q$ decreases as $n_p$ increases. Combined with the calibration inequality in Eq.(1), this implies that classification accuracy approaches its maximum possible value as $n_p$ tends to infinity. This theoretical result suggests that once a certain sample size $n_p$ is reached, collecting additional data for $D_p$ no longer significantly improves the performance of CCP-based methods. This insight provides a guideline for determining when to stop collecting more samples for $D_p$. Specifically, in addition to the current dataset $D_p$, we collect $n_{\Delta}$ samples from $P$. If the accuracy of the new model, fitted on $n_p + n_{\Delta}$ samples, does not exceed the accuracy of the current model fitted on $D_p$ by a given threshold, we can stop further data collection.
>
> As shown in Figure 1(a) of the revised manuscript, as $n_p$ increases, the accuracy of the CCP-based estimator fitted on $D_p$ improves, though the rate of improvement slows down. This empirically demonstrates that after reaching a certain sample size, further increases in $n_p$ become less cost-effective.
>
> *(b)*
> In label shift adaptation, the source data $D_p$ typically represents historical data or data collected from other sources, while the target data $D_q^u$ is typically collected more recently or from a nearby source. Therefore, we consider a fixed sample size $n_p$ for the source domain, while the sample size $n_q$ for the target domain gradually increases.
>
> Theorem 4.4 shows that the convergence rate of the CCP estimator $\widehat{q}(y|x)$ is primarily influenced by the larger of the two terms, $n_p^{-\theta}$ and $n_q^{-1}$, where $\theta := \frac{(1+\beta \wedge 1)\alpha}{(1+\beta \wedge 1)\alpha + d}$. The terms $n_p^{\xi}$ and $\log n_q$ in Eq.(13) are not as critical to the rate of decay. As $n_q$ increases from zero to approximately $n_p^{\theta}$, $n_q^{-1}$ decreases from infinity to $n_p^{-\theta}$, resulting in a faster convergence rate that eventually matches the order of $n_p^{-\theta}$. If $n_q$ continues to grow beyond this point, the convergence rate in Eq.(13) remains unchanged. This demonstrates that, given the labeled source domain data, a certain number of unlabeled target domain samples are sufficient to achieve efficient performance.
>
> This insight leads us to a rule for determining when to stop collecting samples for the target domain data $D_q^u$. Specifically, in addition to the current dataset $D_q^u$, we also collect $n_{\Delta}$ unlabeled samples from $Q$. If the accuracy of the new model, fitted on $n_q + n_{\Delta}$ samples, does not exceed the accuracy of the model fitted on $D_q^u$ by a given threshold, we can stop further data collection.
>
> Figure 1(b) of the revised manuscript shows that, for a fixed $n_p$, the accuracy improves as $n_q$ increases from $500$ to $3000$. However, as $n_q$ continues to increase from $3000$ to $5000$, the performance remains largely steady. This trend illustrates that, with a fixed labeled source domain data size $n_p$, a certain number of unlabeled target domain samples $n_q$ is sufficient to achieve efficient performance. This finding provides empirical support for the rule of choosing an appropriate sample size $n_q$.
>
> *(c)*
> In transfer learning with label bias, the pre-trained data $D_p$ is fixed and unobserved, while the auxiliary data can be gradually collected. Therefore, we consider a fixed sample size $n_p$ for the pre-trained data, while the auxiliary data sample size $n_s$ increases over time.
>
> Theorem 3.6 presents the convergence rates in a form similar to Theorem 3.4, with $n_q$ replaced by $n_s$. This allows us to apply a similar rule for deciding when to stop collecting auxiliary samples for $D_s$, analogous to the rule for collecting unlabeled target domain samples $D_q^u$ in the label shift adaptation problem.
>
> Figure 1(c) of the revised manuscript shows that the effects of $n_p$ and $n_s$ on accuracy follow trends similar to those observed for $n_p$ and $n_q$ in label shift adaptation (as shown in Figure 1(b) of the revised manuscript). This similarity provides empirical support for the rule of choosing an appropriate sample size $n_s$ for the auxiliary data $D_s$.

---

> > ### Comment · Reviewer_ZQYY · 2024-11-28
> > **Response**
> >
> > I appreciate the effort and thoughtfulness you put into your responses. Your clarifications have addressed my questions.

---

### Meta-Review · Area_Chair_7SSM · 2024-12-21

**Metareview:**

The paper proposes a theoretical framework using Kernel Logistic Regression to address the Conditional
Class Probability (CCP) estimation problem in complex classification tasks.
Despite the lack of experimental analysis and weak experimental impact, reviewers generally believe that
the paper has relevant contributions and are happy to accept the paper.

**Additional Comments On Reviewer Discussion:**

most reviewers were happy about the rebuttals.

---

### Decision · Program_Chairs · 2025-01-22

Accept (Poster)